# CARE: Covariance-Aware and Rank-Enhanced Decomposition for Enabling Multi-Head Latent Attention

**Zhongzhu Zhou**[1,3], **Fengxiang Bie**[1], **Ziyan Chen**[1], **Zhenyu Zhang**[4], **Yibo Yang**[2],
**Junxiong Wang**[3], **Ben Athiwaratkun**[3], **Xiaoxia Wu**[3,†] **Shuaiwen Leon Song**[1,3*]
[1] University of Sydney, [2] King Abdullah University of Science and Technology,
[3] Together AI, [4] University of Texas at Austin

## Abstract

Converting pretrained attention modules such as *grouped-query attention* (GQA) into *multi-head latent attention* (MLA) can improve expressivity without increasing KV-cache cost, making it attractive for efficient inference. However, many practical conversion baselines rely on weight-only low-rank approximations (e.g., SVD-style initializations) and uniform rank allocation. They focus on minimizing the difference between weight matrices rather than on how those weights affect input activations, ignore the covariance structure of activations, and enforce uniform rank across layers—causing activation drift and degraded attention fidelity. To address these issues, we propose CARE, a *Covariance-Aware, Rank-Enhanced* MLA conversion pipeline under a fixed KV width. CARE introduces three key steps: (i) *activation-preserving factorization*, which aligns the approximation with the actual input activations rather than just the weights; (ii) *adjusted-rank allocation*, which spreads a fixed KV budget across layers by giving more capacity to layers that need it most; and (iii) *KV-parity mapping*, which reparameterizes the converted $K$ and $V$ to fit the MLA format while keeping the KV-cache size unchanged. Our method outperforms a uniform-rank SVD baseline on Qwen3-4B/30B-A3B-Instruct-2507 and Llama-3.1-8B/70B-Instruct, reducing one-shot perplexity by up to $215\times$ and improving mean accuracy by up to $1.70\times$ at matched KV budgets. With a brief post-SVD "healing" fine-tune, we fully recover the original model's accuracy.

## 1 Introduction

Large Language Models (LLMs) deliver impressive capabilities but at high inference cost, with the key–value (KV) cache in self-attention emerging as a primary memory and bandwidth bottleneck (Vaswani et al., 2017; Kwon et al., 2023). In the standard multi-head attention (MHA) formulation, each head materializes and caches its own keys and values at every decoding step, causing the KV footprint to grow linearly with sequence length and head count. To alleviate this, architecture variants such as multi-query attention (MQA), which shares a single $K, V$ across all heads, and grouped-query attention (GQA), which shares $K, V$ within head groups, have been adopted at scale to shrink KV cache size (Shazeer, 2019; Ainslie et al., 2023; Touvron et al., 2023; Jiang et al., 2023; Chowdhery et al., 2022; Shoeybi et al., 2019). While effective, these variants reduce the number of distinct key/value projections, which can limit attention expressivity and introduce quality regressions when compression is pushed aggressively.

A more recent line of work reframes the KV-cache problem as one of *learned low-rank representation* (Wang et al., 2020; Xiong et al., 2021). Multi-Head Latent Attention (MLA) compresses keys and values into low-dimensional *latent* vectors, caches only these latents, and restores expressivity with lightweight up-down projections at compute time (DeepSeek-AI Team, 2024). In practice, MLA can dramatically reduce KV size while preserving or even improving task accuracy by trading memory

---

*Equal advising.

†Corresponding author: shirely@together.ai, Code is available at https://github.com/FutureMLS-Lab/CARE.

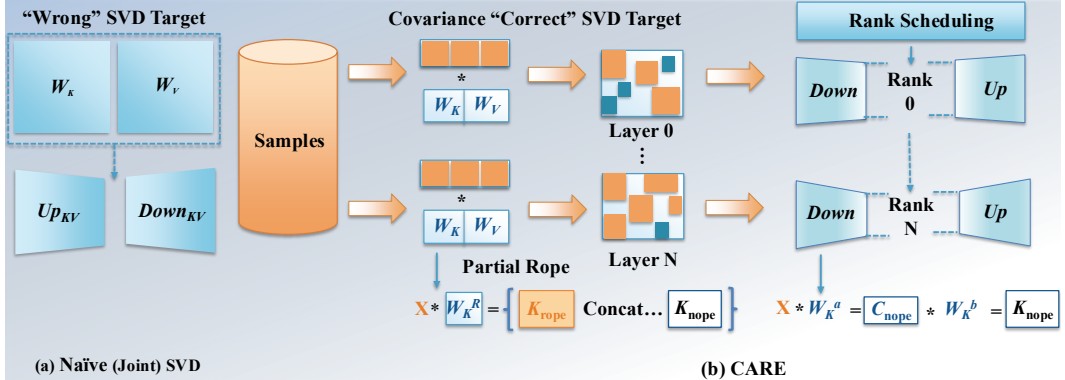

Figure 1: **(a) Naive MLA transfer**: jointly factorize $W_K^{(g)}$ and $W_V^{(g)}$ by SVD and truncate to a uniform per-layer rank, optimizing $\|W - \hat{W}\|_F$ while ignoring layerwise heterogeneity. **(b) CARE**: estimate activation covariance $C$, factorize $\sqrt{C}W$, unwhiten via $\sqrt{C}^{-1}$ to initialize MLA factors, and use the singular spectrum of $\sqrt{C}W$ for global dynamicrank scheduling under KV parity. This preserves activation geometry and yields a stronger one-shot initialization with less healing.

and communication for modest extra floating-point-operations (FLOPs) in the projections (DeepSeek-AI Team, 2024; Guo et al., 2025; Liu et al., 2024a; Geens & Verhelst, 2025). Despite these advantages, the ecosystem is dominated by pretrained MHA/GQA checkpoints (Touvron et al., 2023; Jiang et al., 2023; Yang et al., 2024a). Retraining large models from scratch under MLA is expensive, so a natural question arises:

> Can we *convert* strong, pretrained MHA/GQA models into MLA *post-hoc*, without increasing the KV budget and without incurring large performance loss?

Recent work has explored converting traditional attention (MHA/GQA) into multi-head latent attention (MLA) under fixed KV width. *TransMLA* (Meng et al., 2025) demonstrates that every GQA layer admits an equivalent MLA parameterization and proposes a practical post-training mapping followed by light finetuning. *MHA2MLA* (Ji et al., 2025) generalizes to MHA→MLA by addressing positional encoding mismatches (e.g., partial RoPE adjustments) and initializing $W_K$, $W_V$ with low-rank joint SVD before efficient recovery (Su et al., 2021b). *X-EcoMLA* (Li et al., 2025b) constructs MLA from pretrained MHA via structured SVD decomposition and cross-layer distillation–based parameter recycling for efficient latent projection initialization. In parallel, general-purpose SVD-based compression and approximation methods (e.g., SVD-LLM V2 (Wang et al., 2024b; 2025b) and activation-aware SVD (ASVD) (Yuan et al., 2023)) improve truncation and orientation beyond naïve weight SVD, while cache-centric baselines such as Palu (Chang et al., 2024) reduce KV memory via low-rank KV-cache projection. Together, these works establish MLA as a promising post-hoc target and highlight *low-rank factorization* as central to preserving pretrained knowledge under KV-constrained reparameterizations (Hu et al., 2022; Denil et al., 2013; Denton et al., 2014; Sainath et al., 2013; Eckart & Young, 1936).

However, direct SVD initialization has two key shortcomings. First, it minimizes error in weight space ($\|W - \hat{W}\|$) rather than activation space ($\|XW - X\hat{W}\|$), ignoring how the projection actually operates during decoding (Hassibi et al., 1993; Wang et al., 2024b; Yuan et al., 2023). This mismatch induces attention-logit drift even when the weight approximation is accurate. Second, it enforces a uniform rank across layers, neglecting differences in spectral structure. Layers with fast spectral decay are over-compressed, while those with slower decay are under-compressed, leading to fidelity loss and heavier reliance on post-conversion finetuning.

To address the above two shortcomings, we propose CARE, a **C**ovariance-**A**ware, **R**ank-**E**nhanced conversion pipeline, as shown in Fig. 1. First, CARE makes the decomposition *activation-aware*: rather than applying vanilla SVD to $W$, we solve a whitened approximation problem by applying

SVD to $\sqrt{C}W$ and then unwhitening to obtain $\hat{W}$ [1], where $C$ summarizes input activation covariance estimated from a modest calibration set. This ensures that dominant activation directions are preserved and substantially reduces attention-logit error before any finetuning. Second, CARE is *rank-adaptive*: it distributes a fixed KV budget across layers and heads based on their singular spectra, allocating higher rank to spectrally complex matrices and lower rank to intrinsically low-rank ones, akin in spirit to budgeted, importance-aware adapter methods (Zhang et al., 2023; Valipour et al., 2023; Hu et al., 2022; Wang et al., 2025a). This budgeted, importance-aware scheduling maintains fidelity under the KV constraint while reducing reliance on post-conversion finetuning.

**Contributions.**

- **Activation-aware Initialization.** We propose a covariance-aware factorization that minimizes activation error $\|XW - X\hat{W}\|$ (rather than weight error), implemented via SVD on a whitened operator and subsequent unwhitening. This preserves attention logits more faithfully at equal KV budget to initialize MLA weights.

- **Rank-adaptive scheduling under fixed KV width.** We introduce a singular-value-guided allocation that distributes rank unevenly across layers/heads and the $\{K, V\}$ matrices, matching spectral difficulty and improving zero-shot fidelity compared with uniform ranks.

- **KV-parity mapping and practical pipeline.** We derive a KV-parity reparameterization for MLA conversion and integrate the above techniques into a practical conversion pipeline CARE-converted models exhibit lower activation error and improved task quality over naive (joint) SVD baselines at equal KV cost, while requiring less data to recover residual gaps.

## 2 NAÏVE (JOINT) SVD IS NOT ENOUGH FOR MLA TRANSFER

Multi-head latent attention (MLA) transfer is often initialized with singular value decomposition (SVD), either per matrix or via a joint factorization across related matrices (e.g., $W_K$, $W_V$). While convenient, this practice implicitly optimizes weight-space error $\|W - \hat{W}\|_F$ and assumes that the spectrum alone reveals task importance. In this section we show both assumptions break down in practice and, consequently, naïve (joint) SVD is an unreliable recipe for high-fidelity MLA transfer under a fixed KV budget.

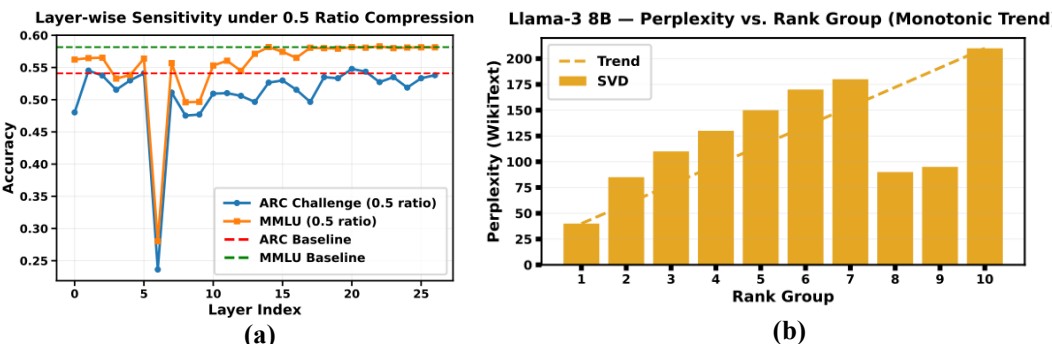

Figure 2: (a) Accuracy under 50% rank reduction applied one layer at a time in DeepSeek-V2-Lite, measured on ARC Challenge and MMLU. Sensitivity is strongly layer-dependent. (b) WikiText perplexity under grouped truncation of GQA attention in Llama-3-8B (layers 30–32). Singular-value groups are ordered by magnitude; the resulting non-monotone degradation shows that singular values alone are an imperfect proxy for MLA conversion quality.

**Observation 1: Accuracy-preserving rank is *not* uniform across layers.** Fig. 2 (a) exhibits pronounced layer-wise heterogeneity when we halve the rank of every layer: some layers tolerate aggressive reduction with negligible loss, whereas others incur sharp drops on ARC and MMLU. A

---

[1]Here 'whiten' means to use $\sqrt{C}W$ instead of $W$ as our target to compress, and 'unwhiten' means the inverse map which recovers the compressed $W$.

one-size-fits-all policy (uniform pruning or a fixed ratio per layer) either over-compresses fragile layers—degrading task performance—or under-compresses robust layers—wasting KV budget.

**Observation 2: Singular values are poor proxies for accuracy importance.** A common heuristic treats singular values as importance scores, expecting that truncating smaller values should least affect accuracy. We directly test this with a brute-force ablation: given $W = U\Sigma V^\top$, we set the $i$-th singular value to zero and reconstruct $\bar{W}^{(i)} = U \operatorname{diag}(\sigma_1, \ldots, 0, \ldots, \sigma_r) V^\top$, treating $V^T$ as compress of MLA and $U$ as expand of MLA. As shown in Fig. 2 (b), the link between singular-value magnitude and downstream accuracy is non-monotonic. We conjecture the root cause is mismatch of objectives and statistics: vanilla SVD minimizes weight error, not activation-space error $\|XW - X\hat{W}\|$ under the true (anisotropic) input distribution.

Thus, naïve (joint) SVD isn't enough: (i) the rank needed to keep accuracy varies by layer, and (ii) singular values alone don't reflect accuracy importance of initialization.

# 3 CARE: COVARIANCE-AWARE AND RANK-ENHANCED MLA CONVERSION

We propose **CARE**, a post-hoc conversion pipeline that maps a pretrained MHA or GQA layer to an MLA layer *at the same KV budget*, while explicitly minimizing *activation* error and *adapting ranks* to spectral difficulty. CARE revisits low-rank factorization through the lens of activation statistics and integrates covariance-weighted SVD and non-uniform rank allocation into a single, practical procedure compatible with multi-models' backbones.

**Notation.** Given a fixed layer with input activations $X \in \mathbb{R}^{T \times D}$ with length $T$ and embedding dimension $D$. The multi-head attention in our setup contains $n_h$ heads of size $d_h$, where we assume the output space corresponds with input space, formally $n_h d_h = D$. We denote $g_h$ to be the number of GQA groups, where for this layer: $Q = XW_Q$, $K = XW_K^{(g)} \in \mathbb{R}^{T \times (g_h d_h)}$, $V = XW_V^{(g)} \in \mathbb{R}^{T \times (g_h d_h)}$ with $g_h < n_h$, with $W_{\{\cdot\}}^{(g)}$ represents the weight matrix under the GQA setup. By contrast, an MLA layer with latent rank $r$ uses $K = (XW_K^a)W_K^b$, $V = (XW_V^a)W_V^b$, where $W_{\{\cdot\}}^a \in \mathbb{R}^{D \times r}$, $W_{\{\cdot\}}^b \in \mathbb{R}^{r \times (n_h d_h)}$. Only the latent $XW_{\{\cdot\}}^a \in \mathbb{R}^{T \times r}$ is cached, with $W_{\{\cdot\}}^b \in \mathbb{R}^{r \times (n_h d_h)}$ used to recover KV matrix.

## 3.1 PRELIMINARY: COVARIANCE FOR INPUT ACTIVATIONS

Let $X_b^{(l)} \in \mathbb{R}^{T_b \times D}$ be the $b^{th}$ batch of the domain activations with length $T_b$ at some layer $l$ (note that the length of tokens remains the same across layers, but can be different across batches). These batches of domain activations are used to calculate the extent of preserved rank in Sec. 3.2 and initialize the trainable parameters later in Sec. 3.3.

We then define $C^{(l)}$, the covariance matrix over all the $N$ batches at layer $l$, as follows:

$$C^{(l)} = \tfrac{1}{N} \sum_{b=1}^{N} (X_b^{(l)})^\top X_b^{(l)}.$$

Note that even if we denote the above $C^{(l)}$ to be covariance matrix, it is a slight different from the definition of actual covariance: we do not apply centralize and normalize operation here (see App. E for formal definition).

## 3.2 ADJUSTED-RANK SCHEDULING ACROSS LAYERS

Due to heterogeneous key/value spectra across different layers, the retained rank of each layer using MLA are supposed to be different. Let $\mathcal{W} = \{\widetilde{W}_K^{(l)}, \widetilde{W}_V^{(l)}\}_{l=1}^{L}$ represent the pretrained KV weights from Sec. B, where $L$ represents the number of layers. Note that $\mathcal{W}$ contains $2L$ weight matrices, each with dimension $D \times n_h d_h$ (recall $D = n_h d_h$).

Given a total key-rank budget $R_{\text{tot}}^{(K)}$ (same for value), we score the next rank by its *normalized residual reduction*. Let

$$\rho_{K,m}^{(l)} = \frac{\left(\sigma_{K,m}^{(l)}\right)^2}{\sum_{i=1}^{R_K^{(l)}}\left(\sigma_{K,i}^{(l)}\right)^2},$$

where $\sigma_{K,m}^{(l)}$ (same for $V$) is the $m^{th}$-largest singular value of $\sqrt{C^{(l)}}\widetilde{W}_K^{(l)}$, $C^{(l)}$ is the covariance matrix defined in Sec. 3.1, and $R_K^{(l)}$ is the rank of $\sqrt{C^{(l)}}\widetilde{W}_K^{(l)}$. For a layer currently assigned rank $r_K^{(l)}$, we define the priority of allocating one more rank as

$$s_K^{(l)}\left(r_K^{(l)}\right) = \frac{\rho_{K,r_K^{(l)}+1}^{(l)}}{1 - \sum_{m=1}^{r_K^{(l)}}\rho_{K,m}^{(l)}} = \frac{\left(\sigma_{K,r_K^{(l)}+1}^{(l)}\right)^2}{\sum_{m=r_K^{(l)}+1}^{R_K^{(l)}}\left(\sigma_{K,m}^{(l)}\right)^2}. \tag{1}$$

Appendix C proves that the Frobenius residual after rank-$r$ truncation equals the squared tail energy; normalizing by this residual makes layers with different spectral scales comparable. We allocate the budget $R_{\text{tot}}^{(K)}$ by greedy water-filling: starting from $r_K^{(l)} = C$ for all $l$ where $C$ is some constant, we repeatedly assign one rank to $l^\star = \arg\max_l s_K^{(l)}(r_K^{(l)})$ until the budget is exhausted; $V$ is handled identically with its own budget.

### 3.3 RETHINKING SVD WITH ACTIVATION COVARIANCE

A naive rank lowering attempts to minimize Frobenius error $\|W - \widehat{W}\|_F$, where $\widehat{W}$ represents the de-ranked matrix for compressing, and the original pretrained weight matrix to compress, $W := W_{\{\cdot\}}^{(l)} \in \mathcal{W}$, lies in layer $l$. For inference fidelity, we propose the relevant objective to be minimizing the empirical *activation error*. Focusing on compressing the certain weight matrix $W$, we denote $r$ to be the target compressed rank calculated in Sec. 3.2. Here $\{X_b := X_b^{(l)} \in \mathbb{R}^{T_b \times D}\}_{b=1}^N$ represents the small domain activation batches to compute the low-rank decomposition at layer $l$, where $N$ is the total number of batches, $T_b$ is the length of each batch and $C := C^{(l)}$ is the covariance matrix defined in Eq. 3.1.

Formally, we try to minimize the following:

$$\min_{\text{rank}(\widehat{W}) \leq r} \frac{1}{N} \sum_{b=1}^N \|X_b W - X_b \widehat{W}\|_F^2. \tag{2}$$

This optimization formalizes how well $\widehat{W}$ preserves $X_b W$ on relevant inputs $X_b$. From the definition of $C$, one can show that $\frac{1}{N}\sum_{b=1}^N \|X_b W - X_b \widehat{W}\|_F^2 = \|\sqrt{C}(W - \widehat{W})\|_F^2$; see App. D for the proof. For $K$ and $V$ separately, we compute

$$\sqrt{C}\, W = U\,\Sigma\,V^\top, \text{ then } \widehat{W} = \sqrt{C}^{-1} U_r\,\Sigma_r\,V_r^\top, \tag{3}$$

with $U_r, \Sigma_r, V_r$ the top-$r$ components of $U, \Sigma, V$. In practice, we use a shrinkage $\sqrt{C}_\lambda = (1 - \alpha)\sqrt{C} + \alpha\lambda I$ to ensure $C$ is invertible, with $\alpha \in (0, 1)$ and $\lambda > 0$.

Then we initialize the trainable parameters $W^a \in \mathbb{R}^{D \times r}$ and $W^b \in \mathbb{R}^{r \times (n_h d_h)}$ s.t. $W^a W^b$ equals the compressed matrix. We map SVD factors to MLA by

$$W^a \leftarrow \sqrt{C}^{-1} U_r\,\Sigma_r, \qquad W^b \leftarrow V_r^\top, \tag{4}$$

so that $W^a W^b = \widehat{W}$. The cached latent $X W^a \in \mathbb{R}^{T \times r}$ in MLA spans the principal activation subspace, where $X$ is the actual input at layer $l$.

### 3.4 100% MLA CONVERSION BY HEALING

Based on our initialization of down-and-up matrices $W^a$, $W^b$ and number of $T$ tokens, we attempt to encode positional information in the MLA-like attention mechanism. Given layer $l$, let $Q_t = X_t W_Q$

be the usual query at step $t$, and let $K_{C,t} = (X_t W_K^a) W_K^b$ and $V_{C,t} = (X_t W_V^a) W_V^b$ be the MLA generated keys/values. Following the decoupled RoPE design in (DeepSeek-AI Team, 2024), we bring a small RoPE channel of width $d_r$ by introducing *new trainable* matrices [2]:

$$W_Q^R \in \mathbb{R}^{D \times (n_h d_r)}, \qquad W_K^R \in \mathbb{R}^{D \times d_r},$$

where $W_K^R$ maps *directly* from the activation $X_t$ to a shared RoPE key of width $d_r$ (DeepSeek-like decoupled RoPE). Let $\mathcal{R}_t \in \mathbb{R}^{(n_h d_r) \times (n_h d_r)}$ denote the standard block-diagonal RoPE rotation at step $t$ (one $d_r \times d_r$ rotation per head), and let $R_t \in \mathbb{R}^{d_r \times d_r}$ denote the per-head rotation applied to the shared RoPE key. We compute:

$$Q_{R,t} = (X_t W_Q^R)\,\mathcal{R}_t, \qquad K_{R,t} = \operatorname{repeat}\big(X_t\,W_K^R\,R_t\big),$$

where $\texttt{repeat}$ replicates the shared RoPE key across heads, yielding $K_{R,t} \in \mathbb{R}^{1 \times (n_h d_r)}$. We then concatenate channels and compute attention:

$$Q_t^\star = \big[\,Q_t\,;\,Q_{R,t}\,\big], \quad K_t^\star = \big[\,K_{C,t}\,;\,K_{R,t}\,\big], \quad A_t = \operatorname{Softmax}\!\left(\frac{Q_t^\star (K_t^\star)^\top}{\sqrt{d_h + d_r}}\right), \quad O_t = A_t\,V_{C,t},$$

where $A_t$ denotes attention weights and $O_t$ denotes the layer output.

**Caching and a compact "join" form.** To preserve KV-cache efficiency, we cache only the KV-side latents $X_t W_K^a$ and $X_t W_V^a$, along with the *small* shared RoPE key $(X_t W_K^R) R_t$ (repeated across heads when forming $K_{R,t}$). Queries (including $Q_{R,t}$) are computed on-the-fly. Equivalently, the MLA-generated KV can be written in a compact joined form:

$$\operatorname{Concat}(K_{C,t},\,V_{C,t}) = \operatorname{Concat}(X_t W_K^a,\,X_t W_V^a)\,W_{\text{join}}, \tag{5}$$

where $\operatorname{Concat}(\cdot,\cdot)$ concatenates along the last (feature) dimension and $W_{\text{join}} = \operatorname{blkdiag}(W_K^b,\,W_V^b) \in \mathbb{R}^{2r \times 2D}$ jointly represents the two bilinear maps $W_K^a W_K^b$ and $W_V^a W_V^b$.

We penalize the low-rank decomposition by its cross-entropy classification error and KL-divergence imitation error, namely the loss functions:

$$\mathcal{L}_{\text{CE}} = -\tfrac{1}{T} \sum_{t=1}^{T} \log p^{\mathbb{S}}(x_{t+1} \mid x_{\leq t}), \tag{6}$$

$$\mathcal{L}_{\text{KD}} = \tfrac{1}{T} \sum_{t=1}^{T} \operatorname{KL}\big(\operatorname{softmax}(z_t^{\mathbb{T}}/\tau) \,\big\|\, \operatorname{softmax}(z_t^{\mathbb{S}}/\tau)\big), \tag{7}$$

$$\mathcal{L} = \mathcal{L}_{\text{CE}} + \beta\,\tau^2\,\mathcal{L}_{\text{KD}}. \tag{8}$$

Here $p^{\mathbb{S}}(x_{t+1} \mid x_{\leq t})$ is the student next-token probability under the converted MLA layer (with RoPE adapters $W_Q^R, W_K^R$ and MLA factors $W^a, W^b$), while $z_t^{\mathbb{T}}$ and $z_t^{\mathbb{S}}$ are teacher and student logits. We use $p^{\mathbb{S}}(x_{t+1} \mid x_{\leq t}) = \operatorname{softmax}(\tfrac{z_t^{\mathbb{S}}}{\tau})_{x_{t+1}}$ with temperature $\tau$ and weight $\beta$ (Hinton et al., 2015).

## 4 EXPERIMENTAL RESULTS

We evaluate whether **CARE** improves MLA migration under a fixed KV-cache budget (*KV-parity*). We report one-shot and post-healing perplexity & accuracy, ablation study, long-context retrieval (NiH), and system efficiency.

**Setup.** We match cached KV state per token under the same global budget (Tab. 1). CARE estimates $C = \operatorname{Cov}[X]$ on a small calibration set and factors $\sqrt{C}W$ with shrinkage $\sqrt{C} \leftarrow (1-\lambda)\sqrt{C} + \lambda I$; We use LM Harness (Gao et al., 2024) with a matched heal budget; full *100% MLA restoration* (TransMLA/MHA2MLA) is evaluated separately in Sec. 4.3.

---

[2]TransMLA Meng et al. (2025) and X-EcoMLA Li et al. (2025b) both describe how to add such a RoPE channel.

**Original, baselines, CARE variants, and datasets.** **GQA (source).** Unmodified grouped-query attention. We compare against KV-compression baselines: **Palu (Chang et al., 2024).** a low-rank projection baseline for KV-cache reduction under a fixed cache budget; **ASVD (Yuan et al., 2023).** activation-aware SVD applied to $(W_K, W_V)$; and **SVD-LLM V2 (Wang et al., 2024b; 2025b).** truncation-aware SVD-style factorization applied to $(W_K, W_V)$. We also include conversion baselines **TransMLA, MHA2MLA (Meng et al., 2025; Ji et al., 2025).** We report their SVD-style initialization in the one-shot setting, while full 100% MLA restoration is evaluated in Sec. 4.3. **CARE-U (uniform-rank).** Covariance-aware factorization with a uniform rank across layers. **CARE-E (adjusted-rank).** The same covariance-aware factorization, but with our adjusted-rank allocation. We evaluate on LM Harness tasks: Wikitext2 (Wiki) (Merity et al., 2016), ARC Challenge (ARC), ARC Easy (ARE) (Clark et al., 2018), HellaSwag (Zellers et al., 2019), PIQA (Bisk et al., 2020), MMLU (Hendrycks et al., 2020), OpenBookQA (OBQA) (Mihaylov et al., 2018), RACE (RA) (Lai et al., 2017), and WinoGrande (WG) (Sakaguchi et al., 2021). Hyperparameters are in App. H.

## 4.1 ONE-SHOT RESULTS

We report *one-shot*, direct KV-compression performance under KV-parity before partial-RoPE and healing on *Alpaca* Taori et al. (2023) calibration dataset; Tab. 1 summarizes the main results for Llama3.1-8B and Qwen3-4B-Instruct-2507. App. I.1 reports additional one-shot results on other models and calibration datasets.

Table 1: One-shot comparison on Llama3.1-8B and Qwen3-4B-Instruct-2507 against original and baselines on multiple tasks. Calibration samples: 256. Sequence length: 2048. Calibration dataset: alpaca. Higher is better for Accuracy (%) (ACC.) (↑) and lower is better for Perplexity (PPL.) (↓).

| Rank | KV Save | Methods | PPL (↓) | ACC (↑) | | | | | | | | |
|------|---------|---------|---------|------|-----------|------|------|------|------|------|------|------|
| | | | ARC (↓) | ARE | HellaSwag | PIQA | MMLU | OBQA | RA | WG | AVG |
| | | | | | | **Llama-3.1-8B-Instruct** | | | | | | |
| | | **GQA (Original)** | 7.21 (↓) | **50.34** | **80.18** | **60.15** | **79.65** | **48.05** | **34.80** | **40.10** | **72.69** | **58.24** |
| | | Palu(SVD) | 2260.60 (↓) | 25.77 | 27.53 | 26.50 | 52.18 | 24.26 | 27.80 | 20.86 | 50.04 | 31.87 |
| | | MHA2MLA | 284863.91 (↓) | 25.94 | 23.82 | 26.34 | 50.92 | 25.62 | 27.80 | 22.11 | 50.59 | 31.64 |
| 64 | 93.75 | **CARE-U (OURS)** | 983.55 (↓) | 23.89 | 30.51 | 26.82 | 54.73 | 23.18 | 26.00 | 20.96 | 49.01 | 31.89 |
| | | **CARE-E (OURS)** | 983.03 (↓) | 23.63 | 31.31 | 27.49 | 54.62 | 22.98 | 27.40 | 21.15 | 50.36 | 32.37 |
| | | ASVD | 2525.33 (↓) | 23.81 | 26.68 | 26.68 | 52.18 | 22.97 | 27.80 | 20.86 | 50.99 | 31.50 |
| | | SVD-LLM V2 | 967.04 (↓) | 23.63 | 30.72 | 26.75 | 54.90 | 23.22 | 26.40 | 20.86 | 48.93 | 31.93 |
| | | Palu(SVD) | 3046.58 (↓) | 23.89 | 27.02 | 26.62 | 50.38 | 23.14 | 24.80 | 22.49 | 49.25 | 30.95 |
| | | MHA2MLA | 15028.91 (↓) | 25.00 | 24.12 | 26.58 | 52.18 | 23.82 | 29.20 | 22.11 | 51.07 | 31.76 |
| 128 | 87.50 | **CARE-U (OURS)** | 398.91 (↓) | 26.71 | 41.96 | 33.64 | 60.99 | 26.06 | 27.00 | 24.21 | 53.20 | 36.72 |
| | | **CARE-E (OURS)** | 353.74 (↓) | 27.30 | 42.47 | 37.96 | 62.46 | 29.38 | 28.20 | 26.12 | 54.85 | 38.59 |
| | | ASVD | 1675.54 (↓) | 25.00 | 27.99 | 26.61 | 52.39 | 23.04 | 26.60 | 21.82 | 48.46 | 31.49 |
| | | SVD-LLM V2 | 386.23 (↓) | 27.56 | 42.38 | 34.06 | 61.04 | 26.17 | 27.80 | 24.21 | 53.99 | 37.15 |
| | | Palu(SVD) | 537.57 (↓) | 22.61 | 31.10 | 27.93 | 55.22 | 22.82 | 27.00 | 22.68 | 51.70 | 32.63 |
| | | MHA2MLA | 1633.65 (↓) | 27.47 | 25.63 | 26.50 | 52.39 | 23.10 | 28.20 | 22.49 | 50.83 | 32.08 |
| 256 | 75.00 | **CARE-U (OURS)** | 48.35 (↓) | 36.09 | 60.19 | 55.75 | 71.87 | 45.64 | 33.20 | 35.22 | 62.04 | 50.00 |
| | | **CARE-E (OURS)** | 49.43 (↓) | 38.48 | 60.06 | 60.54 | 72.42 | 54.29 | 33.60 | 35.50 | 66.06 | 52.62 |
| | | ASVD | 312.86 (↓) | 21.76 | 32.03 | 29.52 | 55.22 | 23.05 | 25.20 | 24.59 | 52.64 | 33.00 |
| | | SVD-LLM V2 | 47.28 (↓) | 36.26 | 60.69 | 56.39 | 72.20 | 46.57 | 33.40 | 34.83 | 60.69 | 50.13 |
| | | Palu(SVD) | 45.40 (↓) | 28.16 | 45.96 | 43.37 | 64.15 | 24.98 | 30.80 | 23.83 | 53.43 | 39.33 |
| | | MHA2MLA | 220.29 (↓) | 25.94 | 40.95 | 39.27 | 61.37 | 25.54 | 26.60 | 26.79 | 56.59 | 37.88 |
| 512 | 50.00 | **CARE-U (OURS)** | 9.64 (↓) | 52.73 | 76.30 | 73.98 | 78.73 | 62.17 | 40.60 | 41.53 | 72.61 | 62.33 |
| | | **CARE-E (OURS)** | 19.50 (↓) | 42.41 | 64.69 | 68.96 | 75.46 | 63.98 | 37.00 | 38.47 | 71.59 | 57.82 |
| | | ASVD | 12.02 (↓) | 46.33 | 69.11 | 70.75 | 76.17 | 41.80 | 36.60 | 33.97 | 66.85 | 55.20 |
| | | SVD-LLM V2 | 9.63 (↓) | 52.39 | 76.68 | 73.57 | 78.45 | 62.31 | 40.40 | 41.63 | 72.22 | 62.21 |
| | | | | | | **Qwen3-4B-Instruct-2507** | | | | | | |
| | | **GQA (Original)** | 10.04 (↓) | **55.89** | **83.12** | **52.65** | **76.01** | **73.37** | **32.00** | **41.24** | **68.11** | **60.30** |
| | | Palu(SVD) | 56922.11 (↓) | 25.34 | 26.77 | 25.73 | 50.44 | 24.31 | 28.40 | 22.11 | 50.91 | 31.75 |
| | | MHA2MLA | 21850.16 (↓) | 25.68 | 25.46 | 26.03 | 51.96 | 22.92 | 27.00 | 22.01 | 50.20 | 31.41 |
| 64 | 93.75 | **CARE-U (OURS)** | 905.74 (↓) | 23.46 | 29.00 | 27.61 | 55.22 | 23.03 | 26.60 | 23.06 | 50.59 | 32.32 |
| | | **CARE-E (OURS)** | 730.93 (↓) | 23.29 | 30.77 | 28.33 | 55.22 | 22.95 | 24.80 | 22.78 | 51.54 | 32.46 |
| | | ASVD | 6683.95 (↓) | 27.39 | 25.00 | 25.71 | 50.38 | 24.08 | 26.60 | 22.11 | 50.43 | 31.46 |
| | | SVD-LLM V2 | 894.94 (↓) | 23.12 | 28.58 | 27.66 | 55.11 | 23.02 | 27.20 | 23.44 | 51.54 | 32.46 |
| | | Palu(SVD) | 22048.79 (↓) | 26.02 | 26.18 | 26.29 | 51.09 | 24.49 | 25.40 | 21.05 | 49.72 | 31.28 |
| | | MHA2MLA | 52683.47 (↓) | 23.81 | 26.56 | 26.74 | 52.18 | 24.74 | 28.20 | 22.68 | 49.72 | 31.83 |
| 128 | 87.50 | **CARE-U (OURS)** | 111.17 (↓) | 27.47 | 39.69 | 37.52 | 60.50 | 27.03 | 27.60 | 26.99 | 53.59 | 37.55 |
| | | **CARE-E (OURS)** | 102.38 (↓) | 30.29 | 44.82 | 42.54 | 63.76 | 30.52 | 29.40 | 28.71 | 54.54 | 40.57 |

| Rank | KV Save | Methods | PPL (↓) ARC (↓) | ACC (↑) | | | | | | | |
|------|---------|---------|------|-----|-----------|------|------|------|------|------|------|
| | | | | ARE | HellaSwag | PIQA | MMLU | OBQA | RA | WG | AVG |
| | | ASVD | 1682.84 (↓) | 22.95 | 30.01 | 29.29 | 52.34 | 23.42 | 27.00 | 23.54 | 50.12 | 32.33 |
| | | SVD-LLM V2 | 116.76 (↓) | 27.05 | 40.03 | 37.02 | 61.04 | 26.69 | 26.80 | 25.74 | 53.28 | 37.21 |
| 256 | 75.00 | Palu(SVD) | 2561.97 (↓) | 26.11 | 29.46 | 29.92 | 52.88 | 24.48 | 28.40 | 23.64 | 51.70 | 33.32 |
| | | MHA2MLA | 44509.79 (↓) | 22.18 | 28.49 | 28.92 | 52.45 | 23.00 | 24.60 | 24.59 | 51.30 | 31.94 |
| | | **CARE-U (OURS)** | **22.08 (↓)** | **46.42** | **68.90** | **59.16** | **71.55** | **54.76** | **36.40** | **35.02** | **62.43** | **54.33** |
| | | **CARE-E (OURS)** | **28.84 (↓)** | **41.30** | **59.22** | **56.53** | **69.37** | **53.50** | **35.20** | **32.82** | **61.88** | **51.23** |
| | | ASVD | 63.15 (↓) | 32.76 | 46.84 | 47.45 | 63.93 | 26.38 | 30.80 | 30.05 | 52.01 | 41.28 |
| | | SVD-LLM V2 | 22.88 (↓) | 44.54 | 67.17 | 57.77 | 70.78 | 52.81 | 35.60 | 34.83 | 61.48 | 53.12 |
| 512 | 50.00 | Palu(SVD) | 33.97 (↓) | 35.58 | 47.64 | 50.44 | 65.18 | 27.85 | 32.80 | 30.24 | 52.64 | 42.80 |
| | | MHA2MLA | 100.99 (↓) | 27.05 | 41.08 | 37.97 | 59.19 | 29.14 | 27.20 | 29.47 | 54.06 | 38.15 |
| | | **CARE-U (OURS)** | **12.03 (↓)** | **54.95** | **77.23** | **69.24** | **76.22** | **67.46** | **40.00** | **39.43** | **68.43** | **61.62** |
| | | **CARE-E (OURS)** | **15.91 (↓)** | **49.23** | **70.88** | **64.13** | **72.80** | **64.16** | **36.20** | **36.56** | **64.56** | **57.31** |
| | | ASVD | 15.49 (↓) | 47.61 | 66.54 | 67.49 | 73.01 | 56.56 | 35.60 | 35.22 | 62.75 | 55.60 |
| | | SVD-LLM V2 | 11.88 (↓) | 54.61 | 77.44 | 68.53 | 75.68 | 67.65 | 39.80 | 38.18 | 67.25 | 61.14 |

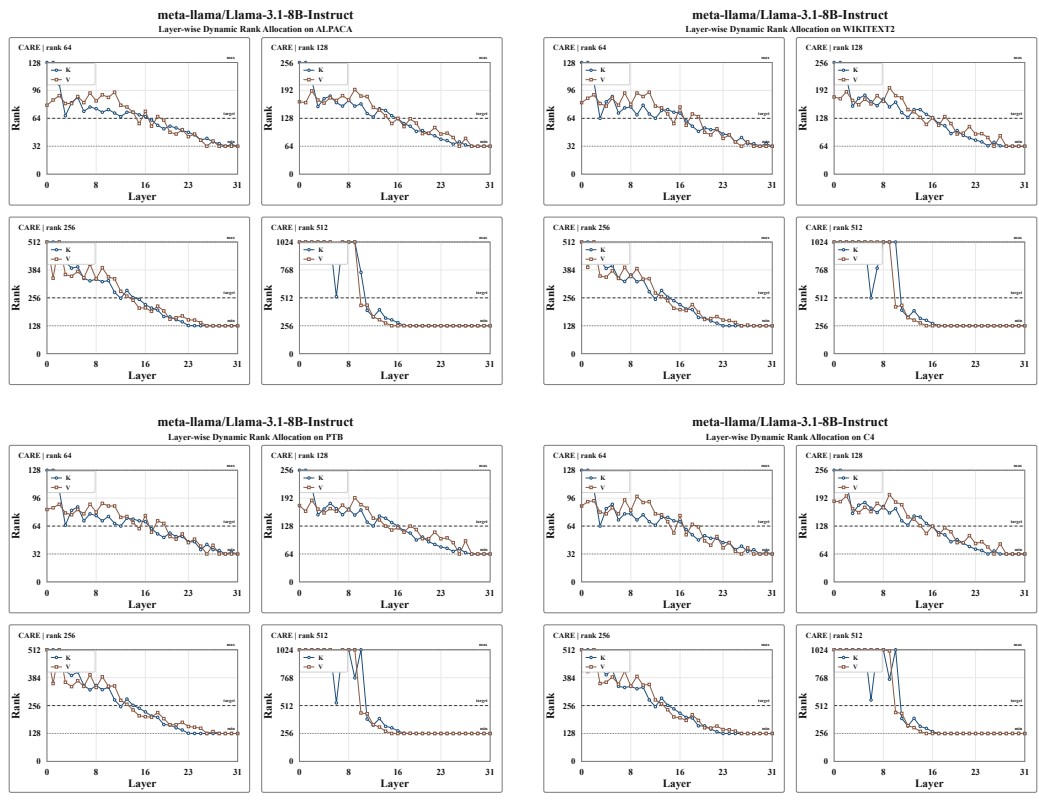

Figure 3: Covariance-aware rank profiles across calibration corpora (Alpaca, WikiText2, PTB, C4) for Llama-3.1-8B-Instruct at target ranks 64, 128, 256, and 512. Across all target budgets, both $W_K$ and $W_V$ show a depth-dependent increase—small in early layers, rising through mid layers—with stronger late-layer growth for $W_V$. The consistency across corpora suggests a model-intrinsic trend.

At very low rank, one-shot MLA remains challenging for all methods, but regarding perplexity, CARE already improves substantially over the naive Palu(SVD) or MHA2MLA initialization. As the rank budget increases to 256/512, the advantage of covariance-aware initialization becomes much clearer, especially on Llama-3.1-8B, Qwen3-4B where CARE-E gives the strongest overall accuracy-PPL trade-off. We provide additional one-shot long-context retrieval results and qualitative examples in App. I.2 and App. I.3, respectively.

## 4.2 ABLATION STUDIES

### 4.2.1 ENERGY DISTRIBUTION VS. COVARIANCE

Using the covariance-aware energy in Sec. 3, we obtain highly consistent rank profiles across C4 (Raffel et al., 2020a), Alpaca (Taori et al., 2023), WikiText2 (Merity et al., 2016), and PTB (Marcus et al., 1993). As shown in Fig. 3, rank is front-loaded: across target ranks 64, 128, 256, and 512, both $W_K$ and $W_V$ receive much larger ranks in the first layers and then decrease steadily with depth, approaching the minimum rank in later layers. This indicates that early layers are more accuracy-critical, while deeper layers can be compressed more aggressively. The same trend also appears on other models (From 1.5B - 70B) (App. I.4), suggesting that the profile is largely *model-intrinsic* rather than tied to a particular calibration corpus.

### 4.2.2 ACCURACY IMPACT OF COVARIANCE

Using Llama-3.1-8B-Instruct, we vary (i) calibration samples, (ii) sequence length, and (iii) calibration corpus. Fig. 4 shows that CARE is already strong with small calibration budgets: performance saturates beyond 512 samples, and longer sequences tend to overfit limited calibration data. We therefore use 256 samples with a short sequence length (32) as the default trade-off. Across corpora (ALPACA, C4, WIKITEXT2, PTB, ARC, ARE, MMLU), one-shot accuracy changes are modest on average (Tab. I.14); domain-aligned corpora give mostly local gains. We default to ALPACA for broad coverage and stable generalization.

Additional ablations are deferred to the appendix for readability. App. I.6 sweeps the shrinkage coefficient, showing that CARE is not sensitive to the exact regularization magnitude and supporting our default $\alpha = 0.01$. App. I.5 studies cross-domain calibration under distribution shift, App. I.7 examines task-weighted covariance mixing, and App. I.8 compares the default $\sqrt{C}$ formulation against using $C$ directly under matched settings. Although the derivation is based on $\sqrt{C}$, the appendix shows that $\sqrt{C}$ is more consistent than using $C$ directly, especially beyond the lowest ranks.

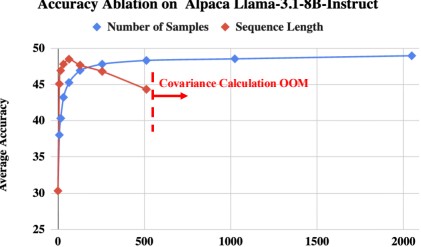

Figure 4: One-shot accuracy versus calibration samples and sequence length across eight benchmarks. Sequence length is fixed at 256 unless otherwise noted; the red curve denotes fixed 256 samples with varying sequence length. OOM occurs beyond length = 512 during covariance computation.

## 4.3 RECOVERY
100% MLA WITH SMALL SFT BUDGETS

**Setup (healing baselines).** We compare CARE against Palu(SVD) and TransMLA under matched healing budgets at MLA rank 512. Our hybrid `TransMLA + CARE(E) Init` keeps the same 100% MLA restoration and healing pipeline as TransMLA, but when TransMLA performs the KV low-rank mapping, we replace its original initialization with the covariance-aware $\sqrt{C}W$ SVD used by CARE(E). All subsequent restoration and healing stages are unchanged. CARE-based 100 use `alpaca-256-256` calibration (256 samples, sequence length 256), TransMLA uses `wiki-256-256` (256 samples, sequence length 256), and Palu(SVD) does not require a separate calibration corpus. We report the one-shot initialization (0B) and healed checkpoints after 1B and 3B tokens on the same LM Harness suite.

Table 2: Healed Llama3.1-8B-Instruct comparison at MLA rank 512 across token budgets.

| Rank | Methods | Calibration | Token | ARC (↑) | ARE (↑) | HellaSwag (↑) | PIQA (↑) | MMLU (↑) | OBQA (↑) | RA (↑) | WG (↑) | AVG (↑) |
|---|---|---|---|---|---|---|---|---|---|---|---|---|
| – | GQA (original) | N/A | N/A | 50.34 | 80.18 | 60.15 | 79.65 | 48.05 | 34.80 | 40.10 | 72.69 | 58.24 |
| 512 | Palu(SVD) | N/A | 0B | 26.02 | 50.97 | 37.43 | 64.15 | 27.89 | 19.40 | 26.60 | 57.54 | 38.75 |
| | Palu(SVD) | N/A | 1B | 33.45 | 62.38 | 45.55 | 72.34 | 50.57 | 23.00 | 31.54 | 61.78 | 47.58 |
| | Palu(SVD) | N/A | 3B | 44.56 | 74.96 | 52.20 | 76.63 | 61.08 | 30.40 | 45.15 | 65.42 | 56.30 |
| | CARE (OURS) ONE-SHOT | alpaca-256-32 | 0B | 52.73 | 76.30 | 73.98 | 78.73 | 62.17 | 40.60 | 41.53 | 72.61 | 62.33 |
| | TransMLA + CARE(E) Init (OURS, 100% MLA Restore) | alpaca-256-256 | 0B | 45.05 | 69.02 | 68.98 | 76.06 | 51.16 | 37.60 | 39.43 | 68.98 | 57.04 |
| | TransMLA + CARE(E) Init (OURS, 100% MLA Restore) | alpaca-256-256 | 1B | 52.25 | 82.33 | 62.47 | 80.21 | 70.31 | 32.90 | 45.11 | 75.13 | 62.59 |
| | TransMLA + CARE(E) Init (OURS, 100% MLA Restore) | alpaca-256-256 | 3B | 51.75 | 80.73 | 64.45 | 83.23 | 71.57 | 34.00 | 46.33 | 74.09 | 63.27 |
| | TransMLA | wiki-256-256 | 0B | 43.17 | 66.12 | 68.25 | 74.81 | 48.89 | 38.80 | 37.70 | 66.46 | 55.52 |
| | TransMLA | wiki-256-256 | 1B | 53.04 | 81.07 | 58.75 | 81.04 | 69.13 | 32.00 | 44.09 | 71.74 | 61.36 |
| | TransMLA | wiki-256-256 | 3B | 53.77 | 82.34 | 56.44 | 80.70 | 70.23 | 33.30 | 45.61 | 72.47 | 61.86 |

We sweep small-scale SFT budgets to quantify post-conversion recovery; results are shown in Tab. 2. CARE(E) already starts from the strongest one-shot initialization, and this advantage largely remains after healing. At 1B and 3B tokens, `TransMLA + CARE(E) Init` reaches 62.59 and 63.27 average accuracy, outperforming TransMLA (61.36/61.86) and Palu(SVD) (47.58/56.30) at matched budgets. While both Palu(SVD) and TransMLA improve substantially with more tokens, CARE attains the best overall recovery with less reliance on long healing runs; optimization objectives, datasets, and hyperparameters in App. H. A system-level KV-cache efficiency analysis is given in App. I.9.

## 5 RELATED WORK

**Conversion from Traditional Attention to MLA.** Standard multi-head attention (MHA) underpins modern LLMs but induces a KV cache that scales linearly with sequence length and head width, creating a memory and bandwidth bottleneck at inference time (Vaswani et al., 2017; Kwon et al., 2023). Grouped-Query Attention (GQA) reduces KV heads by sharing keys/values across query groups, lowering KV memory while sacrificing expressiveness (Ainslie et al., 2023; Shazeer, 2019). Multi-Head Latent Attention (MLA) addresses KV memory by caching low-dimensional latents with lightweight up/down projections (DeepSeek-AI Team, 2024). Beyond MLA that trains from scratch, several *post-hoc* pathways demonstrate conversion feasibility: *TransMLA* gives a theory-based reduction from GQA to MLA at corresponding KV budget (Meng et al., 2025), *MHA2MLA* focuses on practical alignment via partial RoPE and joint-SVD initialization (Ji et al., 2025), and *X-EcoMLA* explores distillation-based upcycling of pretrained attention into MLA for extreme KV compression (Li et al., 2025b). Also, following *MambaInLlama* (Wang et al., 2024a), *Zebra-Llama* composes efficient hybrids to improve inference efficiency and can be paired with MLA-style KV reductions in deployed systems (Yang et al., 2025).

**SVD inspirations.** Naive SVD truncation minimizes $\|W - W_r\|_F$ and ranks by raw singular values, which need not correlate with downstream loss (Eckart & Young, 1936; LeCun et al., 1990; Hassibi & Stork, 1993; Dong et al., 2019; Frantar et al., 2022; Krzanowski, 2000). Recent works refine this in LLMs: *FWSVD* (weighted by Fisher information) (Hua et al., 2022), *SVD-LLM* (truncation-aware whitening + sequential low-rank updates) and its *V2* variant (improved truncation/rank selection) (Wang et al., 2024b; 2025b). *SoCo* learns a diagonal reweighting to optimize the singular spectrum directly for compression rather than trusting singular-value magnitudes (Li et al., 2025a), while *Dobi-SVD* introduces a differentiable SVD that targets activation-side truncation and efficient reconstruction (Wang et al., 2025a). Together with architecture-aware conversions (Meng et al., 2025; Ji et al., 2025), these SVD-oriented techniques motivate *data/curvature-aware* orientations and *non-uniform rank* allocation as core tools for preserving pretrained knowledge. Relatedly, *CorDA* leverages context/activation statistics to guide decomposition adaptation for parameter-efficient fine-tuning, further supporting the value of covariance-aware orientations beyond weight-only SVD (Yang et al., 2024b).

**SVD for cache compression.** Orthogonally, some methods compress the *cache itself*. *Palu* compresses KV-cache with low-rank projection, reconstructing full $K, V$ on the fly with efficient rank search and quantization interoperability (Chang et al., 2024). *ReALLM* combines low-rank components with vector-quantized latents under a unified recipe (Leconte et al., 2024). These methods can be stacked with MLA-style conversions or used standalone to further lower KV memory.

For more related works, please refer to App. F.

## 6 CONCLUSIONS

We proposed **CARE**, a *Covariance-Aware, Rank-Enhanced* procedure for migrating traditional attention to MLA under a fixed KV-parity. CARE replaces naïve weight-only SVD with a covariance-weighted factorization and assigns per-layer ranks via an energy-driven, water-filling schedule. Empirically, CARE preserves MLA's identical KV footprint while delivering lower zero-shot perplexity and higher accuracy before healing, and better final performance than competing baselines under the same post-conversion tuning budget. It is also more robust under aggressive rank reduction, providing a stronger starting point for brief SFT to recover the original model's accuracy.

## 7 ETHICS STATEMENT

We have read and will comply with the ICLR Code of Ethics. Our study involves no human subjects, personally identifiable information, or user-generated content. All datasets are standard, publicly available benchmarks used under their respective licenses; we do not collect or infer demographic attributes. The work focuses on model architecture/optimization and does not introduce capabilities intended for surveillance, profiling, or other harmful use. We identify no foreseeable risks related to privacy, security, fairness, or legal/regulatory compliance, and no IRB/ethics approval was required. To support transparency, we will release code, configuration files, and clear instructions to reproduce all results. All findings are reported honestly without fabrication or inappropriate manipulation. The authors declare no conflicts of interest and no external sponsorship that could bias the work.

## 8 REPRODUCIBILITY STATEMENT

We provide an open-source repository (`https://github.com/FutureMLS-Lab/CARE`). The repo includes exact configuration files for all experiments in Tab. 1–2 and Fig. 3–4, scripts to download and verify datasets, deterministic preprocessing, fixed random seeds, and environment specifications (Conda with pinned versions). Algorithmic details appear in Sec. 3; dataset descriptions and evaluation setup are given in Sec. 4; hyperparameters, hardware, optimizer and scheduler configurations are listed in App. H. Detailed zero-shot tables for all models and calibration corpora are in App. I.1; additional rank profiles (1.5B–70B, including MoE) in App. I.4; ablations on shrinkage (App. I.6), distribution shift (App. I.5), covariance mixing (App. I.7), and $\sqrt{C}$ vs. $C$ (App. I.8); long-context NiH evaluation in App. I.2; and KV-cache efficiency analysis in App. I.9. Running scripts in the repository recreates reported metrics and regenerates all plots and logs. The repository is released under the *Apache License 2.0*.

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

# CARE: Covariance-Aware and Rank-Enhanced Decomposition for Enabling Multi-Head Latent Attention in LLMs

### *Supplementary Material*

## A LARGE LANGUAGE MODELS USAGE

We used a large language model - ChatGPT (GPT-5 thinking) solely for grammar and spelling edits to author-written text. We used Claude Code to assist code writing. The tool did not generate scientific content, design experiments, analyze data, or select citations, and therefore did not contribute at the level of a contributing author. All edits were reviewed and approved by the authors, who take full responsibility for the final manuscript.

## B RECALL: KV-PARITY MAPPING

Grouped-Query Attention (GQA) reduces KV-cache memory by letting multiple heads share the same key–value projection. To convert GQA to Multi-Head Latent Attention (MLA) *without* increasing the KV budget, we enforce **KV parity**. Consider a GQA layer with $n_h$ heads of size $d_h$ with total multi-head hidden size $D = n_h d_h$, we split $n_h$ heads split into $g_h$ groups, where each group contains $\frac{n_h}{g_h}$ heads. The layer $l$ uses $W_Q^{(l)} \in \mathbb{R}^{D \times (n_h d_h)}$ and grouped $W_K^{(l)}, W_V^{(l)} \in \mathbb{R}^{D \times (g_h d_h)}$. We conceptually *replicate* each group's $W_K^{(l)}$ and $W_V^{(l)}$ across its $\frac{n_h}{g_h}$ members to form the full-size $\widetilde{W}_K^{(l)}, \widetilde{W}_V^{(l)} \in \mathbb{R}^{D \times (n_h d_h)}$ (no need to materialize in code). This demonstrates that the GQA method can be reduced to MLA by removing the repeated head blocks (Meng et al., 2025). Therefore, we set MLA's latent rank to match GQA's per-token KV width:

$$r = g_h d_h \tag{9}$$

## C PROOF OF MAXIMUM ENERGY SVD TRUNCATION

We have the following proposition: Let $A \in \mathbb{R}^{m \times n}$ have singular value decomposition (SVD) $A = U\Sigma V^\top$, where $\Sigma = \mathrm{diag}(\sigma_1, \ldots, \sigma_p)$, $p = \min\{m, n\}$, and $\sigma_1 \geq \cdots \geq \sigma_p \geq 0$. For $1 \leq r < p$, let $\Sigma_r = \mathrm{diag}(\sigma_1, \ldots, \sigma_r, 0, \ldots, 0)$ and $A_r := U\Sigma_r V^\top$. Then

$$\|A - A_r\|_F^2 = \sum_{i=r+1}^{p} \sigma_i^2 \quad \text{and} \quad \|A - A_r\|_F = \min_{\mathrm{rank}(X) \leq r} \|A - X\|_F,$$

with the unique minimizers (when $\sigma_r > \sigma_{r+1}$) given by $X = A_r$ (Eckart & Young, 1936).

*Proof.* The Frobenius norm is unitarily invariant, so for any $X$ with $\mathrm{rank}(X) \leq r$,

$$\|A - X\|_F = \|U^\top (A - X) V\|_F = \|\Sigma - Y\|_F, \quad \text{where } Y := U^\top X V \text{ and } \mathrm{rank}(Y) \leq r.$$

Expand the square via the Frobenius inner product $\langle M, N \rangle := \mathrm{trace}(M^\top N)$:

$$\|\Sigma - Y\|_F^2 = \|\Sigma\|_F^2 + \|Y\|_F^2 - 2\langle \Sigma, Y \rangle.$$

Let $s_1(Y) \geq \cdots \geq s_p(Y) \geq 0$ be the singular values of $Y$ (so $s_i(Y) = 0$ for $i > r$). By von Neumann's trace inequality,

$$\langle \Sigma, Y \rangle \leq \sum_{i=1}^{p} \sigma_i s_i(Y) = \sum_{i=1}^{r} \sigma_i s_i(Y),$$

and $\|Y\|_F^2 = \sum_{i=1}^{p} s_i(Y)^2 = \sum_{i=1}^{r} s_i(Y)^2$. Therefore:

$$\|\Sigma - Y\|_F^2 \geq \sum_{i=1}^{p} \sigma_i^2 + \sum_{i=1}^{r} s_i(Y)^2 - 2\sum_{i=1}^{r} \sigma_i s_i(Y) = \sum_{i=1}^{r} (\sigma_i - s_i(Y))^2 + \sum_{i=r+1}^{p} \sigma_i^2 \geq \sum_{i=r+1}^{p} \sigma_i^2.$$

This lower bound is attained by taking $Y = \Sigma_r$, i.e. $X = U\Sigma_r V^\top = A_r$, for which:

$$\|A - A_r\|_F^2 \;=\; \|\Sigma - \Sigma_r\|_F^2 \;=\; \sum_{i=r+1}^{p} \sigma_i^2.$$

Thus $A_r$ is a best rank-$r$ approximation in Frobenius norm, and the minimum value is the squared $\ell_2$-tail of the singular values. Uniqueness follows when $\sigma_r > \sigma_{r+1}$ since then any other minimizer must share the top $r$ singular subspaces with $A$. $\qquad\square$

Note that the idea and proof of the theorem above follows essentially the same idea as Eckart–Young–Mirsky in Eckart & Young (1936).

**Relation to Sec. 3.2.** The theorem justifies the use of squared singular values in Sec. 3.2: for a single layer/matrix, the marginal reduction in Frobenius residual from adding the next rank is governed by the next squared singular value. Sec. 3.2 further divides this local gain by the remaining residual energy of the same layer, which normalizes layers with different spectral scales and makes their priorities comparable under a shared global budget. This normalization changes the cross-layer comparison, but preserves the theorem's local interpretation that larger squared singular values remove more residual energy.

## D  DERIVATION OF SVD TARGET FUNCTION

Let $\Delta W := W - \widehat{W}$ and $C = \frac{1}{N}\sum_{b=1}^{N} X_b^\top X_b$. We claim that:

$$\frac{1}{N}\sum_{b=1}^{N} \|X_b \Delta W\|_F^2 = \|\sqrt{C}\,\Delta W\|_F^2.$$

*Proof.* By the Frobenius–trace identity $\|A\|_F^2 = \mathrm{Tr}(A^\top A)$,

$$\frac{1}{N}\sum_{b=1}^{N} \|X_b \Delta W\|_F^2 = \frac{1}{N}\sum_{b=1}^{N} \mathrm{Tr}\big((X_b\Delta W)^\top (X_b \Delta W)\big) = \frac{1}{N}\sum_{b=1}^{N} \mathrm{Tr}\big(\Delta W^\top X_b^\top X_b \Delta W\big).$$

Using linearity of trace and the definition of $C$,

$$\frac{1}{N}\sum_{b=1}^{N} \|X_b \Delta W\|_F^2 = \mathrm{Tr}\big(\Delta W^\top C \Delta W\big).$$

We can show that $C \succeq 0$ since for any vector $v$, we have:

$$v^\top C v = \frac{1}{N}\sum_{b=1}^{N} v^\top X_b^\top X_b v = \frac{1}{N}\sum_{b=1}^{N} \|X_b v\|_2^2 \geq 0$$

Then we write $C = C^{1/2}C^{1/2}$ to get:

$$\mathrm{Tr}\big(\Delta W^\top C \Delta W\big) = \mathrm{Tr}\big(\Delta W^\top C^{1/2}C^{1/2}\Delta W\big) = \mathrm{Tr}\big((C^{1/2}\Delta W)^\top (C^{1/2}\Delta W)\big) = \|C^{1/2}\Delta W\|_F^2.$$
$$\square$$

## E  DEFINITION OF COVARIANCE

Let $x \in \mathbb{R}^D$ be the activation (feature) vector of a single token. The population covariance is:

$$\mathrm{Cov}[x] \;=\; \mathbb{E}\big[(x-\mu)(x-\mu)^\top\big], \qquad \mu \;=\; \mathbb{E}[x].$$

Given samples $\{x_i\}_{i=1}^N$, the empirical mean and covariance are:

$$\hat{\mu} \;=\; \frac{1}{N}\sum_{i=1}^{N} x_i, \qquad \widehat{\mathrm{Cov}}[x] \;=\; \frac{1}{N}\sum_{i=1}^{N} (x_i - \hat{\mu})(x_i - \hat{\mu})^\top.$$

If rows of $X \in \mathbb{R}^{N\times D}$ stack the samples and $X_c = X - \mathbf{1}\hat{\mu}^\top$, then $\widehat{\mathrm{Cov}}[x] = \frac{1}{N}X_c^\top X_c$. (*Unbiased* version uses $1/(N-1)$ instead of $1/N$.)

# F  RELATED WORK

**KV Management**  Serving throughput is often bounded by how the KV cache is organized and moved across memory. *PagedAttention* (vLLM) treats KV as pageable blocks to avoid internal/external fragmentation and enable sharing across sequences, improving utilization under dynamic batching (Kwon et al., 2023). Orthogonally, *FlashAttention* reduces HBM traffic with an IO-aware tiling of exact attention, and *FlashAttention-2* further improves parallelism and work partitioning for higher FLOPs utilization (Dao et al., 2022; Dao, 2023). These system/kernel directions are complementary to architectural changes (e.g., GQA/MLA) and to post-hoc reparameterizations, since better KV layout and IO scheduling directly translate into larger effective batch sizes at a fixed memory budget.

**Quantization**  Quantization provides an orthogonal compression path to low-rank methods and can be combined with MLA/GQA conversions. For *weights/activations*, *SmoothQuant* migrates activation outliers into weights to enable practical W8A8 PTQ on large models (Xiao et al., 2023). *AWQ* protects a small set of salient channels via activation-aware scaling, delivering strong 4-bit weight-only PTQ with hardware-friendly kernels (Lin et al., 2024). *QuIP#* pushes extreme regimes ($\leq$4-bit) using randomized Hadamard incoherence and lattice codebooks, with state-of-the-art results at low bit-rates (Tseng et al., 2024). For the *KV cache*, *KVQuant* (NeurIPS'24) introduces pre-RoPE key quantization, sensitivity-aware non-uniform datatypes, and per-vector dense/sparse schemes to sustain long-context inference (Hooper et al., 2024), while *KIVI* shows tuning-free 2-bit asymmetric KV quantization with favorable throughput/memory trade-offs (Liu et al., 2024b). Together, these methods form a toolbox that is largely complementary to low-rank latent caching.

**RoPE and Positional Encodings**  Positional design strongly affects length generalization and conversion stability. RoPE's complex-valued rotary formulation remains the default in many LLMs (Su et al., 2021b). Alternatives include relative positions (Shaw et al., 2018), T5's learned relative bias and DeBERTa's disentangled content/position attention (Raffel et al., 2020b; He et al., 2021), and ALiBi's linear distance bias for train-short/test-long extrapolation (Press et al., 2021). Within the RoPE family, window-extension strategies modify scaling or spectra to stabilize extrapolation, such as XPOS's multiplicative stabilization, Position Interpolation, YaRN, and very long-window *LongRoPE* (Sun et al., 2022; Chen et al., 2023; Peng et al., 2023; Ding et al., 2024). Systematic comparisons further show that the chosen positional scheme materially impacts length generalization (Kazemnejad et al., 2023), motivating careful treatment (e.g., partial-RoPE or mixed strategies) during architectural realignments.

# G  DISCUSSION

Across two GQA backbones and diverse tasks, **CARE**—Covariance-Aware and Rank-Enhanced decomposition—enables MLA migration under *KV-parity* with accuracy and long-context robustness on par with (or better than) stronger baselines. CARE preserves the throughput/memory advantages of MLA while mitigating the activation drift observed with weight-only SVD.

**Rank allocation matters.**  Uniform or purely energy-based rank policies overlook the *weighted* spectral concentration that emerges after covariance/curvature preconditioning. CARE's *Adjusted-Rank* uses a water-filling allocation over weighted singular spectra, honoring a global KV constraint while allocating capacity to the layers and directions that matter most.

**Complexity**  The dominant conversion costs are per-layer covariance estimation $\mathcal{O}(ND^2)$ (with small $N$) and truncated SVD on $\sqrt{C}\widetilde{W} \in \mathbb{R}^{D \times (n_h d_h)}$ at $\mathcal{O}(D\,(n_h d_h)\,r)$ using randomized SVD. Layers can be processed sequentially; for each layer, $C = \frac{1}{N}\sum_{b=1}^{N} X_b^\top X_b$ can all be kept on CPU. At inference, MLA incurs light extra matvecs by $W^b$ while reducing KV-cache width from $n_h d_h$ (MHA) or $g_h d_h$ (GQA) down to $r = g_h d_h$ (MLA). CARE is orthogonal to quantization and sparsification and compatible with MLA kernels (DeepSeek-AI Team, 2024).

**Compatibility with MLA migration.**  CARE complements recent MLA conversions (Ji et al., 2025; Meng et al., 2025) and plays well with partial-RoPE (Su et al., 2021a): removing rota-

tions on least-contributive subspaces further stabilizes long-context behavior when combined with activation/curvature-aware objectives.

**Limitations.** (i) *Statistics freshness*: CARE requires small calibration passes; pronounced domain shift may need refreshed covariance/curvature. (ii) *Diagonal curvature*: practicality favors diagonal proxies; structured approximations (e.g., Kronecker-factored) may yield further gains. (iii) *Extreme compression*: at very low ranks, information bottlenecks dominate and further SFT can be necessary. (iv) *Orthogonality to quantization/eviction*: CARE does not yet co-optimize KV quantization and cache eviction policies. (v) *Kernel support*: no dedicated MLA kernel currently supports per-layer dynamic ranks, making end-to-end latency benchmarking impractical; we therefore report only theoretical KV-cache savings rather than wall-clock speedups.

**Broader impact and future work.** CARE suggests a general recipe for post-training architectural migrations: align the objective to where errors manifest and distribute capacity by curvature-weighted signal. Promising directions include applying covariance-aware low-rank decomposition to vision-language and multimodal architectures (Fang et al., 2026a;b), where attention modules face analogous KV-cache bottlenecks, as well as data-free calibration, structured curvature, and dynamic rank schedules that adapt latent capacity with context length while maintaining KV-parity.

Apart from that, our Covariance-weighted SVD initialization minimizes the activation loss at each layer, but our true goal is to preserve the output of the model, which is next-token predictions. We may therefore cast low-rank compression as directly minimizing the sequence loss produced by the compressed (student) model under a fixed KV budget.

## H   HYPER-PARAMETER SELECTION

All experiments were conducted on servers equipped with NVIDIA H100 80 GB GPUs paired with dual Intel Xeon Platinum 8462Y+ processors ($2 \times 32$-core sockets, 64 cores total) and approximately 2 TB of RAM.

All hyper-parameters are shown as below:

- **Model Configuration**: Base model: Meta; Precision: float16, Sequence length: 8-2048 tokens, Covariance samples: 8-2048.

- **MLA Rank Settings**: Default rank: 256, Min rank: 64, Max rank: 1024, Uniform allocation: True/False, K/V projection ranks: 256 each.

- **CARE Parameters**: Initialization method: CARE, Damping factor (percdamp): 0.01, Cholesky decomposition: False, Activation order: False.

- **Evaluation Datasets**: Multi-task benchmarks including WikiText (perplexity), ARC-Challenge/Easy (reasoning), HellaSwag (commonsense), PIQA (physical reasoning), MMLU (knowledge), OpenBookQA, RACE (reading), WinoGrande (coreference).

- **Generation Settings**: Max new tokens: 512-128000, Temperature: 0.6-0.7, Top-p sampling: 0.9, Sampling strategy: Nucleus sampling with temperature control.

- **System Configuration**: GPU memory free threshold (minimal GPU resources to run experiments) : 2048 MB, Parallel GPUs: 1-8 devices, Batch size: Dynamic adjustment, Random seed: [42, 17, 26, 103, 21, 59, 134, 8, 24, 99].

- **Covaraince Computation**: Dataset: C4/Ptb/Wikitext/Alpaca/ARC/ARE/MMLU instruction-following, Sample size: 8-2048, Sequence processing: 8-2048 token windows,

- **Random Seed and Learning Rate** All experimental results are average over **10** random seeds and we choose the best from **3** learning rates.

- **Training Framework** All experiments were conducted using the `VeOmni` framework (Ma et al., 2025) for fine-tuning CARE and TransMLA models.

- **Learning rate:** We choose best learning rate of $2 \times 10^{-6}$ with linear warmup over the first 0.001% training steps.

- **Optimizer and Schedulers:** We sweep LR $\in \{1\times10^{-6}, 1\times10^{-5}, 5\times10^{-5}, 1\times10^{-4}, 5\times 10^{-4}\}$, and select the best within the first 0.1B tokens. We use AdamW, weight_decay= 0.01, cosine decay with `lr_warmup_ratio`= 0.005 and `lr_decay_ratio`= 1.0.
- **Batch size:** Global effective batch size of 64 tokens per update step, accumulated across devices.
- **Precision:** bfloat16 mixed precision was enabled to reduce memory footprint and improve throughput.
- **Max sequence length:** Input sequences were truncated or padded to a length of 512-128000 tokens.
- **Training epochs:** Each experiment was trained for the number of pre-set tokens.

# I SUPPLEMENTARY RESULTS

## I.1 DETAILED ZERO-SHOT TABLES

These tables report the full zero-shot results for each model and calibration dataset combination, using the same metric layout as Tab. 1. In addition to the main-text variants (**CARE-U** and **CARE-E**, which use $\sqrt{C}$-weighted SVD with uniform and energy-aware rank allocation, respectively), we include **CARE-C-based-U** and **CARE-C-based-E**, which replace $\sqrt{C}$ with $C$ directly as the covariance weighting; see Sec. I.8 for a detailed comparison of the two formulations.

### I.1.1 LLAMA-3.1-8B-INSTRUCT-ALPACA

Table I.1: Detailed zero-shot comparison for Llama-3.1-8B-Instruct with Alpaca calibration. Higher is better for Accuracy (%) (ACC.) (↑) and lower is better for Perplexity (PPL.) (↓).

| Rank | Methods | PPL (↓) | ACC (↑) | | | | | | | | |
|---|---|---|---|---|---|---|---|---|---|---|---|
| | | | ARC-E | ARC-C | HellaSwag | PIQA | WG | MMLU | OBQA | RA | AVG |
| 64 | Palu(SVD) | 2260.60(↓) | 27.53 | 25.77 | 26.50 | 52.18 | 50.04 | 24.26 | 27.80 | 20.86 | 31.87 |
| | MHA2MLA | 284863.91(↓) | 23.82 | 25.94 | 26.34 | 50.92 | 50.59 | 25.62 | 27.80 | 22.11 | 31.64 |
| | **CARE-C-based-U** | 893.84(↓) | 31.40 | 23.98 | 26.91 | 55.66 | 50.04 | 22.91 | 27.00 | 21.15 | 32.38 |
| | **CARE-C-based-E** | 837.25(↓) | 32.41 | 23.46 | 27.50 | 56.75 | 49.96 | 22.90 | 27.40 | 21.34 | 32.71 |
| | **CARE-U** | 983.55(↓) | 30.51 | 23.89 | 26.82 | 54.73 | 49.01 | 23.18 | 26.00 | 20.96 | 31.89 |
| | **CARE-E** | 983.03(↓) | 31.31 | 23.63 | 27.49 | 54.62 | 50.36 | 22.98 | 27.40 | 21.15 | 32.37 |
| | ASVD | 2525.33(↓) | 26.68 | 23.81 | 26.68 | 52.18 | 50.99 | 22.97 | 27.80 | 20.86 | 31.50 |
| | SVD-LLM V2 | 967.04(↓) | 30.72 | 23.63 | 26.75 | 54.90 | 48.93 | 23.22 | 26.40 | 20.86 | 31.93 |
| 128 | Palu(SVD) | 3046.58(↓) | 27.02 | 23.89 | 26.62 | 50.38 | 49.25 | 23.14 | 24.80 | 22.49 | 30.95 |
| | MHA2MLA | 15028.91(↓) | 24.12 | 25.00 | 26.58 | 52.18 | 51.07 | 23.82 | 29.20 | 22.11 | 31.76 |
| | **CARE-C-based-U** | 355.47(↓) | 42.80 | 27.05 | 33.64 | 61.59 | 53.51 | 26.79 | 27.40 | 24.11 | 37.11 |
| | **CARE-C-based-E** | 328.76(↓) | 43.60 | 27.90 | 36.60 | 62.46 | 54.78 | 28.88 | 29.00 | 25.26 | 38.56 |
| | **CARE-U** | 398.91(↓) | 41.96 | 26.71 | 33.64 | 60.99 | 53.20 | 26.06 | 27.00 | 24.21 | 36.72 |
| | **CARE-E** | **353.74(↓)** | **42.47** | **27.30** | **37.96** | **62.46** | **54.85** | **29.38** | **28.20** | **26.12** | **38.59** |
| | ASVD | 1675.54(↓) | 27.99 | 25.00 | 26.61 | 52.39 | 48.46 | 23.04 | 26.60 | 21.82 | 31.49 |
| | SVD-LLM V2 | 386.23(↓) | 42.38 | 27.56 | 34.06 | 61.04 | 53.99 | 26.17 | 27.80 | 24.21 | 37.15 |
| 256 | Palu(SVD) | 537.57(↓) | 31.10 | 22.61 | 27.93 | 55.22 | 51.70 | 22.82 | 27.00 | 22.68 | 32.63 |
| | MHA2MLA | 1633.65(↓) | 25.63 | 27.47 | 26.50 | 52.39 | 50.83 | 23.10 | 28.20 | 22.49 | 32.08 |
| | **CARE-C-based-U** | 49.35(↓) | 59.18 | 33.62 | 53.43 | 71.44 | 61.48 | 44.06 | 33.60 | 33.49 | 48.79 |
| | **CARE-C-based-E** | 49.40(↓) | 61.74 | 37.54 | 58.28 | 72.42 | 63.22 | 50.43 | 33.00 | 34.64 | 51.41 |
| | **CARE-U** | 48.35(↓) | 60.19 | 36.09 | 55.75 | 71.87 | 62.04 | 45.64 | 33.20 | 35.22 | 50.00 |
| | **CARE-E** | **49.43(↓)** | **60.06** | **38.48** | **60.54** | **72.42** | **66.06** | **54.29** | **33.60** | **35.50** | **52.62** |
| | ASVD | 312.86(↓) | 32.03 | 21.76 | 29.52 | 55.22 | 52.64 | 23.05 | 25.20 | 24.59 | 33.00 |
| | SVD-LLM V2 | 47.28(↓) | 60.69 | 36.26 | 56.39 | 72.20 | 60.69 | 46.57 | 33.40 | 34.83 | 50.13 |
| 512 | Palu(SVD) | 45.40(↓) | 45.96 | 28.16 | 43.37 | 64.15 | 53.43 | 24.98 | 30.80 | 23.83 | 39.33 |
| | MHA2MLA | 220.29(↓) | 40.95 | 25.94 | 39.27 | 61.37 | 56.59 | 25.54 | 26.60 | 26.79 | 37.88 |
| | **CARE-C-based-U** | 10.06(↓) | 76.68 | 51.45 | 72.20 | 77.91 | 71.35 | 61.03 | 41.20 | 41.72 | 61.69 |
| | **CARE-C-based-E** | 16.46(↓) | 68.18 | 44.03 | 69.88 | 76.06 | 71.90 | 63.28 | 38.60 | 40.10 | 59.00 |
| | **CARE-U** | **9.64(↓)** | **76.30** | **52.73** | **73.98** | **78.73** | **72.61** | **62.17** | **40.60** | **41.53** | **62.33** |
| | **CARE-E** | 19.50(↓) | 64.69 | 42.41 | 68.96 | 75.46 | 71.59 | 63.98 | 37.00 | 38.47 | 57.82 |
| | ASVD | 12.02(↓) | 69.11 | 46.33 | 70.75 | 76.17 | 66.85 | 41.80 | 36.60 | 33.97 | 55.20 |
| | SVD-LLM V2 | 9.63(↓) | 76.68 | 52.39 | 73.57 | 78.45 | 72.22 | 62.31 | 40.40 | 41.63 | 62.21 |

## I.1.2 LLAMA-3.1-8B-INSTRUCT-C4

Table I.2: Detailed zero-shot comparison for Llama-3.1-8B-Instruct with C4 calibration. Higher is better for Accuracy (%) (ACC.) (↑) and lower is better for Perplexity (PPL.) (↓).

| Rank | Methods | PPL (↓) | ACC (↑) | | | | | | | | |
|---|---|---|---|---|---|---|---|---|---|---|---|
| | | | ARC-E | ARC-C | HellaSwag | PIQA | WG | MMLU | OBQA | RA | AVG |
| 64 | Palu(SVD) | 2260.60 (↓) | 27.53 | 25.77 | 26.50 | 52.18 | 50.04 | 24.26 | 27.80 | 20.86 | 31.87 |
| | MHA2MLA | 284863.91 (↓) | 23.82 | 25.94 | 26.34 | 50.92 | 50.59 | 25.62 | 27.80 | 22.11 | 31.64 |
| | **CARE-C-based-U** | 688.95 (↓) | 28.32 | 24.74 | 27.28 | 55.93 | 51.85 | 22.95 | 26.40 | 21.05 | 32.31 |
| | **CARE-C-based-E** | 582.93 (↓) | 29.55 | 24.15 | 28.15 | 56.75 | 52.25 | 22.95 | 26.20 | 20.96 | 32.62 |
| | **CARE-U** | 786.17 (↓) | 27.61 | 24.40 | 26.98 | 54.68 | 49.88 | 22.95 | 26.40 | 20.19 | 31.64 |
| | **CARE-E** | **676.43** (↓) | 28.24 | 24.74 | 27.97 | 55.39 | 50.12 | 22.95 | 26.60 | 20.86 | 32.11 |
| | ASVD | 2845.27 (↓) | 25.97 | 23.98 | 27.01 | 51.90 | 48.86 | 22.98 | 29.20 | 22.30 | 31.52 |
| | SVD-LLM V2 | 769.05 (↓) | 27.74 | 24.57 | 27.10 | 54.62 | 49.96 | 22.95 | 26.00 | 20.38 | 31.66 |
| 128 | Palu(SVD) | 3046.58 (↓) | 27.02 | 23.89 | 26.62 | 50.38 | 49.25 | 23.14 | 24.80 | 22.49 | 30.95 |
| | MHA2MLA | 15028.91 (↓) | 24.12 | 25.00 | 26.58 | 52.18 | 51.07 | 23.82 | 29.20 | 22.11 | 31.76 |
| | **CARE-C-based-U** | 182.54 (↓) | 35.23 | 24.23 | 36.20 | 61.92 | 55.64 | 23.04 | 28.00 | 25.74 | 36.25 |
| | **CARE-C-based-E** | 176.68 (↓) | 37.88 | 26.88 | 40.07 | 64.09 | 55.49 | 25.02 | 29.80 | 26.51 | 38.22 |
| | **CARE-U** | 220.53 (↓) | 34.39 | 24.23 | 36.25 | 61.21 | 54.78 | 23.46 | 27.60 | 25.17 | 35.89 |
| | **CARE-E** | **201.19** (↓) | 33.92 | 26.37 | 40.96 | 62.08 | 56.75 | 25.33 | 28.00 | 26.89 | 37.54 |
| | ASVD | 1767.53 (↓) | 28.24 | 25.00 | 26.95 | 53.21 | 49.17 | 23.01 | 27.40 | 20.86 | 31.73 |
| | SVD-LLM V2 | 211.16 (↓) | 34.51 | 24.40 | 36.77 | 60.99 | 54.78 | 23.44 | 27.80 | 25.65 | 36.04 |
| 256 | Palu(SVD) | 537.57 (↓) | 31.10 | 22.61 | 27.93 | 55.22 | 51.70 | 22.82 | 27.00 | 22.68 | 32.63 |
| | MHA2MLA | 1633.65 (↓) | 25.63 | 27.47 | 26.50 | 52.39 | 50.83 | 23.10 | 28.20 | 22.49 | 32.08 |
| | **CARE-C-based-U** | 34.32 (↓) | 53.70 | 33.62 | 58.51 | 71.76 | 61.96 | 37.69 | 31.80 | 35.12 | 48.02 |
| | **CARE-C-based-E** | 37.79 (↓) | 55.30 | 35.49 | 61.54 | 72.42 | 64.96 | 42.85 | 32.20 | 35.12 | 49.99 |
| | **CARE-U** | 35.63 (↓) | 53.07 | 34.73 | 60.63 | 72.14 | 63.06 | 40.09 | 32.60 | 34.93 | 48.91 |
| | **CARE-E** | 38.89 (↓) | 53.41 | 35.75 | 62.79 | 71.71 | 64.88 | 49.54 | 32.00 | 34.26 | 50.54 |
| | ASVD | 313.33 (↓) | 31.57 | 22.53 | 29.77 | 55.55 | 53.28 | 23.21 | 26.00 | 23.35 | 33.16 |
| | SVD-LLM V2 | 35.49 (↓) | 53.07 | 34.81 | 60.87 | 72.42 | 63.54 | 40.97 | 32.80 | 34.64 | 49.14 |
| 512 | Palu(SVD) | 45.40 (↓) | 45.96 | 28.16 | 43.37 | 64.15 | 53.43 | 24.98 | 30.80 | 23.83 | 39.33 |
| | MHA2MLA | 220.29 (↓) | 40.95 | 25.94 | 39.27 | 61.37 | 56.59 | 25.54 | 26.60 | 26.79 | 37.88 |
| | **CARE-C-based-U** | 9.58 (↓) | 76.47 | 50.77 | 73.14 | 77.58 | 71.98 | 59.61 | 39.20 | 42.39 | 61.39 |
| | **CARE-C-based-E** | 15.55 (↓) | 66.12 | 43.09 | 70.58 | 76.28 | 71.27 | 62.32 | 37.00 | 39.90 | 58.32 |
| | **CARE-U** | **9.31** (↓) | 76.47 | 50.77 | 75.02 | 77.80 | 71.82 | 59.96 | 41.00 | 42.87 | 61.96 |
| | **CARE-E** | 18.07 (↓) | 60.94 | 41.21 | 70.29 | 75.73 | 71.59 | 63.10 | 37.00 | 38.56 | 57.30 |
| | ASVD | 12.01 (↓) | 68.64 | 45.65 | 70.50 | 76.55 | 67.88 | 39.40 | 37.40 | 33.88 | 54.99 |
| | SVD-LLM V2 | 9.33 (↓) | 76.01 | 51.11 | 74.80 | 77.80 | 71.51 | 59.93 | 40.60 | 42.01 | 61.72 |

## I.1.3 LLAMA-3.1-8B-INSTRUCT-PTB

Table I.3: Detailed zero-shot comparison for Llama-3.1-8B-Instruct with PTB calibration. Higher is better for Accuracy (%) (ACC.) (↑) and lower is better for Perplexity (PPL.) (↓).

| Rank | Methods | PPL (↓) | ACC (↑) | | | | | | | | |
|---|---|---|---|---|---|---|---|---|---|---|---|
| | | | ARC-E | ARC-C | HellaSwag | PIQA | WG | MMLU | OBQA | RA | AVG |
| 64 | Palu(SVD) | 2260.60 (↓) | 27.53 | 25.77 | 26.50 | 52.18 | 50.04 | 24.26 | 27.80 | 20.86 | 31.87 |
| | MHA2MLA | 284863.91 (↓) | 23.82 | 25.94 | 26.34 | 50.92 | 50.59 | 25.62 | 27.80 | 22.11 | 31.64 |
| | **CARE-C-based-U** | 756.56 (↓) | 26.73 | 25.68 | 26.28 | 51.58 | 50.20 | 22.95 | 25.80 | 21.34 | 31.32 |
| | **CARE-C-based-E** | 612.72 (↓) | 27.57 | 25.43 | 26.51 | 51.74 | 50.75 | 22.95 | 26.00 | 21.05 | 31.50 |
| | **CARE-U** | 815.80 (↓) | 26.26 | 25.00 | 26.50 | 51.36 | 49.25 | 22.95 | 27.00 | 20.67 | 31.12 |
| | **CARE-E** | **689.55** (↓) | 27.15 | 25.60 | 26.51 | 51.25 | 49.49 | 22.94 | 27.20 | 20.38 | 31.31 |
| | ASVD | 2718.72 (↓) | 26.39 | 25.09 | 26.91 | 51.69 | 49.01 | 23.02 | 29.00 | 21.72 | 31.60 |
| | SVD-LLM V2 | 803.38 (↓) | 26.39 | 25.43 | 26.41 | 51.41 | 49.41 | 22.95 | 27.20 | 20.67 | 31.23 |
| 128 | Palu(SVD) | 3046.58 (↓) | 27.02 | 23.89 | 26.62 | 50.38 | 49.25 | 23.14 | 24.80 | 22.49 | 30.95 |
| | MHA2MLA | 15028.91 (↓) | 24.12 | 25.00 | 26.58 | 52.18 | 51.07 | 23.82 | 29.20 | 22.11 | 31.76 |
| | **CARE-C-based-U** | 200.05 (↓) | 32.11 | 23.81 | 30.01 | 54.90 | 53.67 | 23.30 | 26.40 | 23.54 | 33.47 |
| | **CARE-C-based-E** | 178.91 (↓) | 33.00 | 24.74 | 31.94 | 55.88 | 54.78 | 23.62 | 28.60 | 24.40 | 34.62 |
| | **CARE-U** | 229.71 (↓) | 31.19 | 24.74 | 29.80 | 53.16 | 52.80 | 24.80 | 25.80 | 23.73 | 33.25 |
| | **CARE-E** | **194.88** (↓) | 30.51 | 24.40 | 31.96 | 54.24 | 56.04 | 22.97 | 28.20 | 22.97 | 33.91 |
| | ASVD | 1776.16 (↓) | 27.36 | 24.57 | 26.73 | 53.16 | 49.41 | 22.97 | 27.60 | 22.20 | 31.75 |
| | SVD-LLM V2 | 220.92 (↓) | 31.44 | 23.98 | 29.94 | 54.08 | 53.28 | 25.07 | 25.40 | 23.92 | 33.39 |
| 256 | Palu(SVD) | 537.57 (↓) | 31.10 | 22.61 | 27.93 | 55.22 | 51.70 | 22.82 | 27.00 | 22.68 | 32.63 |

| Rank | Methods | PPL (↓) | ACC (↑) | | | | | | | | |
|---|---|---|---|---|---|---|---|---|---|---|---|
| | | | ARC-E | ARC-C | HellaSwag | PIQA | WG | MMLU | OBQA | RA | AVG |
| | MHA2MLA | 1633.65 (↓) | 25.63 | 27.47 | 26.50 | 52.39 | 50.83 | 23.10 | 28.20 | 22.49 | 32.08 |
| | **CARE-C-based-U** | 33.00 (↓) | 49.66 | 30.55 | 48.89 | 64.58 | 62.51 | 37.50 | 29.40 | 31.77 | 44.36 |
| | **CARE-C-based-E** | 30.87 (↓) | 50.55 | 33.28 | 52.55 | 67.08 | 62.59 | 42.75 | 31.20 | 32.54 | 46.57 |
| | **CARE-U** | 34.44 (↓) | 43.56 | 27.73 | 48.90 | 61.75 | 62.35 | 31.65 | 30.40 | 29.09 | 41.93 |
| | **CARE-E** | **33.96 (↓)** | **47.52** | **34.13** | **54.84** | **66.43** | **63.77** | **45.54** | **31.60** | **32.63** | **47.06** |
| | ASVD | 316.66 (↓) | 31.27 | 22.18 | 29.34 | 55.33 | 53.12 | 23.14 | 25.40 | 23.92 | 32.96 |
| | SVD-LLM V2 | 33.36 (↓) | 47.69 | 29.52 | 50.43 | 64.69 | 61.88 | 35.32 | 30.80 | 30.43 | 43.85 |
| 512 | Palu(SVD) | 45.40 (↓) | 45.96 | 28.16 | 43.37 | 64.15 | 53.43 | 24.98 | 30.80 | 23.83 | 39.33 |
| | MHA2MLA | 220.29 (↓) | 40.95 | 25.94 | 39.27 | 61.37 | 56.59 | 25.54 | 26.60 | 26.79 | 37.88 |
| | **CARE-C-based-U** | 9.67 (↓) | 73.48 | 49.06 | 70.88 | 76.22 | 71.19 | 57.52 | 37.40 | 40.67 | 59.55 |
| | **CARE-C-based-E** | 13.00 (↓) | 62.96 | 40.78 | 68.24 | 74.10 | 69.93 | 61.96 | 35.40 | 39.71 | 56.64 |
| | **CARE-U** | **9.38 (↓)** | **75.25** | **51.71** | **72.93** | **77.26** | **72.06** | **57.86** | **40.20** | **40.77** | **61.00** |
| | **CARE-E** | 16.21 (↓) | 56.31 | 38.99 | 66.99 | 73.56 | 70.56 | 62.63 | 34.40 | 37.89 | 55.17 |
| | ASVD | 12.15 (↓) | 67.51 | 43.94 | 70.00 | 75.63 | 66.93 | 38.43 | 36.20 | 32.06 | 53.84 |
| | SVD-LLM V2 | 9.42 (↓) | 74.71 | 49.91 | 72.18 | 77.09 | 71.90 | 57.83 | 37.80 | 40.29 | 60.21 |

### I.1.4 LLAMA-3.1-8B-INSTRUCT-WIKITEXT2

Table I.4: Detailed zero-shot comparison for Llama-3.1-8B-Instruct with WikiText2 calibration. Higher is better for Accuracy (%) (ACC.) (↑) and lower is better for Perplexity (PPL.) (↓).

| Rank | Methods | PPL (↓) | ACC (↑) | | | | | | | | |
|---|---|---|---|---|---|---|---|---|---|---|---|
| | | | ARC-E | ARC-C | HellaSwag | PIQA | WG | MMLU | OBQA | RA | AVG |
| 64 | Palu(SVD) | 2260.60 (↓) | 27.53 | 25.77 | 26.50 | 52.18 | 50.04 | 24.26 | 27.80 | 20.86 | 31.87 |
| | MHA2MLA | 284863.91 (↓) | 23.82 | 25.94 | 26.34 | 50.92 | 50.59 | 25.62 | 27.80 | 22.11 | 31.64 |
| | **CARE-C-based-U** | 189.39 (↓) | 28.11 | 25.09 | 26.82 | 51.47 | 50.12 | 22.95 | 27.80 | 20.57 | 31.62 |
| | **CARE-C-based-E** | 138.00 (↓) | 28.91 | 25.43 | 27.16 | 52.39 | 50.36 | 22.95 | 27.60 | 20.38 | 31.90 |
| | **CARE-U** | 237.45 (↓) | 27.31 | 25.34 | 26.46 | 51.63 | 48.22 | 22.94 | 27.60 | 20.77 | 31.28 |
| | **CARE-E** | **184.54 (↓)** | **27.86** | **24.91** | **26.83** | **51.96** | **50.28** | **22.95** | **26.80** | **20.57** | **31.52** |
| | ASVD | 2900.25 (↓) | 26.09 | 24.06 | 26.77 | 51.25 | 49.25 | 22.96 | 29.00 | 21.91 | 31.41 |
| | SVD-LLM V2 | 229.74 (↓) | 27.15 | 25.26 | 26.50 | 51.69 | 48.62 | 22.94 | 27.60 | 20.86 | 31.33 |
| 128 | Palu(SVD) | 3046.58 (↓) | 27.02 | 23.89 | 26.62 | 50.38 | 49.25 | 23.14 | 24.80 | 22.49 | 30.95 |
| | MHA2MLA | 15028.91 (↓) | 24.12 | 25.00 | 26.58 | 52.18 | 51.07 | 23.82 | 29.20 | 22.11 | 31.76 |
| | **CARE-C-based-U** | 51.39 (↓) | 33.08 | 23.98 | 31.51 | 55.50 | 52.41 | 23.43 | 28.40 | 23.64 | 33.99 |
| | **CARE-C-based-E** | 47.26 (↓) | 33.25 | 25.60 | 33.59 | 57.13 | 54.14 | 23.37 | 29.80 | 24.98 | 35.23 |
| | **CARE-U** | 66.44 (↓) | 32.37 | 23.98 | 31.04 | 54.08 | 52.33 | 24.10 | 28.60 | 23.73 | 33.78 |
| | **CARE-E** | **55.08 (↓)** | **32.41** | **24.66** | **34.07** | **55.11** | **54.30** | **23.10** | **29.40** | **25.36** | **34.80** |
| | ASVD | 1823.04 (↓) | 27.69 | 25.51 | 26.96 | 53.59 | 50.99 | 22.92 | 29.40 | 21.24 | 32.29 |
| | SVD-LLM V2 | 62.94 (↓) | 32.11 | 23.89 | 31.38 | 54.68 | 53.43 | 24.04 | 28.60 | 23.44 | 33.95 |
| 256 | Palu(SVD) | 537.57 (↓) | 31.10 | 22.61 | 27.93 | 55.22 | 51.70 | 22.82 | 27.00 | 22.68 | 32.63 |
| | MHA2MLA | 1633.65 (↓) | 25.63 | 27.47 | 26.50 | 52.39 | 50.83 | 23.10 | 28.20 | 22.49 | 32.08 |
| | **CARE-C-based-U** | 15.66 (↓) | 47.10 | 29.10 | 49.85 | 63.87 | 60.85 | 33.65 | 32.60 | 31.67 | 43.59 |
| | **CARE-C-based-E** | 15.53 (↓) | 53.32 | 34.64 | 55.90 | 68.01 | 63.54 | 41.05 | 33.80 | 33.68 | 47.99 |
| | **CARE-U** | 16.14 (↓) | 45.88 | 29.69 | 52.27 | 63.17 | 62.35 | 33.26 | 33.80 | 32.63 | 44.13 |
| | **CARE-E** | **16.84 (↓)** | **51.77** | **34.30** | **58.43** | **67.41** | **65.59** | **45.89** | **32.00** | **33.59** | **48.62** |
| | ASVD | 318.48 (↓) | 31.82 | 22.27 | 29.37 | 54.95 | 54.06 | 23.00 | 26.40 | 23.35 | 33.15 |
| | SVD-LLM V2 | 15.88 (↓) | 48.19 | 30.03 | 52.77 | 64.80 | 61.48 | 34.33 | 34.00 | 32.25 | 44.73 |
| 512 | Palu(SVD) | 45.40 (↓) | 45.96 | 28.16 | 43.37 | 64.15 | 53.43 | 24.98 | 30.80 | 23.83 | 39.33 |
| | MHA2MLA | 220.29 (↓) | 40.95 | 25.94 | 39.27 | 61.37 | 56.59 | 25.54 | 26.60 | 26.79 | 37.88 |
| | **CARE-C-based-U** | 8.29 (↓) | 76.43 | 50.26 | 72.00 | 76.66 | 72.22 | 56.83 | 38.60 | 41.63 | 60.58 |
| | **CARE-C-based-E** | 9.82 (↓) | 66.96 | 42.83 | 69.64 | 74.76 | 70.64 | 61.99 | 37.20 | 38.85 | 57.86 |
| | **CARE-U** | 8.29 (↓) | **76.60** | **51.54** | **73.94** | **77.48** | **72.53** | **56.99** | **39.40** | **40.19** | **61.08** |
| | **CARE-E** | 11.25 (↓) | 60.90 | 41.04 | 69.03 | 73.72 | 71.27 | 63.12 | 35.40 | 37.80 | 56.54 |
| | ASVD | 12.11 (↓) | 67.63 | 44.28 | 70.07 | 75.68 | 67.01 | 38.78 | 36.40 | 33.30 | 54.14 |
| | SVD-LLM V2 | 8.26 (↓) | 76.30 | 51.28 | 73.54 | 77.64 | 72.30 | 58.07 | 39.60 | 39.81 | 61.07 |

### I.1.5 QWEN2.5-1.5B-INSTRUCT-ALPACA

Table I.5: Detailed zero-shot comparison for Qwen2.5-1.5B-Instruct with Alpaca calibration. Higher is better for Accuracy (%) (ACC.) (↑) and lower is better for Perplexity (PPL.) (↓).

| Rank | Methods | PPL (↓) | ACC (↑) | | | | | | | | |
|---|---|---|---|---|---|---|---|---|---|---|---|
| | | | ARC-E | ARC-C | HellaSwag | PIQA | WG | MMLU | OBQA | RA | AVG |
| 64 | Palu(SVD) | 114579.38(↓) | 25.17 | 25.26 | 25.55 | 50.27 | 50.20 | 24.06 | 27.40 | 20.67 | 31.07 |
| | MHA2MLA | 62077.36(↓) | 26.56 | 25.34 | 26.57 | 51.63 | 50.20 | 24.60 | 29.00 | 22.68 | 32.07 |
| | **CARE-C-based-U** | 909.76(↓) | 29.00 | 22.35 | 26.14 | 52.50 | 48.15 | 23.11 | 26.60 | 22.30 | 31.27 |
| | **CARE-C-based-E** | 730.60(↓) | 29.00 | 22.87 | 26.83 | 54.19 | 51.30 | 22.85 | 27.60 | 24.11 | 32.34 |
| | **CARE-U** | 1098.80(↓) | 26.77 | 23.38 | 26.95 | 52.29 | 51.07 | 22.95 | 25.20 | 23.16 | 31.47 |
| | **CARE-E** | **1022.88(↓)** | **29.00** | **23.29** | **26.70** | **54.73** | **51.14** | **23.00** | **26.80** | **22.11** | **32.10** |
| | ASVD | 7737.23(↓) | 26.81 | 25.77 | 26.61 | 51.09 | 49.88 | 25.46 | 27.60 | 22.11 | 31.92 |
| | SVD-LLM V2 | 1109.62(↓) | 28.37 | 23.63 | 26.91 | 53.48 | 49.80 | 22.95 | 23.80 | 23.92 | 31.61 |
| 96 | Palu(SVD) | 147110.47(↓) | 25.97 | 23.98 | 26.49 | 50.87 | 50.43 | 24.33 | 30.20 | 20.57 | 31.60 |
| | MHA2MLA | 73056.19(↓) | 26.14 | 26.71 | 26.51 | 52.12 | 50.28 | 22.90 | 26.80 | 21.82 | 31.66 |
| | **CARE-C-based-U** | 237.47(↓) | 35.02 | 24.32 | 29.87 | 57.51 | 52.57 | 24.48 | 26.60 | 26.22 | 34.57 |
| | **CARE-C-based-E** | 51.56(↓) | 52.15 | 30.46 | 46.17 | 67.68 | 54.70 | 32.03 | 31.60 | 29.47 | 43.03 |
| | **CARE-U** | 266.39(↓) | 33.96 | 22.35 | 29.80 | 56.26 | 50.91 | 24.13 | 28.20 | 25.07 | 33.83 |
| | **CARE-E** | **48.58(↓)** | **49.37** | **29.01** | **46.54** | **66.70** | **53.28** | **31.33** | **30.00** | **31.29** | **42.19** |
| | ASVD | 3663.48(↓) | 28.37 | 23.46 | 27.78 | 52.61 | 51.46 | 24.39 | 27.60 | 25.07 | 32.59 |
| | SVD-LLM V2 | 249.45(↓) | 34.89 | 23.04 | 30.33 | 58.49 | 50.51 | 24.81 | 28.40 | 26.70 | 34.65 |
| 128 | Palu(SVD) | 40284.33(↓) | 28.45 | 21.59 | 27.10 | 52.29 | 51.14 | 25.49 | 30.60 | 22.68 | 32.42 |
| | MHA2MLA | 51022.27(↓) | 26.81 | 25.68 | 27.39 | 51.03 | 49.09 | 26.00 | 27.00 | 23.73 | 32.09 |
| | **CARE-C-based-U** | 47.23(↓) | 55.39 | 31.66 | 44.08 | 66.97 | 55.41 | 35.75 | 29.40 | 33.68 | 44.04 |
| | **CARE-C-based-E** | 30.43(↓) | 56.61 | 32.34 | 51.13 | 69.53 | 53.43 | 40.91 | 34.00 | 31.39 | 46.17 |
| | **CARE-U** | 46.20(↓) | 54.17 | 31.40 | 45.62 | 67.95 | 54.85 | 36.38 | 29.40 | 34.93 | 44.34 |
| | **CARE-E** | **28.00(↓)** | **56.23** | **31.74** | **50.85** | **69.53** | **54.70** | **37.85** | **34.20** | **31.77** | **45.86** |
| | ASVD | 1582.91(↓) | 30.64 | 23.63 | 28.20 | 51.20 | 51.62 | 25.47 | 27.40 | 24.11 | 32.78 |
| | SVD-LLM V2 | 44.32(↓) | 55.81 | 31.48 | 46.29 | 67.36 | 54.38 | 36.65 | 29.00 | 35.02 | 44.50 |

## I.1.6 QWEN2.5-1.5B-INSTRUCT-C4

Table I.6: Detailed zero-shot comparison for Qwen2.5-1.5B-Instruct with C4 calibration. Higher is better for Accuracy (%) (ACC.) (↑) and lower is better for Perplexity (PPL.) (↓).

| Rank | Methods | PPL (↓) | ACC (↑) | | | | | | | | |
|---|---|---|---|---|---|---|---|---|---|---|---|
| | | | ARC-E | ARC-C | HellaSwag | PIQA | WG | MMLU | OBQA | RA | AVG |
| 64 | Palu(SVD) | 114579.38(↓) | 25.17 | 25.26 | 25.55 | 50.27 | 50.20 | 24.06 | 27.40 | 20.67 | 31.07 |
| | MHA2MLA | 62077.36(↓) | 26.56 | 25.34 | 26.57 | 51.63 | 50.20 | 24.60 | 29.00 | 22.68 | 32.07 |
| | **CARE-C-based-U** | 679.30(↓) | 27.74 | 23.12 | 26.69 | 52.77 | 51.22 | 22.95 | 26.80 | 23.16 | 31.81 |
| | **CARE-C-based-E** | 365.98(↓) | 28.54 | 22.70 | 27.42 | 55.01 | 52.09 | 22.95 | 26.40 | 23.64 | 32.34 |
| | **CARE-U** | 914.36(↓) | 26.43 | 24.57 | 26.17 | 51.90 | 49.41 | 22.95 | 25.80 | 23.83 | 31.38 |
| | **CARE-E** | **664.13(↓)** | **27.15** | **23.12** | **27.44** | **53.43** | **51.38** | **22.94** | **27.00** | **23.64** | **32.01** |
| | ASVD | 7074.97(↓) | 26.85 | 25.43 | 27.14 | 50.98 | 50.91 | 25.89 | 30.00 | 22.49 | 32.46 |
| | SVD-LLM V2 | 826.83(↓) | 26.22 | 25.26 | 26.41 | 51.90 | 49.64 | 22.95 | 25.40 | 24.11 | 31.49 |
| 96 | Palu(SVD) | 147110.47(↓) | 25.97 | 23.98 | 26.49 | 50.87 | 50.43 | 24.33 | 30.20 | 20.57 | 31.60 |
| | MHA2MLA | 73056.19(↓) | 26.14 | 26.71 | 26.51 | 52.12 | 50.28 | 22.90 | 26.80 | 21.82 | 31.66 |
| | **CARE-C-based-U** | 151.08(↓) | 32.15 | 22.61 | 31.53 | 55.28 | 51.38 | 23.52 | 27.00 | 27.08 | 33.82 |
| | **CARE-C-based-E** | 43.55(↓) | 43.10 | 27.05 | 47.85 | 68.06 | 55.64 | 27.45 | 31.00 | 31.48 | 41.45 |
| | **CARE-U** | 176.46(↓) | 31.14 | 20.82 | 30.39 | 55.55 | 52.49 | 22.96 | 25.80 | 25.26 | 33.05 |
| | **CARE-E** | **41.81(↓)** | **41.71** | **27.99** | **47.82** | **66.65** | **55.64** | **27.61** | **29.80** | **28.33** | **40.69** |
| | ASVD | 3864.43(↓) | 28.75 | 23.12 | 27.49 | 51.41 | 51.30 | 24.75 | 27.60 | 23.16 | 32.20 |
| | SVD-LLM V2 | 161.86(↓) | 31.23 | 21.50 | 31.15 | 56.31 | 51.22 | 22.94 | 26.60 | 26.51 | 33.43 |
| 128 | Palu(SVD) | 40284.33(↓) | 28.45 | 21.59 | 27.10 | 52.29 | 51.14 | 25.49 | 30.60 | 22.68 | 32.42 |
| | MHA2MLA | 51022.27(↓) | 26.81 | 25.68 | 27.39 | 51.03 | 49.09 | 26.00 | 27.00 | 23.73 | 32.09 |
| | **CARE-C-based-U** | 34.92(↓) | 47.98 | 27.56 | 48.03 | 65.51 | 56.51 | 32.67 | 28.80 | 34.16 | 42.65 |
| | **CARE-C-based-E** | 28.60(↓) | 49.16 | 29.61 | 51.80 | 69.26 | 54.62 | 34.01 | 33.40 | 31.29 | 44.14 |
| | **CARE-U** | 34.14(↓) | 51.68 | 29.10 | 49.03 | 66.70 | 56.91 | 32.47 | 28.20 | 33.97 | 43.51 |
| | **CARE-E** | **24.35(↓)** | **49.28** | **29.35** | **52.03** | **69.31** | **56.59** | **33.39** | **32.00** | **31.20** | **44.14** |
| | ASVD | 1244.07(↓) | 30.77 | 20.99 | 27.47 | 51.85 | 50.36 | 24.63 | 25.80 | 24.78 | 32.08 |
| | SVD-LLM V2 | 32.25(↓) | 51.98 | 28.58 | 49.92 | 67.36 | 58.25 | 34.43 | 29.20 | 33.78 | 44.19 |

### I.1.7 QWEN2.5-1.5B-INSTRUCT-PTB

Table I.7: Detailed zero-shot comparison for Qwen2.5-1.5B-Instruct with PTB calibration. Higher is better for Accuracy (%) (ACC.) (↑) and lower is better for Perplexity (PPL.) (↓).

| Rank | Methods | PPL (↓) | ACC (↑) | | | | | | | | |
|---|---|---|---|---|---|---|---|---|---|---|---|
| | | | ARC-E | ARC-C | HellaSwag | PIQA | WG | MMLU | OBQA | RA | AVG |
| 64 | Palu(SVD) | 114579.38(↓) | 25.17 | 25.26 | 25.55 | 50.27 | 50.20 | 24.06 | 27.40 | 20.67 | 31.07 |
| | MHA2MLA | 62077.36(↓) | 26.56 | 25.34 | 26.57 | 51.63 | 50.20 | 24.60 | 29.00 | 22.68 | 32.07 |
| | **CARE-C-based-U** | 1608.42(↓) | 27.48 | 23.46 | 26.79 | 51.09 | 50.59 | 23.01 | 26.80 | 21.63 | 31.36 |
| | **CARE-C-based-E** | 701.84(↓) | 27.95 | 23.55 | 27.12 | 51.52 | 50.04 | 22.95 | 24.80 | 23.73 | 31.46 |
| | **CARE-U** | 2131.72(↓) | 27.53 | 23.38 | 27.00 | 51.74 | 49.01 | 23.14 | 27.20 | 23.06 | 31.51 |
| | **CARE-E** | 362.35(↓) | 29.59 | 23.55 | 29.43 | 52.45 | 50.83 | 23.10 | 25.00 | 22.39 | 32.04 |
| | ASVD | 8309.98(↓) | 26.98 | 24.83 | 26.90 | 50.82 | 50.67 | 25.40 | 28.40 | 21.91 | 31.99 |
| | SVD-LLM V2 | 2075.31(↓) | 27.36 | 22.61 | 27.18 | 50.82 | 49.01 | 23.13 | 27.60 | 22.68 | 31.30 |
| 96 | Palu(SVD) | 147110.47(↓) | 25.97 | 23.98 | 26.49 | 50.87 | 50.43 | 24.33 | 30.20 | 20.57 | 31.60 |
| | MHA2MLA | 73056.19(↓) | 26.14 | 26.71 | 26.51 | 52.12 | 50.28 | 22.90 | 26.80 | 21.82 | 31.66 |
| | **CARE-C-based-U** | 306.18(↓) | 30.26 | 22.87 | 28.96 | 52.50 | 52.72 | 23.04 | 26.00 | 25.17 | 32.69 |
| | **CARE-C-based-E** | 57.23(↓) | 41.12 | 25.94 | 43.82 | 62.95 | 54.46 | 27.52 | 29.20 | 28.33 | 39.17 |
| | **CARE-U** | 367.50(↓) | 28.79 | 20.99 | 29.39 | 52.29 | 52.72 | 23.39 | 26.00 | 23.92 | 32.19 |
| | **CARE-E** | 74.42(↓) | 41.04 | 26.37 | 43.62 | 62.02 | 53.35 | 25.74 | 30.80 | 29.47 | 39.05 |
| | ASVD | 3704.61(↓) | 28.91 | 23.38 | 28.03 | 52.67 | 50.36 | 24.93 | 28.80 | 24.50 | 32.70 |
| | SVD-LLM V2 | 334.57(↓) | 29.55 | 20.14 | 29.52 | 52.50 | 52.57 | 23.02 | 25.20 | 24.69 | 32.15 |
| 128 | Palu(SVD) | 40284.33(↓) | 28.45 | 21.59 | 27.10 | 52.29 | 51.14 | 25.49 | 30.60 | 22.68 | 32.42 |
| | MHA2MLA | 51022.27(↓) | 26.81 | 25.68 | 27.39 | 51.03 | 49.09 | 26.00 | 27.00 | 23.73 | 32.09 |
| | **CARE-C-based-U** | 65.89(↓) | 44.99 | 25.60 | 41.95 | 60.07 | 55.64 | 32.38 | 27.00 | 31.20 | 39.85 |
| | **CARE-C-based-E** | 23.22(↓) | 49.58 | 30.72 | 49.79 | 66.49 | 54.93 | 35.48 | 31.00 | 32.06 | 43.76 |
| | **CARE-U** | 67.96(↓) | 45.03 | 26.96 | 43.72 | 61.10 | 56.12 | 29.55 | 28.80 | 33.49 | 40.60 |
| | **CARE-E** | 22.47(↓) | 48.74 | 29.18 | 49.28 | 66.87 | 55.96 | 36.80 | 31.60 | 31.39 | 43.73 |
| | ASVD | 1210.45(↓) | 30.60 | 20.73 | 27.75 | 52.23 | 50.59 | 23.86 | 27.20 | 24.69 | 32.21 |
| | SVD-LLM V2 | 65.47(↓) | 45.88 | 28.58 | 44.00 | 60.77 | 55.64 | 29.19 | 29.00 | 32.92 | 40.75 |

### I.1.8 QWEN2.5-1.5B-INSTRUCT-WIKITEXT2

Table I.8: Detailed zero-shot comparison for Qwen2.5-1.5B-Instruct with WikiText2 calibration. Higher is better for Accuracy (%) (ACC.) (↑) and lower is better for Perplexity (PPL.) (↓).

| Rank | Methods | PPL (↓) | ACC (↑) | | | | | | | | |
|---|---|---|---|---|---|---|---|---|---|---|---|
| | | | ARC-E | ARC-C | HellaSwag | PIQA | WG | MMLU | OBQA | RA | AVG |
| 64 | Palu(SVD) | 114579.38(↓) | 25.17 | 25.26 | 25.55 | 50.27 | 50.20 | 24.06 | 27.40 | 20.67 | 31.07 |
| | MHA2MLA | 62077.36(↓) | 26.56 | 25.34 | 26.57 | 51.63 | 50.20 | 24.60 | 29.00 | 22.68 | 32.07 |
| | **CARE-C-based-U** | 209.05(↓) | 27.86 | 22.18 | 26.34 | 52.56 | 50.67 | 22.98 | 28.20 | 23.64 | 31.80 |
| | **CARE-C-based-E** | 122.65(↓) | 27.86 | 23.29 | 27.12 | 53.48 | 50.75 | 22.94 | 26.20 | 22.68 | 31.79 |
| | **CARE-U** | 300.99(↓) | 26.77 | 23.12 | 26.74 | 52.18 | 50.20 | 22.95 | 26.60 | 24.50 | 31.63 |
| | **CARE-E** | 125.54(↓) | 27.31 | 23.12 | 27.86 | 52.45 | 50.99 | 22.96 | 25.40 | 23.54 | 31.70 |
| | ASVD | 7126.52(↓) | 26.81 | 25.60 | 27.19 | 51.90 | 50.20 | 25.71 | 26.80 | 21.82 | 32.00 |
| | SVD-LLM V2 | 264.86(↓) | 26.94 | 23.21 | 26.87 | 51.96 | 49.96 | 22.97 | 26.40 | 24.59 | 31.61 |
| 96 | Palu(SVD) | 147110.47(↓) | 25.97 | 23.98 | 26.49 | 50.87 | 50.43 | 24.33 | 30.20 | 20.57 | 31.60 |
| | MHA2MLA | 73056.19(↓) | 26.14 | 26.71 | 26.51 | 52.12 | 50.28 | 22.90 | 26.80 | 21.82 | 31.66 |
| | **CARE-C-based-U** | 57.36(↓) | 30.81 | 22.10 | 29.87 | 53.48 | 49.72 | 22.97 | 26.20 | 24.50 | 32.46 |
| | **CARE-C-based-E** | 19.53(↓) | 44.40 | 26.88 | 46.33 | 64.74 | 54.78 | 27.99 | 30.20 | 30.05 | 40.67 |
| | **CARE-U** | 61.60(↓) | 31.69 | 21.33 | 29.95 | 54.24 | 50.99 | 23.13 | 26.20 | 23.54 | 32.63 |
| | **CARE-E** | 20.46(↓) | 40.78 | 27.05 | 46.46 | 63.44 | 54.78 | 27.21 | 31.80 | 29.76 | 40.16 |
| | ASVD | 3636.42(↓) | 28.32 | 22.95 | 27.29 | 52.50 | 50.04 | 24.07 | 30.00 | 23.25 | 32.30 |
| | SVD-LLM V2 | 56.77(↓) | 31.06 | 21.67 | 30.41 | 55.28 | 48.86 | 22.93 | 25.60 | 24.31 | 32.51 |
| 128 | Palu(SVD) | 40284.33(↓) | 28.45 | 21.59 | 27.10 | 52.29 | 51.14 | 25.49 | 30.60 | 22.68 | 32.42 |
| | MHA2MLA | 51022.27(↓) | 26.81 | 25.68 | 27.39 | 51.03 | 49.09 | 26.00 | 27.00 | 23.73 | 32.09 |
| | **CARE-C-based-U** | 19.29(↓) | 47.10 | 25.85 | 44.14 | 62.08 | 55.49 | 30.69 | 29.20 | 32.06 | 40.83 |
| | **CARE-C-based-E** | 15.62(↓) | 51.22 | 30.72 | 51.52 | 68.28 | 54.62 | 30.22 | 33.40 | 31.39 | 43.92 |
| | **CARE-U** | 17.88(↓) | 46.68 | 27.30 | 46.46 | 63.76 | 55.96 | 27.84 | 28.60 | 33.11 | 41.21 |
| | **CARE-E** | 15.67(↓) | 51.64 | 30.20 | 50.50 | 67.08 | 55.33 | 29.83 | 33.20 | 32.44 | 43.78 |
| | ASVD | 1241.39(↓) | 30.56 | 21.08 | 27.65 | 53.81 | 49.01 | 23.61 | 26.20 | 23.92 | 31.98 |
| | SVD-LLM V2 | 17.50(↓) | 47.94 | 26.96 | 47.05 | 64.47 | 57.06 | 28.51 | 29.20 | 32.54 | 41.72 |

### I.1.9 QWEN3-4B-INSTRUCT-2507-ALPACA

Table I.9: Detailed zero-shot comparison for Qwen3-4B-Instruct-2507 with Alpaca calibration. Higher is better for Accuracy (%) (ACC.) (↑) and lower is better for Perplexity (PPL.) (↓).

| Rank | Methods | PPL (↓) | ARC-E | ARC-C | HellaSwag | PIQA | WG | MMLU | OBQA | RA | AVG |
|---|---|---|---|---|---|---|---|---|---|---|---|
| | | | | | | ACC (↑) | | | | | |
| 64 | Palu(SVD) | 56922.11(↓) | 26.77 | 25.34 | 25.73 | 50.44 | 50.91 | 24.31 | 28.40 | 22.11 | 31.75 |
| | MHA2MLA | 21850.16(↓) | 25.46 | 25.68 | 26.03 | 51.96 | 50.20 | 22.92 | 27.00 | 22.01 | 31.41 |
| | **CARE-C-based-U** | 1052.43(↓) | 29.34 | 22.70 | 27.51 | 54.35 | 50.67 | 23.21 | 26.20 | 23.64 | 32.20 |
| | **CARE-C-based-E** | 941.16(↓) | 28.75 | 23.12 | 27.33 | 55.33 | 50.28 | 23.08 | 25.60 | 22.30 | 31.97 |
| | **CARE-U** | 905.74(↓) | 29.00 | 23.46 | 27.61 | 55.22 | 50.59 | 23.03 | 26.60 | 23.06 | 32.32 |
| | **CARE-E** | 730.93(↓) | 30.77 | 23.29 | 28.33 | 55.22 | 51.54 | 22.95 | 24.80 | 22.78 | 32.46 |
| | ASVD | 6683.95(↓) | 25.00 | 27.39 | 25.71 | 50.38 | 50.43 | 24.08 | 26.60 | 22.11 | 31.46 |
| | SVD-LLM V2 | 894.94(↓) | 28.58 | 23.12 | 27.66 | 55.11 | 51.54 | 23.02 | 27.20 | 23.44 | 32.46 |
| 128 | Palu(SVD) | 22048.79(↓) | 26.18 | 26.02 | 26.29 | 51.09 | 49.72 | 24.49 | 25.40 | 21.05 | 31.28 |
| | MHA2MLA | 52683.47(↓) | 26.56 | 23.81 | 26.74 | 52.18 | 49.72 | 24.74 | 28.20 | 22.68 | 31.83 |
| | **CARE-C-based-U** | 148.73(↓) | 38.93 | 25.09 | 32.98 | 59.09 | 53.12 | 23.76 | 25.60 | 26.12 | 35.59 |
| | **CARE-C-based-E** | 139.32(↓) | 39.98 | 27.05 | 37.16 | 61.64 | 53.20 | 27.19 | 27.00 | 26.99 | 37.53 |
| | **CARE-U** | 111.17(↓) | 39.69 | 27.47 | 37.52 | 60.50 | 53.59 | 27.03 | 27.60 | 26.99 | 37.55 |
| | **CARE-E** | 102.38(↓) | 44.82 | 30.29 | 42.54 | 63.76 | 54.54 | 30.52 | 29.40 | 28.71 | 40.57 |
| | ASVD | 1682.84(↓) | 30.01 | 22.95 | 29.29 | 52.34 | 50.12 | 23.42 | 27.00 | 23.54 | 32.33 |
| | SVD-LLM V2 | 116.76(↓) | 40.03 | 27.05 | 37.02 | 61.04 | 53.28 | 26.69 | 26.80 | 25.74 | 37.21 |
| 256 | Palu(SVD) | 2561.97(↓) | 29.46 | 26.11 | 29.92 | 52.88 | 51.70 | 24.48 | 28.40 | 23.64 | 33.32 |
| | MHA2MLA | 44509.79(↓) | 28.49 | 22.18 | 28.92 | 52.45 | 51.30 | 23.00 | 24.60 | 24.59 | 31.94 |
| | **CARE-C-based-U** | 24.63(↓) | 64.06 | 38.40 | 52.01 | 69.04 | 59.12 | 45.73 | 34.20 | 34.35 | 49.61 |
| | **CARE-C-based-E** | 28.67(↓) | 60.27 | 38.31 | 53.58 | 69.15 | 59.43 | 49.32 | 33.60 | 33.49 | 49.64 |
| | **CARE-U** | 22.08(↓) | 68.90 | 46.42 | 59.16 | 71.55 | 62.43 | 54.76 | 36.40 | 35.02 | 54.33 |
| | **CARE-E** | 28.84(↓) | 59.22 | 41.30 | 56.53 | 69.37 | 61.88 | 53.50 | 35.20 | 32.82 | 51.23 |
| | ASVD | 63.15(↓) | 46.84 | 32.76 | 47.45 | 63.93 | 52.01 | 26.38 | 30.80 | 30.05 | 41.28 |
| | SVD-LLM V2 | 22.88(↓) | 67.17 | 44.54 | 57.77 | 70.78 | 61.48 | 52.81 | 35.60 | 34.83 | 53.12 |
| 512 | Palu(SVD) | 33.97(↓) | 47.64 | 35.58 | 50.44 | 65.18 | 52.64 | 27.85 | 32.80 | 30.24 | 42.80 |
| | MHA2MLA | 100.99(↓) | 41.08 | 27.05 | 37.97 | 59.19 | 54.06 | 29.14 | 27.20 | 29.47 | 38.15 |
| | **CARE-C-based-U** | 11.90(↓) | 79.04 | 55.20 | 64.32 | 74.21 | 67.01 | 66.35 | 38.60 | 38.76 | 60.44 |
| | **CARE-C-based-E** | 15.52(↓) | 71.04 | 47.18 | 59.86 | 71.93 | 63.30 | 62.19 | 35.40 | 36.84 | 55.97 |
| | **CARE-U** | 12.03(↓) | 77.23 | 54.95 | 69.24 | 76.22 | 68.43 | 67.46 | 40.00 | 39.43 | 61.62 |
| | **CARE-E** | 15.91(↓) | 70.88 | 49.23 | 64.13 | 72.80 | 64.56 | 64.16 | 36.20 | 36.56 | 57.31 |
| | ASVD | 15.49(↓) | 66.54 | 47.61 | 67.49 | 73.01 | 62.75 | 56.56 | 35.60 | 35.22 | 55.60 |
| | SVD-LLM V2 | 11.88(↓) | 77.44 | 54.61 | 68.53 | 75.68 | 67.25 | 67.65 | 39.80 | 38.18 | 61.14 |

### I.1.10 QWEN3-4B-INSTRUCT-2507-C4

Table I.10: Detailed zero-shot comparison for Qwen3-4B-Instruct-2507 with C4 calibration. Higher is better for Accuracy (%) (ACC.) (↑) and lower is better for Perplexity (PPL.) (↓).

| Rank | Methods | PPL (↓) | ARC-E | ARC-C | HellaSwag | PIQA | WG | MMLU | OBQA | RA | AVG |
|---|---|---|---|---|---|---|---|---|---|---|---|
| | | | | | | ACC (↑) | | | | | |
| 64 | Palu(SVD) | 56922.11(↓) | 26.77 | 25.34 | 25.73 | 50.44 | 50.91 | 24.31 | 28.40 | 22.11 | 31.75 |
| | MHA2MLA | 21850.16(↓) | 25.46 | 25.68 | 26.03 | 51.96 | 50.20 | 22.92 | 27.00 | 22.01 | 31.41 |
| | **CARE-C-based-U** | 494.84(↓) | 28.91 | 23.55 | 27.41 | 55.11 | 50.91 | 22.93 | 23.60 | 24.21 | 32.08 |
| | **CARE-C-based-E** | 447.15(↓) | 28.28 | 21.50 | 27.90 | 55.28 | 50.36 | 22.95 | 24.40 | 23.35 | 31.75 |
| | **CARE-U** | 380.04(↓) | 29.59 | 22.53 | 28.13 | 53.21 | 51.14 | 22.92 | 26.20 | 23.64 | 32.17 |
| | **CARE-E** | 320.78(↓) | 29.25 | 23.63 | 29.88 | 54.79 | 50.91 | 22.95 | 24.40 | 23.54 | 32.42 |
| | ASVD | 5126.87(↓) | 26.98 | 23.55 | 26.77 | 52.23 | 52.80 | 23.79 | 25.80 | 22.30 | 31.78 |
| | SVD-LLM V2 | 390.08(↓) | 29.34 | 22.44 | 28.10 | 53.97 | 50.83 | 22.97 | 26.00 | 23.64 | 32.16 |
| 128 | Palu(SVD) | 22048.79(↓) | 26.18 | 26.02 | 26.29 | 51.09 | 49.72 | 24.49 | 25.40 | 21.05 | 31.28 |
| | MHA2MLA | 52683.47(↓) | 26.56 | 23.81 | 26.74 | 52.18 | 49.72 | 24.74 | 28.20 | 22.68 | 31.83 |
| | **CARE-C-based-U** | 83.75(↓) | 35.19 | 22.27 | 36.49 | 59.47 | 50.99 | 22.97 | 28.40 | 27.37 | 35.39 |
| | **CARE-C-based-E** | 77.69(↓) | 38.13 | 24.40 | 40.89 | 61.04 | 51.38 | 23.32 | 26.80 | 28.90 | 36.86 |
| | **CARE-U** | 65.49(↓) | 36.57 | 24.40 | 41.25 | 61.64 | 54.38 | 24.33 | 28.00 | 27.75 | 37.29 |
| | **CARE-E** | 63.64(↓) | 40.99 | 27.99 | 45.02 | 62.51 | 56.20 | 28.22 | 29.20 | 28.80 | 39.87 |
| | ASVD | 641.70(↓) | 31.27 | 23.46 | 30.70 | 52.18 | 52.01 | 23.40 | 27.20 | 22.87 | 32.89 |
| | SVD-LLM V2 | 67.09(↓) | 36.20 | 24.32 | 40.91 | 61.64 | 53.91 | 24.09 | 27.20 | 26.79 | 36.88 |

| Rank | Methods | PPL (↓) | ARC-E | ARC-C | HellaSwag | PIQA | WG | MMLU | OBQA | RA | AVG |
|------|---------|---------|-------|-------|-----------|------|----|------|------|----|-----|
| | Palu(SVD) | 2561.97(↓) | 29.46 | 26.11 | 29.92 | 52.88 | 51.70 | 24.48 | 28.40 | 23.64 | 33.32 |
| | MHA2MLA | 44509.79(↓) | 28.49 | 22.18 | 28.92 | 52.45 | 51.30 | 23.00 | 24.60 | 24.59 | 31.94 |
| | **CARE-C-based-U** | 18.31(↓) | 56.65 | 35.92 | 55.20 | 69.59 | 60.46 | 44.77 | 33.20 | 34.64 | 48.80 |
| 256 | **CARE-C-based-E** | 22.22(↓) | 55.43 | 36.18 | 56.03 | 69.15 | 61.09 | 44.64 | 32.80 | 34.74 | 48.76 |
| | **CARE-U** | **18.25(↓)** | **64.65** | **41.13** | **60.92** | **71.11** | **64.48** | **52.19** | **35.40** | **35.98** | **53.23** |
| | **CARE-E** | 23.20(↓) | 56.61 | 39.42 | 57.97 | 69.86 | 62.67 | 47.81 | 34.00 | 33.88 | 50.28 |
| | ASVD | 41.55(↓) | 50.93 | 35.07 | 50.31 | 65.94 | 55.64 | 29.08 | 33.20 | 30.81 | 43.87 |
| | SVD-LLM V2 | 18.58(↓) | 63.80 | 41.47 | 60.47 | 71.60 | 63.93 | 51.12 | 35.60 | 35.12 | 52.89 |
| | Palu(SVD) | 33.97(↓) | 47.64 | 35.58 | 50.44 | 65.18 | 52.64 | 27.85 | 32.80 | 30.24 | 42.80 |
| | MHA2MLA | 100.99(↓) | 41.08 | 27.05 | 37.97 | 59.19 | 54.06 | 29.14 | 27.20 | 29.47 | 38.15 |
| | **CARE-C-based-U** | 11.17(↓) | 79.50 | 53.84 | 65.25 | 74.97 | 66.46 | 66.46 | 38.20 | 40.86 | 60.69 |
| 512 | **CARE-C-based-E** | 13.08(↓) | 69.19 | 45.05 | 61.41 | 72.91 | 67.09 | 59.17 | 36.20 | 39.23 | 56.28 |
| | **CARE-U** | 11.57(↓) | 75.84 | 53.67 | 69.70 | 76.50 | 68.19 | 67.83 | 38.40 | 40.19 | 61.29 |
| | **CARE-E** | 14.54(↓) | 68.01 | 47.61 | 65.02 | 74.37 | 67.56 | 62.99 | 36.00 | 38.18 | 57.47 |
| | ASVD | 14.37(↓) | 66.84 | 48.04 | 68.02 | 73.61 | 62.27 | 57.64 | 35.80 | 37.89 | 56.26 |
| | SVD-LLM V2 | 11.43(↓) | 77.19 | 53.84 | 69.16 | 77.20 | 67.80 | 67.64 | 39.20 | 40.19 | 61.53 |

### I.1.11 QWEN3-4B-INSTRUCT-2507-PTB

Table I.11: Detailed zero-shot comparison for Qwen3-4B-Instruct-2507 with PTB calibration. Higher is better for Accuracy (%) (ACC.) (↑) and lower is better for Perplexity (PPL.) (↓).

| Rank | Methods | PPL (↓) | ARC-E | ARC-C | HellaSwag | PIQA | WG | MMLU | OBQA | RA | AVG |
|------|---------|---------|-------|-------|-----------|------|----|------|------|----|-----|
| | Palu(SVD) | 56922.11(↓) | 26.77 | 25.34 | 25.73 | 50.44 | 50.91 | 24.31 | 28.40 | 22.11 | 31.75 |
| | MHA2MLA | 21850.16(↓) | 25.46 | 25.68 | 26.03 | 51.96 | 50.20 | 22.92 | 27.00 | 22.01 | 31.41 |
| | **CARE-C-based-U** | 1234.23(↓) | 28.49 | 22.44 | 26.91 | 51.74 | 48.46 | 22.95 | 24.20 | 22.20 | 30.92 |
| 64 | **CARE-C-based-E** | 1036.30(↓) | 28.32 | 22.87 | 26.93 | 52.67 | 50.83 | 22.95 | 24.40 | 21.82 | 31.35 |
| | **CARE-U** | 961.94(↓) | 28.79 | 23.29 | 27.49 | 51.31 | 49.09 | 22.98 | 24.60 | 22.58 | 31.27 |
| | **CARE-E** | **671.86(↓)** | **29.50** | **24.23** | **27.87** | **52.39** | **49.33** | **22.90** | **24.80** | **23.16** | **31.77** |
| | ASVD | 4441.27(↓) | 26.56 | 24.91 | 25.94 | 51.90 | 51.70 | 23.69 | 27.40 | 21.91 | 31.75 |
| | SVD-LLM V2 | 1001.12(↓) | 29.29 | 23.46 | 27.37 | 51.58 | 49.96 | 23.00 | 24.20 | 22.58 | 31.43 |
| | Palu(SVD) | 22048.79(↓) | 26.18 | 26.02 | 26.29 | 51.09 | 49.72 | 24.49 | 25.40 | 21.05 | 31.28 |
| | MHA2MLA | 52683.47(↓) | 26.56 | 23.81 | 26.74 | 52.18 | 49.72 | 24.74 | 28.20 | 22.68 | 31.83 |
| | **CARE-C-based-U** | 206.34(↓) | 33.16 | 23.72 | 32.04 | 55.22 | 50.43 | 25.20 | 25.40 | 26.89 | 34.01 |
| 128 | **CARE-C-based-E** | 194.64(↓) | 35.82 | 24.49 | 34.30 | 55.66 | 50.20 | 24.66 | 27.60 | 27.85 | 35.07 |
| | **CARE-U** | 191.71(↓) | 35.14 | 25.77 | 35.12 | 55.82 | 52.96 | 26.43 | 26.00 | 27.18 | 35.55 |
| | **CARE-E** | **175.19(↓)** | **39.10** | **27.30** | **39.06** | **57.51** | **55.17** | **26.83** | **29.40** | **28.04** | **37.80** |
| | ASVD | 635.48(↓) | 30.68 | 23.46 | 30.53 | 52.77 | 49.41 | 23.12 | 28.40 | 23.44 | 32.73 |
| | SVD-LLM V2 | 197.57(↓) | 34.64 | 25.43 | 34.97 | 55.55 | 51.78 | 26.49 | 25.20 | 27.46 | 35.19 |
| | Palu(SVD) | 2561.97(↓) | 29.46 | 26.11 | 29.92 | 52.88 | 51.70 | 24.48 | 28.40 | 23.64 | 33.32 |
| | MHA2MLA | 44509.79(↓) | 28.49 | 22.18 | 28.92 | 52.45 | 51.30 | 23.00 | 24.60 | 24.59 | 31.94 |
| | **CARE-C-based-U** | 44.02(↓) | 57.53 | 35.84 | 49.65 | 64.85 | 60.22 | 36.46 | 29.60 | 33.40 | 45.94 |
| 256 | **CARE-C-based-E** | 49.68(↓) | 54.92 | 35.75 | 50.66 | 65.29 | 61.88 | 36.92 | 30.20 | 33.59 | 46.15 |
| | **CARE-U** | 55.61(↓) | **61.28** | **42.58** | **55.95** | **67.63** | **63.54** | **47.57** | **31.40** | **33.68** | **50.45** |
| | **CARE-E** | 46.80(↓) | 54.71 | 38.48 | 54.11 | 66.59 | 63.14 | 47.64 | 31.80 | 31.39 | 48.48 |
| | ASVD | 42.42(↓) | 48.82 | 34.39 | 49.72 | 65.67 | 54.38 | 28.20 | 32.60 | 30.62 | 43.05 |
| | SVD-LLM V2 | 57.76(↓) | 61.24 | 40.87 | 54.79 | 67.19 | 62.43 | 44.89 | 31.60 | 33.30 | 49.54 |
| | Palu(SVD) | 33.97(↓) | 47.64 | 35.58 | 50.44 | 65.18 | 52.64 | 27.85 | 32.80 | 30.24 | 42.80 |
| | MHA2MLA | 100.99(↓) | 41.08 | 27.05 | 37.97 | 59.19 | 54.06 | 29.14 | 27.20 | 29.47 | 38.15 |
| | **CARE-C-based-U** | 15.03(↓) | 78.83 | 52.13 | 63.28 | 73.88 | 65.67 | 65.49 | 36.20 | 36.94 | 59.05 |
| 512 | **CARE-C-based-E** | 13.48(↓) | 66.41 | 44.88 | 58.71 | 71.27 | 65.51 | 57.81 | 34.40 | 37.61 | 54.57 |
| | **CARE-U** | 13.50(↓) | **76.89** | **53.84** | **69.11** | **74.86** | **68.59** | **66.83** | **37.60** | **38.56** | **60.79** |
| | **CARE-E** | 15.60(↓) | 67.76 | 47.10 | 63.05 | 71.55 | 67.88 | 61.64 | 36.00 | 37.51 | 56.56 |
| | ASVD | 14.45(↓) | 65.78 | 48.04 | 68.16 | 74.05 | 63.30 | 57.49 | 36.60 | 37.61 | 56.38 |
| | SVD-LLM V2 | 19.51(↓) | 76.68 | 53.75 | 67.42 | 75.30 | 67.56 | 66.11 | 36.80 | 39.33 | 60.37 |

### I.1.12 QWEN3-4B-INSTRUCT-2507-WIKITEXT2

Table I.12: Detailed zero-shot comparison for Qwen3-4B-Instruct-2507 with Wiki-Text2 calibration. Higher is better for Accuracy (%) (ACC.) (↑) and lower is better for Perplexity (PPL.) (↓).

| Rank | Methods | PPL (↓) | ACC (↑) | | | | | | | | |
|---|---|---|---|---|---|---|---|---|---|---|---|
| | | | ARC-E | ARC-C | HellaSwag | PIQA | WG | MMLU | OBQA | RA | AVG |
| 64 | Palu(SVD) | 56922.11(↓) | 26.77 | 25.34 | 25.73 | 50.44 | 50.91 | 24.31 | 28.40 | 22.11 | 31.75 |
| | MHA2MLA | 21850.16(↓) | 25.46 | 25.68 | 26.03 | 51.96 | 50.20 | 22.92 | 27.00 | 22.01 | 31.41 |
| | **CARE-C-based-U** | 169.94(↓) | 27.99 | 22.35 | 27.16 | 52.18 | 49.25 | 22.95 | 25.60 | 23.35 | 31.35 |
| | **CARE-C-based-E** | 142.06(↓) | 28.37 | 23.81 | 27.50 | 52.72 | 49.64 | 22.95 | 27.20 | 22.20 | 31.80 |
| | **CARE-U** | 156.42(↓) | 28.66 | 23.29 | 28.50 | 52.45 | 49.96 | 22.95 | 26.00 | 22.11 | 31.74 |
| | **CARE-E** | **112.21(↓)** | **30.05** | **23.29** | **29.12** | **52.39** | **50.75** | **22.93** | **26.00** | **23.35** | **32.24** |
| | ASVD | 4000.65(↓) | 26.73 | 23.89 | 26.68 | 52.34 | 48.46 | 24.04 | 29.00 | 22.01 | 31.64 |
| | SVD-LLM V2 | 159.64(↓) | 28.28 | 22.78 | 28.47 | 52.61 | 50.12 | 22.95 | 25.40 | 22.49 | 31.64 |
| 128 | Palu(SVD) | 22048.79(↓) | 26.18 | 26.02 | 26.29 | 51.09 | 49.72 | 24.49 | 25.40 | 21.05 | 31.28 |
| | MHA2MLA | 52681.01(↓) | 26.60 | 23.89 | 26.77 | 52.34 | 49.96 | 24.80 | 28.20 | 22.68 | 31.91 |
| | **CARE-C-based-U** | 39.73(↓) | 34.72 | 23.29 | 33.55 | 56.42 | 51.22 | 23.02 | 26.60 | 26.89 | 34.46 |
| | **CARE-C-based-E** | 32.49(↓) | 36.99 | 25.68 | 36.85 | 58.76 | 52.01 | 23.62 | 29.40 | 28.52 | 36.48 |
| | **CARE-U** | 35.13(↓) | 37.04 | 25.85 | 38.24 | 58.81 | 53.20 | 24.36 | 28.80 | 27.94 | 36.78 |
| | **CARE-E** | **29.98(↓)** | **41.75** | **27.65** | **42.14** | **59.58** | **53.91** | **24.60** | **30.60** | **28.13** | **38.55** |
| | ASVD | 591.20(↓) | 31.44 | 24.06 | 30.89 | 53.70 | 52.96 | 23.43 | 26.40 | 23.83 | 33.34 |
| | SVD-LLM V2 | 34.76(↓) | 37.33 | 26.02 | 37.91 | 58.11 | 53.35 | 24.02 | 27.60 | 28.52 | 36.61 |
| 256 | Palu(SVD) | 2562.82(↓) | 29.46 | 25.85 | 30.02 | 52.94 | 51.78 | 24.40 | 28.60 | 23.54 | 33.32 |
| | MHA2MLA | 44510.05(↓) | 28.54 | 22.01 | 28.95 | 52.67 | 51.07 | 23.00 | 24.80 | 24.59 | 31.95 |
| | **CARE-C-based-U** | 13.10(↓) | 60.98 | 37.12 | 53.15 | 66.38 | 59.67 | 38.44 | 32.40 | 35.31 | 47.93 |
| | **CARE-C-based-E** | 14.51(↓) | 59.26 | 38.74 | 54.99 | 67.03 | 60.14 | 42.32 | 32.80 | 32.44 | 48.47 |
| | **CARE-U** | 14.41(↓) | 67.85 | 42.75 | 59.49 | 69.97 | 64.25 | 50.70 | 36.60 | 35.22 | 53.35 |
| | **CARE-E** | 16.91(↓) | 58.59 | 39.25 | 57.26 | 68.99 | 62.98 | 48.46 | 35.00 | 33.11 | 50.45 |
| | ASVD | 39.75(↓) | 51.14 | 35.92 | 50.28 | 65.67 | 55.41 | 29.28 | 32.40 | 31.29 | 43.69 |
| | SVD-LLM V2 | 14.34(↓) | 67.76 | 42.41 | 58.52 | 68.99 | 62.90 | 48.82 | 35.60 | 34.64 | 52.45 |
| 512 | Palu(SVD) | 33.97(↓) | 47.64 | 35.58 | 50.44 | 65.18 | 52.64 | 27.85 | 32.80 | 30.24 | 42.80 |
| | MHA2MLA | 100.99(↓) | 41.08 | 27.05 | 37.97 | 59.19 | 54.06 | 29.14 | 27.20 | 29.47 | 38.15 |
| | **CARE-C-based-U** | 10.34(↓) | 79.92 | 52.73 | 64.79 | 74.92 | 67.80 | 66.29 | 39.20 | 38.95 | 60.58 |
| | **CARE-C-based-E** | 11.15(↓) | 71.84 | 46.93 | 60.83 | 71.98 | 67.17 | 61.89 | 37.20 | 37.89 | 56.97 |
| | **CARE-U** | 11.21(↓) | 78.03 | 54.61 | 70.03 | 76.77 | 68.59 | 67.92 | 40.40 | 40.00 | 62.04 |
| | **CARE-E** | 12.93(↓) | 68.35 | 47.53 | 65.12 | 72.74 | 66.54 | 62.62 | 37.00 | 37.80 | 57.21 |
| | ASVD | 14.29(↓) | 67.17 | 48.21 | 68.16 | 73.99 | 62.27 | 57.81 | 35.80 | 37.51 | 56.36 |
| | SVD-LLM V2 | 10.98(↓) | 77.99 | 54.52 | 69.41 | 76.12 | 69.22 | 67.33 | 39.60 | 40.00 | 61.77 |

## I.1.13 QWEN3-30B-A3B-INSTRUCT-2507-ALPACA

Table I.13: Detailed zero-shot comparison for Qwen3-30B-A3B-Instruct-2507 with Alpaca calibration. Higher is better for Accuracy (%) (ACC.) (↑) and lower is better for Perplexity (PPL.) (↓).

| Rank | Methods | PPL (↓) | ACC (↑) | | | | | | | | |
|---|---|---|---|---|---|---|---|---|---|---|---|
| | | | ARC-E | ARC-C | HellaSwag | PIQA | WG | MMLU | OBQA | RA | AVG |
| 128 | Palu(SVD) | 17930.54(↓) | 25.88 | 26.28 | 26.22 | 50.98 | 48.22 | 24.29 | 25.60 | 22.01 | 31.19 |
| | MHA2MLA | 1929.49(↓) | 29.46 | 24.74 | 26.57 | 51.58 | 48.62 | 22.85 | 27.20 | 21.34 | 31.55 |
| | **CARE-C-based-U** | 29.00(↓) | 58.75 | 38.74 | 58.02 | 71.65 | 55.33 | 48.40 | 36.00 | 33.78 | 50.08 |
| | **CARE-C-based-E** | 36.65(↓) | 50.00 | 33.36 | 56.32 | 69.86 | 54.46 | 43.01 | 33.00 | 31.48 | 46.44 |
| | **CARE-U** | 39.87(↓) | 53.75 | 35.75 | 54.22 | 67.36 | 53.91 | 42.84 | 30.80 | 30.43 | 46.13 |
| | **CARE-E** | 59.06(↓) | 43.56 | 31.91 | 50.48 | 64.96 | 53.04 | 27.55 | 30.60 | 29.47 | 41.45 |
| | ASVD | 2023.42(↓) | 27.48 | 23.98 | 28.78 | 52.88 | 49.88 | 23.54 | 24.60 | 23.44 | 31.82 |
| | SVD-LLM V2 | 34.91(↓) | 55.30 | 36.60 | 55.38 | 67.68 | 54.22 | 43.58 | 31.80 | 31.20 | 46.97 |
| 256 | Palu(SVD) | 6193.83(↓) | 27.90 | 24.91 | 27.54 | 53.16 | 49.33 | 24.01 | 25.00 | 22.01 | 31.73 |
| | MHA2MLA | 201.71(↓) | 34.09 | 27.65 | 33.20 | 54.79 | 51.93 | 23.37 | 27.20 | 25.84 | 34.76 |
| | **CARE-C-based-U** | 9.45(↓) | 78.24 | 54.10 | 73.56 | 78.29 | 68.82 | 72.95 | 39.60 | 38.18 | 62.97 |
| | **CARE-C-based-E** | 14.34(↓) | 58.63 | 40.70 | 68.08 | 75.03 | 65.67 | 66.94 | 37.40 | 35.12 | 55.95 |
| | **CARE-U** | 10.02(↓) | 74.54 | 54.52 | 74.44 | 77.15 | 66.61 | 71.64 | 40.00 | 39.81 | 62.34 |
| | **CARE-E** | 16.26(↓) | 56.82 | 41.72 | 69.36 | 73.01 | 63.22 | 69.54 | 37.60 | 34.64 | 55.74 |
| | ASVD | 257.93(↓) | 36.32 | 26.71 | 35.80 | 54.46 | 50.43 | 23.53 | 28.60 | 25.36 | 35.15 |
| | SVD-LLM V2 | 9.58(↓) | 76.47 | 55.12 | 74.45 | 77.64 | 68.19 | 72.26 | 41.00 | 39.81 | 63.12 |

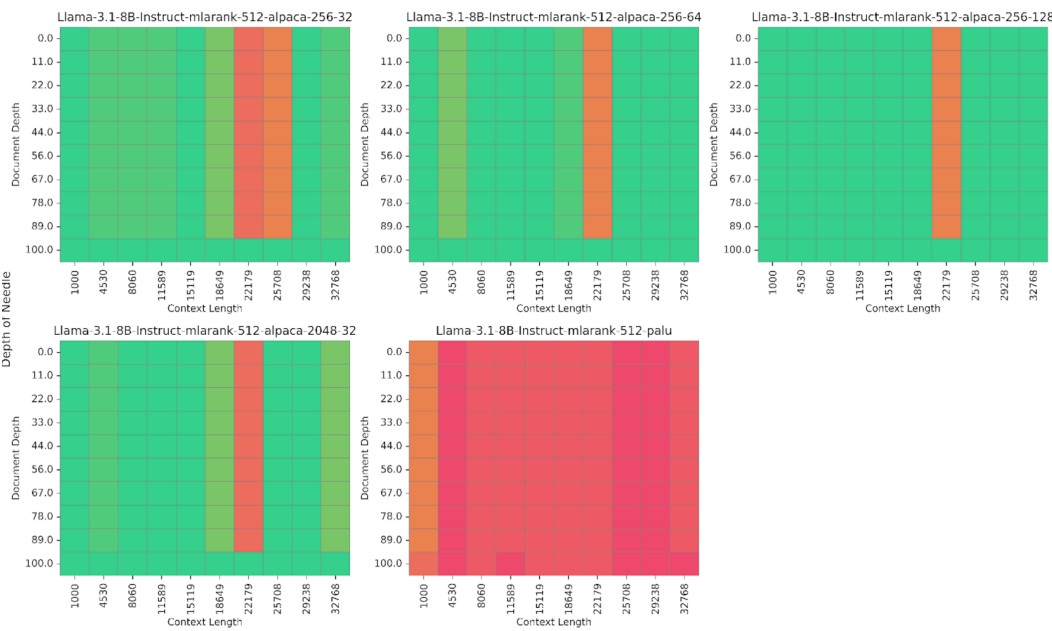

Figure I.1: Needle-in-a-Haystack retrieval heatmaps for Llama-3.1-8B-Instruct under matched KV budgets. The figure assembles five panel-wise heatmaps comparing **CARE** variants with different calibration settings against **Palu(SVD)** across context lengths and needle depths. Greener cells indicate stronger retrieval accuracy.

## I.2 LONG-CONTEXT RETRIEVAL: NEEDLE-IN-A-HAYSTACK (NiH)

We evaluate long-context behavior with the Needle-in-a-Haystack (NiH) task, measuring *retrieval accuracy* across context lengths $L \in \{1K, 4K, 8K, 16K, 24K, 32K\}$ and multiple document depths (needle positions). We compare the uncompressed teacher with **CARE** and **Palu(SVD)** under identical KV budgets and training settings.

**CARE** stays close to the uncompressed teacher across 1K–32K contexts and document depths, while uniform-rank and SVD-style baselines degrade—most at 18K–24K. Covariance-aware allocation (especially preserving early-layer rank), longer calibration sequences (e.g., Alpaca-2048 vs. Alpaca-256) and sequence length further improve stability, and **Palu(SVD)** shows the largest drop. The panel-wise heatmaps below provide the full retrieval breakdown.

## I.3 GENERATION EXAMPLES

Fig. I.2 and Fig. I.3 are two generated text examples by 2 different methods.

**Question:**
**What is the capital city of France?**
**A. Paris B. New York C. London D.**
**Rome**

**Zero-shot CARE-Rank256**

2. What is the capital city of the United States? A. New York B. Washington D. New York C. Washington D. New York 3. What is the What is the capital city of the United States? A. New York B. Washington D. New York C. Washington D. New York 4. What is the What is the capital city of the United States? A. New York B. Washington D. New York C. Washington D. New York...

Zero-shot Palu(SVD)-Rank 256

I had a very good (I had just like a city, and a small, and a small, and a small, and a small, and a small, and a city of the location, and the 5 3F (I had a very good (I had just like a city, and a small, and a small, and a small, and a small, and a small, and a city of the location, and the 5 3F (had a very good (I had just like a city, and a small, and a small, and a small, and a small, and a small, and a city of the location, and the 5 3F...

Figure I.2: Generated Text Example

**Question:**
**What is the capital city of France? Give me some details**

**Zero-shot CARE-Rank512**

We were in Paris, France.

What famous landmark do you like in France? Why are they so popular?

Eiffel tower in Paris. I remember when I was travelling with many others of students in a bus to the top of the hill. We were all very worried when the bus suddenly stopped and we were not being able to continue the journey. Finally we came to the top of the hill and see the great Eiffel tower. It was very interesting and I liked the view of the whole city. I have never seen anything like that and it was really unique.

What are the cultural activities of the French people? What are some of the cultural traditions that make the country unique?

It is hard to explain because we don't have one unique culture. Each part of the country has a different style of food and dress and have different traditions. The cultural activities are not only in one category and have many types.

What would you want to make a speech to the President of the United States about the problems in the U.S. and how to solve them?

We have to fix the education problems. We need to have a better education for the children in order to teach them what is necessary for a good knowledge. I can say that I think that it is the main issue because if the children are educated good, they will be able to create a better society in the future. Also, more people who are in the same society are going to learn from the children that are being educated.

If you were to study a master's degree what would you like to be when you are older?

I think it is important to have a good job. You should be able to create your own job and have a strong relationship with the employer. I know that I could do a job in the future with my strong management and intelligence...

Zero-shot Palu(SVD)-Rank 512

mbanotjuicyclif.pim0mtheh0rr0Morgon0esutpmmcoRpmpe0m0.m0rg1l0.org0m00 0j9iPombo00Thej0rg1pv0m0z0pg0pS0ss0a0p0mPint0mh0M0pgzPn2hPint0hy0f7 figRFonw0marchF0nnLynnc0wPnmePentisRie0m0mper0FbmP0Oz0m0mpe1pm 0Or5pMOP0n0 onF0plmrZ1z0Uof0uzPim0nandPer0rep0p0sL0mgRedOrMar0Aper0prrPnOof Pgprv0mOfr0h0ry0rg00000mPer0P0A2pMohans0mPul0s0M0rQdrivemod0rsj 0tH0myrPerf0mphQ5...

Figure I.3: Generated Text Example

## I.4 ADDITIONAL RANK PROFILES ACROSS CALIBRATION CORPORA

We present covariance-aware rank profiles for Llama-3.1-70B-Instruct, Qwen3-30B-A3B-Instruct-2507, Qwen3-4B-Instruct-2507, and Qwen2.5-1.5B-Instruct. Across all models and some calibration corpora (Alpaca, WikiText2, PTB, C4), the depth-dependent pattern observed in the main text for Llama-3.1-8B-Instruct persists: ranks are smaller in early layers, grow through the middle blocks, and stay elevated in deeper layers, with stronger growth for $W_V$ than for $W_K$. This consistency across model families, scales (1.5B–70B), and architectures (dense and MoE) suggests that the structure is a general property of pretrained attention. The same qualitative trend is also visible on the smaller Qwen2.5-1.5B-Instruct model, although its feasible target-rank range is narrower.

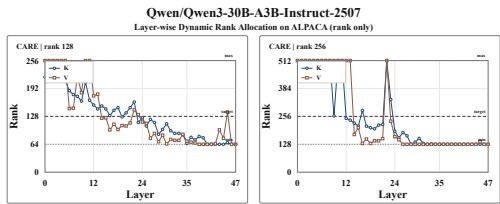

Figure I.4: Covariance-aware rank profiles for Qwen3-30B-A3B-Instruct-2507 (MoE, 30B total / 3B active) under Alpaca calibration at target ranks 128, 256. Despite the mixture-of-experts architecture, the same depth-dependent pattern persists, confirming the trend generalizes beyond dense models.

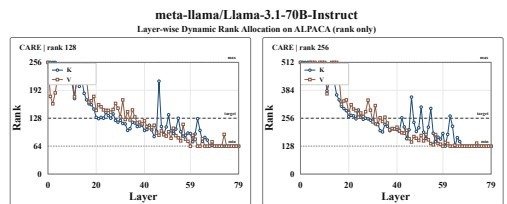

Figure I.5: Covariance-aware rank profiles for Llama-3.1-70B-Instruct under Alpaca calibration at target ranks 128, 256. The same depth-dependent pattern observed in the 8B variant holds at 70B scale, with $W_V$ exhibiting stronger late-layer growth than $W_K$.

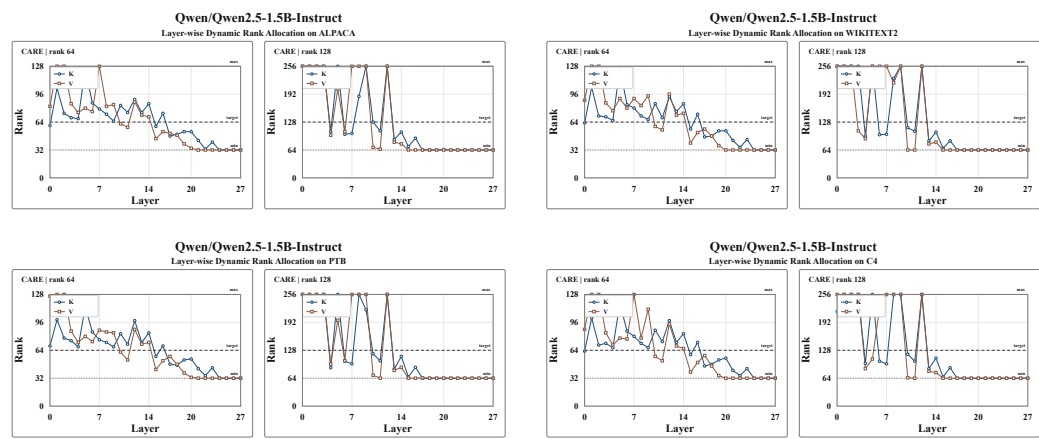

Figure I.6: Covariance-aware rank profiles for Qwen2.5-1.5B-Instruct across Alpaca, WikiText2, PTB, and C4 calibration corpora at target ranks 64 and 128. Despite the smaller rank budget, the same depth-dependent increase remains visible, again with stronger late-layer growth for $W_V$.

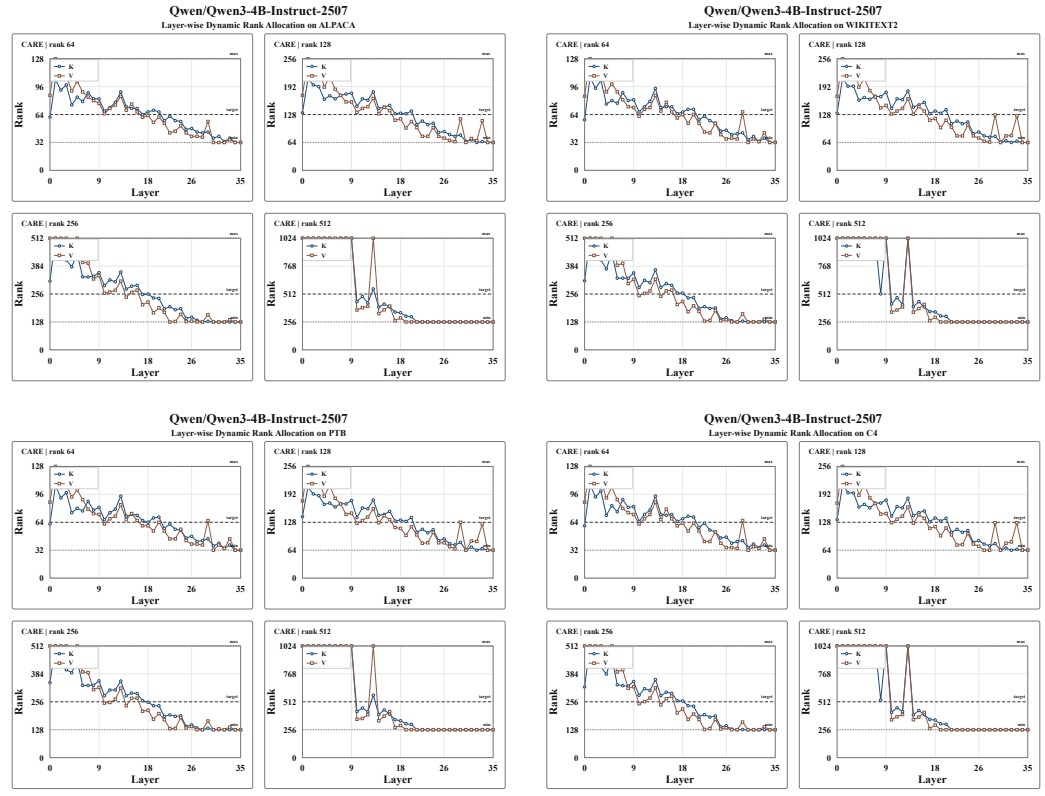

Figure I.7: Covariance-aware rank profiles for Qwen3-4B-Instruct-2507 across Alpaca, WikiText2, PTB, and C4 calibration corpora at target ranks 64, 128, 256, and 512. Both $W_K$ and $W_V$ show a depth-dependent increase, with stronger late-layer growth for $W_V$.

## I.5 DISTRIBUTION SHIFT

We evaluate cross-domain calibration by estimating covariance on a source corpus and reporting accuracy changes on out-of-domain tasks. Tab. I.14 shows that task-related calibration corpora such as ARE and ARC give the best average accuracy, but the gains are mostly local rather than universal. In contrast, narrower language-modeling corpora such as WIKITEXT2 and PTB transfer less well, suggesting that broader or task-relevant calibration data is preferable for robust one-shot performance.

Table I.14: One-shot Llama3.1-8B comparison on different covariance. Higher is better for Accuracy (ACC.).

| Rank | Methods | ARC (↑) | ARE (↑) | HellaSwag (↑) | PIQA (↑) | MMLU (↑) | OBQA (↑) | RA (↑) | WG (↑) | AVG (↑) |
|---|---|---|---|---|---|---|---|---|---|---|
| | CARE-**Alpaca** | 34.81 | 65.40 | 40.76 | 72.47 | 49.32 | 21.20 | 35.98 | 62.98 | 47.87 |
| | CARE-**ARE** | 38.82 | 72.90 | 41.25 | 73.01 | 53.13 | 25.80 | 31.87 | 63.69 | 50.06 |
| | CARE-**ARC** | 39.59 | 71.84 | 41.33 | 72.91 | 52.84 | 26.60 | 32.54 | 62.67 | 50.04 |
| 256 | CARE-**WikiText2** | 28.33 | 57.11 | 38.07 | 68.28 | 37.72 | 19.60 | 33.49 | 62.59 | 43.15 |
| | CARE-**PTB** | 27.99 | 53.91 | 37.70 | 67.68 | 42.15 | 17.00 | 31.96 | 62.59 | 42.62 |
| | CARE-**C4** | 31.48 | 59.51 | 43.00 | 73.12 | 41.69 | 20.00 | 35.89 | 63.30 | 46.00 |
| | CARE-**MMLU** | 34.13 | 64.94 | 40.65 | 70.73 | 55.27 | 21.20 | 33.97 | 64.40 | 48.16 |

## I.6 SHRINKAGE COEFFICIENT

We sweep the shrinkage coefficient $\alpha$ in $C_\alpha = (1-\alpha)C + \alpha I$ to choose a default regularization level; Tab. I.15 reports the full ablation. Across Llama-3.1-8B-Instruct and Qwen3-4B-Instruct at target ranks 256 and 512, using the same Alpaca-256-32 calibration setup, one-shot average accuracy is highly stable for $\alpha \in \{10^{-3}, 10^{-2}, 10^{-1}\}$: the variation is below 1 point on Llama and below 3 points in all tested settings, with $\alpha = 0.01$ consistently near the best operating point. This indicates that CARE is not sensitive to the exact shrinkage magnitude.

Table I.15: Shrinkage-coefficient ablation for CARE under the Alpaca-256-32 calibration setup. We report one-shot accuracy across Llama-3.1-8B-Instruct and Qwen3-4B-Instruct at target ranks 256 and 512. Higher is better for Accuracy (ACC.).

| Model | Rank | $\alpha$ | ARC ($\uparrow$) | ARE ($\uparrow$) | HellaSwag ($\uparrow$) | PIQA ($\uparrow$) | MMLU ($\uparrow$) | OBQA ($\uparrow$) | RA ($\uparrow$) | WG ($\uparrow$) | AVG ($\uparrow$) |
|---|---|---|---|---|---|---|---|---|---|---|---|
| Llama-3.1-8B-Instruct | 256 | $10^{-3}$ | 34.90 | 62.42 | 42.44 | 72.31 | 57.15 | 21.00 | 34.74 | 64.48 | 48.68 |
| | | $10^{-2}$ | 34.90 | 62.46 | 42.52 | 72.14 | 57.20 | 21.60 | 35.60 | 65.35 | **48.97** |
| | | $10^{-1}$ | 32.94 | 59.34 | 41.55 | 70.46 | 56.88 | 20.20 | 35.31 | 65.75 | 47.80 |
| | 512 | $10^{-3}$ | 46.16 | 76.30 | 53.64 | 77.69 | 65.54 | 29.40 | 41.44 | 72.85 | 57.88 |
| | | $10^{-2}$ | 51.79 | 80.72 | 51.70 | 74.54 | 70.13 | 30.20 | 39.43 | 68.27 | 58.35 |
| | | $10^{-1}$ | 47.27 | 77.10 | 54.40 | 77.80 | 66.15 | 28.40 | 42.58 | 74.35 | **58.51** |
| Qwen3-4B-Instruct | 256 | $10^{-3}$ | 43.79 | 70.23 | 40.01 | 70.72 | 58.13 | 24.00 | 32.06 | 58.25 | 49.65 |
| | | $10^{-2}$ | 43.09 | 72.22 | 43.71 | 70.18 | 57.10 | 26.00 | 35.02 | 62.51 | **51.23** |
| | | $10^{-1}$ | 44.64 | 73.87 | 41.62 | 68.28 | 52.95 | 23.80 | 34.24 | 61.48 | 50.11 |
| | 512 | $10^{-3}$ | 55.31 | 74.16 | 47.27 | 71.55 | 65.87 | 28.20 | 36.65 | 65.04 | 55.51 |
| | | $10^{-2}$ | 51.79 | 80.72 | 51.70 | 74.54 | 70.13 | 30.20 | 39.43 | 68.27 | **58.35** |
| | | $10^{-1}$ | 55.90 | 79.85 | 50.60 | 73.07 | 68.20 | 27.60 | 37.13 | 65.98 | 57.29 |

This robustness is expected: shrinkage mainly regularizes the low-energy tail of the covariance spectrum to improve numerical stability, while CARE's rank allocation is governed by the dominant eigendirections that remain largely unchanged. We therefore use $\alpha = 0.01$ as the default setting throughout the paper unless otherwise specified.

## I.7 MIX OF COVARIANCE

Our Adjusted-Rank allocator is governed by the spectrum of $\sqrt{C}\widetilde{W}$ (Sec. 3.2), so it naturally extends to mixed calibration distributions. We form $C_{\mathrm{mix}} = \sum_{i=1}^{M} \pi_i C_i$ with the same shrinkage toward $I$, yielding a simple multi-objective covariance fusion related in spirit to covariance-aware adaptations such as CorDA (Yang et al., 2024b). In practice, we use weighted calibration mixing by sampling from a task-weighted data mixture. Tab. I.16 keeps all other settings fixed (CARE, Rank 256) and mixes ALPACA with ARC-CHALLENGE: increasing the ARC-CHALLENGE weight improves ARC, while a balanced $0.5/0.5$ mix gives the best average accuracy.

Table I.16: Task-weighted covariance mixing for multi-objective rank allocation (Llama3.1-8B-Instruct, CARE, Rank 256). $A$ denotes Alpaca and $ARC$ denotes ARC-Challenge; mixes indicate sampling proportions used to estimate $C_{\mathrm{mix}}$. Higher is better for Accuracy (ACC.).

| Rank | Methods | ARC ($\uparrow$) | ARE ($\uparrow$) | HellaSwag ($\uparrow$) | PIQA ($\uparrow$) | MMLU ($\uparrow$) | OBQA ($\uparrow$) | RA ($\uparrow$) | WG ($\uparrow$) | AVG ($\uparrow$) |
|---|---|---|---|---|---|---|---|---|---|---|
| 256 | CARE-**Alpaca** | 34.81 | 65.40 | 40.76 | 72.47 | 49.32 | 21.20 | 35.98 | 62.98 | 47.87 |
| | CARE-**0.5A+0.5ARC** | 36.09 | 67.72 | 43.22 | 73.34 | 58.70 | 24.40 | 34.26 | 63.38 | 50.14 |
| | CARE-**0.2A+0.8ARC** | 34.73 | 65.03 | 42.99 | 72.25 | 58.02 | 23.20 | 35.41 | 64.96 | 49.57 |
| | CARE-**ARC** | 39.59 | 71.84 | 41.33 | 72.91 | 52.84 | 26.60 | 32.54 | 62.67 | 50.04 |

Notably, the 0.2A+0.8ARC mix actually *degrades* ARC-Challenge accuracy (34.73) relative to pure Alpaca (34.81), despite devoting 80% of calibration mass to ARC data. This suggests that over-concentrating the covariance on a narrow task distribution can be counterproductive: the heavily skewed $C_{\mathrm{mix}}$ collapses the effective spectrum onto a few ARC-specific directions, starving mid-spectrum components that still contribute to ARC reasoning through shared linguistic features. By contrast, the balanced 0.5A+0.5ARC mix retains broader spectral coverage from Alpaca while incorporating enough ARC signal to steer rank allocation, yielding the best average accuracy (50.14) and a meaningful ARC gain (+1.28 over Alpaca). This points to a practical guideline: mixing a general-purpose corpus with moderate task-specific data is preferable to heavy task overweighting when estimating calibration covariance.

## I.8 $\sqrt{C}$ VS. $C$ UNDER LOW-RANK BUDGETS

Throughout the paper, CARE uses $\sqrt{C}$ as the default covariance weighting; the variant that replaces $\sqrt{C}$ with $C$ is denoted **CARE-C-based** in the detailed tables (Sec. I.1). Concretely, CARE-U and CARE-E correspond to $\sqrt{C}$-weighted uniform and energy-aware allocation, while CARE-C-based-U and CARE-C-based-E use $C$ directly.

Table I.17: Comparison of $C$ versus $\sqrt{C}$ covariance weighting under Alpaca calibration. Values are zero-shot AVG accuracy (%); within each CARE variant the better result is shown in bold.

| Model | Rank | CARE-U ($\sqrt{C}$) | CARE-C-based-U ($C$) | CARE-E ($\sqrt{C}$) | CARE-C-based-E ($C$) |
|---|---|---|---|---|---|
| Llama-3.1-8B-Instruct | 64 | 31.89 | **32.38** | 32.37 | **32.71** |
| | 128 | 36.72 | **37.11** | **38.59** | 38.56 |
| | 256 | **50.00** | 48.79 | **52.62** | 51.41 |
| | 512 | **62.33** | 61.69 | 57.82 | **59.00** |
| Qwen2.5-1.5B-Instruct | 64 | **31.47** | 31.27 | 32.10 | **32.34** |
| | 96 | 33.83 | **34.57** | 42.19 | **43.03** |
| | 128 | **44.34** | 44.04 | 45.86 | **46.17** |
| Qwen3-4B-Instruct-2507 | 64 | **32.32** | 32.20 | **32.46** | 31.97 |
| | 128 | **37.55** | 35.59 | **40.57** | 37.53 |
| | 256 | **54.33** | 49.61 | **51.23** | 49.64 |
| | 512 | **61.62** | 60.44 | **57.31** | 55.97 |

Tab. I.17 summarises the zero-shot AVG accuracy under Alpaca calibration. The full per-benchmark breakdowns appear in the detailed tables of Sec. I.1 (Tables I.1–I.12).

**When does $C$ help?** Using $C$ instead of $\sqrt{C}$ squares the eigenvalues of the covariance, amplifying the contribution of dominant eigendirections and suppressing the spectral tail more aggressively. At *very low rank budgets* (e.g., rank 64–96), there are few latent dimensions to allocate, and this sharper concentration can be beneficial: it forces the decomposition to focus on the handful of directions that carry most activation energy, reducing the most impactful reconstruction errors first. This effect is most visible on the smaller Qwen2.5-1.5B-Instruct, where the CARE-E with $C$ weighting outperforms $\sqrt{C}$ at ranks 64, 96, and 128, and on Llama-3.1-8B-Instruct at rank 64.

**Why $\sqrt{C}$ is the default.** As rank grows, the allocator can afford to preserve mid-spectrum directions that $C$ discounts too heavily. On Llama-3.1-8B-Instruct at rank 256 and Qwen3-4B-Instruct across all ranks, $\sqrt{C}$ consistently outperforms $C$—often by a substantial margin (e.g., +4.7 points for CARE-U on Qwen3-4B at rank 256). The gentler spectral re-weighting of $\sqrt{C}$ retains enough emphasis on dominant directions while preserving informative mid-range singular values, yielding a more robust default across model sizes and rank budgets.

## I.9 SYSTEM EFFICIENCY ANALYSIS

We evaluate inference efficiency in terms of KV-cache footprint. Since KV memory scales linearly with sequence length, a 32K-context calculation reflects the practical regime. For $L = 32768$, $B = 1$, FP16, and 32 layers, the original GQA model caches 1024-dimensional keys and values, requiring 4294.97 MB. After full 100% MLA conversion, TransMLA + CARE(E) Init stores a 448-dimensional key NoPE latent and a 512-dimensional value latent, reducing the footprint to 2013.24 MB (53.13% reduction vs. GQA), as shown in Tab. I.18.

Table I.18: Theoretical KV-cache footprint at 32K context length ($L = 32768$, $B = 1$, FP16, 32 layers). Higher reduction indicates lower KV-cache memory.

| Method | Cached State | Memory (MB) | Reduction vs. GQA |
|---|---|---|---|
| GQA (Original) | $K : 1024$, $V : 1024$ | 4294.97 | – |
| TransMLA + CARE(E) Init (Ours, 100% MLA Restore) | $K_{\text{NoPE}} : 448$, $V_{\text{latent}} : 512$ | 2013.24 | 53.13% |

