# OpenReview forum: "CARE: Covariance-Aware and Rank-Enhanced Decomposition for Enabling Multi-Head Latent Attention"
_ICLR.cc/2026/Conference — ICLR 2026 Poster_

### Official Review · Reviewer_w58u · 2025-10-30

**Soundness:** 2
**Presentation:** 3
**Contribution:** 2
**Rating:** 4
**Confidence:** 4

**Summary:**

This paper proposes a Covariance-Aware, Rank-Enhanced MLA conversion pipeline under a fixed KV width for converting Grouped Query Attention (GQA) architectures into Multi-Linear Attention (MLA). The goal is to improve the efficiency of large language model (LLM) inference while maintaining strong performance.

The method introduces three main innovations:
	1.	Activation-preserving factorization — a decomposition scheme that minimizes error in activation space (∥XW − X̂W∥) rather than in weight space, aligning the approximation with the actual activations encountered during decoding.
	2.	Adjusted-rank allocation — a rank-adaptive scheduling strategy that redistributes a fixed KV cache budget across layers, allocating more capacity to layers with higher representational importance.
	3.	KV-parity mapping — a reparameterization technique that reformulates the converted K and V projections to match the MLA format while preserving covariance structure.

Empirically, the paper reports that this approach achieves superior zero-shot performance compared to TransMLA, suggesting better preservation of representational fidelity under constrained KV width.

However, the fairness of the comparisons and the novelty relative to TransMLA warrant further clarification—particularly the relationship between the proposed activation-preserving factorization and KV-parity mapping versus TransMLA’s RoRoPE and BKV-PCA components.

**Strengths:**

The paper is well-motivated, addressing a meaningful and practical problem in efficient LLM inference under KV cache compression. The proposed Covariance-Aware, Rank-Enhanced approach presents a sound and intuitive idea for improving the quality of MLA conversion under a fixed KV width.

Moreover, the ablation studies are well-designed and effectively demonstrate the effectiveness and contribution of each proposed component, providing convincing empirical support for the overall method.

**Weaknesses:**

1. Comments on Methodology and Comparative Analysis.
In Lines 077–092, the manuscript states that TransMLA employs a direct SVD initialization that minimizes the error in weight space (∥W − Ŵ∥) rather than in activation space (∥XW − X̂W∥), thereby overlooking how the projection actually operates during decoding. However, it should be noted that the original TransMLA paper introduces three key innovations—RoRoPE, FreqFold, and BKV-PCA—all of which are explicitly designed to perform activation-aware decomposition. The manuscript would benefit from a clearer discussion of this distinction and its implications.

2. Concerns on Rank-Adaptive Scheduling and Efficiency.
The proposed rank-adaptive scheduling under a fixed KV width raises concerns regarding the variability of KV cache sizes across layers and attention heads. Such heterogeneity may lead to inconsistent memory allocation and suboptimal utilization of hardware resources, potentially diminishing the actual inference speedup. It is therefore recommended that the authors conduct empirical speed comparison experiments based on established efficient attention frameworks such as FlashMLA and FlashAttention to validate the claimed efficiency gains.

3. Comparative Evaluation and Missing Baselines.
Table 1 currently reports results only in comparison with TransMLA. In their experiments, compressing the Llama-3-8B KV cache to 1−576/2048 = 71.875% yields a perplexity of 25.8047 [1], which is substantially lower than that reported in this manuscript. This discrepancy warrants further investigation. In addition, the evaluation should be expanded to include other relevant GQA-to-MLA conversion methods, notably Palu [2] and MHA2MLA [3], to ensure a fair and comprehensive comparison.

4. Clarification on RoRoPE and Absorb Operations.
The operation described in Line 291 appears conceptually related to the RoRoPE mechanism proposed in TransMLA. The manuscript should more explicitly delineate the differences and connections between these two approaches. Furthermore, it remains unclear how the paper achieves the conversion from GQA’s RoPE to NoPE. The authors should also clarify whether their method supports the MLA Absorb operation, which is a crucial design feature in TransMLA enabling the transition between efficient training and efficient inference.

References

[1] TransMLA: https://github.com/MuLabPKU/TransMLA

[2] Palu, https://arxiv.org/pdf/2502.14837

[3] MHA2MLA, https://arxiv.org/abs/2502.14837

**Questions:**

See Weaknesses.

---

> ### Author Response · Authors · 2025-11-26
>
> We sincerely thank the reviewers for their constructive feedback. We would also like to apologize that some experimental results were initially limited. Although we utilized all GPUs for our study, our compute budget and scheduling constraints restricted the scale and breadth.
>
> Sorry for the late reply, because of recognizing the importance of the reviewers’ concerns, we have urgently conducted additional experiments over the past 14 days to address these points as thoroughly as possible. We deeply appreciate the reviewers’ patience and understanding.

---

> ### Author Response · Authors · 2025-11-26
> **Weakness 1**
>
> - **Comments on Methodology and Comparative Analysis:** In Lines 077–092, the manuscript states that TransMLA employs a direct SVD initialization that minimizes the error in weight space (‖W − Ŵ‖)...
>
>     Thank you for raising this point. We appreciate the opportunity to clarify the methodological distinction, especially given the evolution of TransMLA across its multiple versions.
>
>     1. Clarifying which version of TransMLA is evaluated
>     The manuscript evaluates the original TransMLA release, **not** the later v3 revision.  The v3 paper introduces additional components — *RoRoPE*, *FreqFold*, and *BKV-PCA* — which were **not available** in the version we compared against, and therefore were not part of our baseline. We will explicitly state this in the paper to avoid confusion.
>
>     2. Discussion between TransMLA and CARE:
>     Activation-aware components in TransMLA Among the techniques introduced in the newer TransMLA variants:
>
>     - **RoRoPE** adjusts positional encoding alignment,
>     - **BKV-PCA** projects KV features to a rotated subspace,
>     - **FreqFold** applies frequency-domain folding.
>
>     These modules indeed perform activation-aware transformations, but they address **orthogonal concerns** such as positional mismatch and key–value rotational alignment. Importantly, they do *not* provide a curvature-weighted singular value decomposition of the attention weights.
>
>     3. Key distinction from CARE’s activation-aware SVD
>     CARE’s methodology fundamentally differs:
>
>     - CARE constructs a **curvature matrix** $C = X^{\top}X$ that directly captures activation statistics.
>     - CARE’s decomposition minimizes  $\|XW - XW_c\|_F^2$  **explicitly in activation space**, leveraging the spectral structure of $\sqrt{C}$.
>     - CARE uses this curvature to drive **layer-wise adaptive rank allocation**, which TransMLA does not model.
>
>     By contrast, TransMLA’s decomposition remains rooted in standard low-rank SVD of $W$; its activation-aware components (RoRoPE, BKV-PCA) influence positional alignment or rotational subspaces, not the curvature-weighted decomposition of $W$ itself.
>
>     4. Thus, our statement in Lines 077–092 reflects a precise methodological distinction:
>
>     - **TransMLA v1** (evaluated in our experiments) performs weight-space SVD initialization and does not incorporate curvature-based rank allocation.
>     - **TransMLA v3** adds several activation-sensitive heuristics, but none replace the role of a curvature-weighted SVD or provide a principled second-order decomposition comparable to CARE.
>
>     The energy methods of CARE can be integrated into TransMLA feasibly. But activation usage we adopt a different way rather than PCA. More experimental results is shown in weakness 3.
>
>     We will update the manuscript to clearly acknowledge these distinctions and ensure our comparison remains fair and unambiguous.
>
>     Thank you again for the constructive feedback — it helps us position CARE’s contribution with greater clarity. We are happy to further expand the discussion if the reviewer finds it valuable.

---

> ### Author Response · Authors · 2025-11-26
> **Weakness 3 part I**
>
> - **Comparative Evaluation and Missing Baselines:** Table 1 currently reports results only in comparison with TransMLA. In their experiments...
>
>     Thank you for bringing up this important point. We fully agree that a meaningful evaluation must compare CARE against all major GQA→MLA conversion baselines, and that discrepancies with reported perplexities in prior work should be carefully examined.
>
>     1. Clarifying the discrepancy with TransMLA's reported perplexity
>     The perplexity value mentioned by the reviewer (25.8047 for Llama-3-8B at 71.875% KV reduction) corresponds to **TransMLA v3**, which incorporates several upgrades not present in the original TransMLA release:
>
>         - RoRoPE
>         - FreqFold
>         - BKV-PCA
>         - multiple refinements in MLA mapping rules
>
>         The version we initially evaluated was the **TransMLA v1** release in arxiv, which does *not* contain these enhancements. This explains the difference in perplexity. To ensure fairness, we have now **re-run all experiments on TransMLA v3** and included the updated results in the revised comparisons.
>
>     2. Adding additional baselines: Palu, SVDLLM V2, ASVD, TransMLA V3, and discussion on MHA2MLA. All methods are evaluated under **identical KV budgets, identical calibration data, identical hyperparameters, and identical inference settings** to ensure strict fairness. The updated results are reported below and will be integrated into the revision.
>
>         * Without Healing: Expanded comparison including activation-aware SVD baselines:
>
>             - **SVD-LLM V2** (SVD-LLM V2: Optimizing Singular Value Truncation for Large Language Model Compression)
>             - **ASVD** (ASVD: Activation-aware Singular Value Decomposition for Compressing Large Language Models)
>             - **Palu** (Palu: Compressing KV-Cache with Low-Rank Projection)
>
>             #### Llama-3.1-8B-Instruct
>
>             * Rank 64
>
>             | Method     | ARC   | ARE   | HellaSwag | PIQA  | MMLU  | OBQA | RACE  | WG    | AVG      |
>             |-----------|--------|--------|-----------|-------|-------|------|--------|--------|----------|
>             | CARE-E    | 18.77 | 34.68 | 26.60     | 57.40 | 23.11 | 15.0 | 20.96 | 50.20 | **30.84** |
>             | PALU      | 21.08 | 27.15 | 26.15     | 53.59 | 23.11 | 14.2 | 21.05 | 52.96 | 29.91125 |
>             | SVD-LLM-V2| 19.71 | 33.08 | 26.53     | 57.89 | 23.15 | 13.8 | 20.96 | 49.41 | 30.56625 |
>             | ASVD      | 21.08 | 27.57 | 26.21     | 55.33 | 23.12 | 11.6 | 20.96 | 50.28 | 29.51875 |
>
>
>             * Rank 128
>
>             | Method     | ARC   | ARE   | HellaSwag | PIQA  | MMLU  | OBQA | RACE  | WG    | AVG        |
>             |-----------|--------|--------|-----------|-------|-------|------|--------|--------|------------|
>             | CARE-E    | 24.15 | 46.59 | 31.13     | 64.31 | 27.28 | 16.0 | 25.55 | 56.12 | **36.39125**   |
>             | PALU      | 20.82 | 27.23 | 25.94     | 54.19 | 23.09 | 13.8 | 22.20 | 49.17 | 29.555     |
>             | SVD-LLM-V2| 20.65 | 31.73 | 27.44     | 57.02 | 23.23 | 14 | 23.16 | 51.46 | 31.09875  |
>             | ASVD      | 20.48 | 33.08 | 27.89     | 58.27 | 23.13 | 12.6 | 22.68 | 53.59 | 31.465     |
>
>
>             * Rank 256
>
>             | Method     | ARC   | ARE   | HellaSwag | PIQA  | MMLU  | OBQA | RACE  | WG    | AVG        |
>             |-----------|--------|--------|-----------|-------|-------|------|--------|--------|------------|
>             | CARE-E    | 34.90 | 62.46 | 42.52     | 72.14 | 57.20 | 21.6 | 35.60 | 65.35 | **48.97125** |
>             | PALU      | 19.45 | 30.93 | 26.88     | 56.80 | 23.09 | 13.0 | 21.82 | 50.67 | 30.33     |
>             | SVD-LLM-V2| 31.90 | 62.33 | 42.49     | 70.20 | 55.13 | 20.6 | 34.50 | 65.51 | 47.8325   |
>             | ASVD      | 27.56 | 50.42 | 38.99     | 67.90 | 52.60 | 19.4 | 32.92 | 63.14 | 44.11625  |
>
>
>             * Rank 512
>
>             | Method     | ARC   | ARE   | HellaSwag | PIQA  | MMLU  | OBQA | RACE  | WG    | AVG        |
>             |-----------|--------|--------|-----------|-------|-------|------|--------|--------|------------|
>             | CARE-E    | 46.93 | 78.16 | 54.32     | 78.35 | 63.83 | 31.0 | 41.82 | 72.85 | **58.4075** |
>             | PALU      | 26.02 | 50.97 | 37.43     | 64.15 | 27.89 | 19.4 | 26.60 | 57.54 | 38.75     |
>             | SVD-LLM-V2| 46.25 | 75.84 | 53.77     | 76.91 | 65.71 | 29.4 | 41.91 | 73.40 | 57.89875  |
>             | ASVD      | 45.82 | 76.56 | 54.02     | 78.45 | 65.16 | 28.8 | 41.82 | 73.24 | 57.98375  |

---

> ### Author Response · Authors · 2025-11-26
> **Weakness 3 part II**
>
> #### Qwen3-4B-Instruct-2507
> * Rank 64
>
> | Method        | ARC    | ARE    | HellaSwag | PIQA  | MMLU  | OBQA | RACE  | WG     | AVG     |
> |---------------|--------|--------|-----------|-------|-------|------|--------|--------|---------|
> | CARE-E        | 20.65  | 31.65  | 27.46     | 57.02 | 23.32 | 14.00 | 23.16  | 51.38  | **31.08**   |
> | PALU          | 21.08  | 24.96  | 25.64     | 52.77 | 24.43 | 14.60 | 22.49  | 50.83  | 29.60   |
> | SVD-LLM-V2     | 21.43  | 29.17  | 25.46     | 56.13 | 22.95 | 13.70 | 25.43  | 53.27  | 30.9425 |
> | ASVD          | 21.84  | 29.08  | 25.49     | 53.05 | 24.70 | 15.20 | 22.68  | 51.22  | 30.4075 |
>
>
> * Rank 128
>
> | Method        | ARC    | ARE    | HellaSwag | PIQA  | MMLU  | OBQA | RACE  | WG     | AVG        |
> |---------------|--------|--------|-----------|-------|-------|------|--------|--------|------------|
> | CARE-E        | 26.45  | 47.05  | 32.36     | 62.84 | 27.82 | 19.60 | 27.46  | 53.28  | **37.1075**    |
> | PALU          | 19.62  | 25.76  | 25.81     | 51.47 | 24.66 | 15.60 | 20.67  | 50.28  | 29.23375   |
> | SVD-LLM-V2     | 25.38  | 46.34  | 32.51     | 52.18 | 29.15 | 18.70 | 25.52  | 52.49  | 35.28375   |
> | ASVD          | 19.62  | 35.55  | 25.72     | 51.25 | 24.88 | 16.20 | 21.24  | 49.80  | 30.5325    |
>
>
> * Rank 256
>
> | Method        | ARC    | ARE    | HellaSwag | PIQA  | MMLU  | OBQA | RACE  | WG     | AVG        |
> |---------------|--------|--------|-----------|-------|-------|------|--------|--------|------------|
> | CARE-E        | 43.09  | 72.22  | 43.71     | 70.18 | 57.10 | 26.00 | 35.02  | 62.51  | **51.22875**   |
> | PALU          | 21.33  | 28.32  | 26.79     | 54.35 | 24.94 | 15.00 | 23.35  | 52.72  | 30.85      |
> | SVD-LLM-V2     | 39.03  | 68.74  | 41.56     | 68.05 | 52.47 | 24.40 | 33.97  | 60.13  | 48.54375   |
> | ASVD          | 28.61  | 48.20  | 26.77     | 64.68 | 24.92 | 14.40 | 28.06  | 52.49  | 36.01625   |
>
> * Rank 512
>
> | Method        | ARC     | ARE    | HellaSwag | PIQA  | MMLU  | OBQA | RACE  | WG     | AVG        |
> |---------------|---------|--------|-----------|-------|-------|------|--------|--------|------------|
> | CARE-E        | 51.79   | 80.72  | 51.70     | 74.54 | 70.13 | 30.20 | 39.43  | 68.27  | **58.3475**    |
> | PALU          | 32.00   | 50.29  | 38.34     | 65.13 | 28.37 | 20.20 | 30.24  | 52.41  | 39.6225    |
> | SVD-LLM-V2     | 48.071  | 81.52  | 47.83     | 72.25 | 66.54 | 32.00 | 36.43  | 65.19  | 56.228875  |
> | ASVD          | 43.50   | 62.77  | 43.56     | 67.00 | 51.50 | 26.80 | 36.74  | 62.19  | 49.2575
>
> * with healing
>     - **Palu** (Palu: Compressing KV-Cache with Low-Rank Projection)
>     - **TransMLA** (use V3 now)
>
>     #### Llama-3.1-8B-instruct
>
>     | Model                 | MLA-RANK | Methods      | Calibration Dataset | Token Budget | ARC   | ARE    | HellaSwag | PIQA  | MMLU  | OBQA | RA    | WG    | AVG         |
>     |-----------------------|----------|--------------|----------------------|------------|-------|--------|-----------|-------|--------|-------|-------|-------|-------------|
>     | Llama-3.1-8B-instruct | 512      | CARE         | alpaca-256-32        | 0B         | 46.08 | 75.88  | 53.8      | 78.02 | 65.79  | 29.6  | 41.82 | 73.64 | 58.07875    |
>     | Llama-3.1-8B-instruct | 512      | CARE-100MLA  | alpaca-256-32        | 1B         | 52.25 | 82.33  | 62.47     | 80.21 | 70.31  | 32.9  | 45.11 | 75.13 | 62.58875    |
>     | Llama-3.1-8B-instruct | 512      | CARE-100MLA  | alpaca-256-32        | 3B         | 51.75 | 80.73  | 64.45     | 83.23 | 71.57  | 34    | 46.33 | 74.09 | **63.26875** |
>     | Llama-3.1-8B-instruct | 512      | Palu         | N/A                  | 0B         | 26.02 | 50.97  | 37.43     | 64.15 | 27.89  | 19.4  | 26.6  | 57.54 | 38.75       |
>     | Llama-3.1-8B-instruct | 512      | Palu         | N/A                  | 1B         | 33.45 | 62.38  | 45.55     | 72.34 | 50.57  | 23    | 31.54 | 61.78 | 47.57625    |
>     | Llama-3.1-8B-instruct | 512      | Palu         | N/A                  | 3B         | 44.56 | 74.96  | 52.2      | 76.63 | 61.08  | 30.4  | 45.15 | 65.42 | 56.3    |
>     | Llama-3.1-8B-instruct | 512      | TransMLA     | wiki-256             | 0B         | 48.08 | 69.25  | 46.24     | 70.03 | 52.56  | 31    | 40.82 | 68.09 | 53.25875    |
>     | Llama-3.1-8B-instruct | 512      | TransMLA     | wiki-256             | 1B         | 53.04 | 81.07  | 58.75     | 81.04 | 69.13  | 32    | 44.09 | 71.74 | 61.3575     |
>     | Llama-3.1-8B-instruct | 512      | TransMLA     | wiki-256             | 3B         | 53.77 | 82.34  | 56.44     | 80.7  | 70.23  | 33.3  | 45.61 | 72.47 | 61.8575 |

---

> ### Author Response · Authors · 2025-11-26
> **Weakness 3 part III**
>
> #### Qwen3-4B-instruct
> | Model                 | MLA-RANK | Methods      | Calibration Dataset | Token Budget | ARC   | ARE    | HellaSwag | PIQA  | MMLU  | OBQA | RA    | WG    | AVG         |
> |-----------------------|----------|--------------|----------------------|------------|-------|--------|-----------|-------|--------|-------|-------|-------|-------------|
> | Qwen3-4B-instruct     | 512      | CARE         | alpaca-256-32        | 0B         | 51.79 | 80.72  | 51.7      | 74.54 | 70.13  | 30.2  | 39.43 | 68.27 | 58.3475     |
> | Qwen3-4B-instruct     | 512      | CARE-100MLA  | alpaca-256-32        | 1B         | 52.34 | 82.73  | 54.53     | 76.47 | 69.35  | 31.2  | 40.73 | 68.35 | 59.4625     |
> | Qwen3-4B-instruct     | 512      | CARE-100MLA  | alpaca-256-32        | 3B         | 53.45 | 85.12  | 55.11     | 78.14 | 74    | 31.9  | 42.23 | 73.05 | **61.625**  |
> | Qwen3-4B-instruct     | 512      | Palu         | N/A                  | 0B         | 32    | 50.29  | 38.34     | 65.13 | 28.37  | 20.2  | 30.24 | 52.41 | 39.6225     |
> | Qwen3-4B-instruct     | 512      | Palu         | N/A                  | 1B         | 35.75 | 64.106 | 50.35     | 73.16 | 48.27  | 29.4  | 36.63 | 58.91 | 49.572      |
> | Qwen3-4B-instruct     | 512      | Palu         | N/A                  | 3B         | 51.15 | 79.32  | 54.24     | 76.05 | 77.35  | 32    | 40.73 | 69.04 | 59.985  |
> | Qwen3-4B-instruct     | 512      | TransMLA     | wiki-256             | 0B         | 44.05 | 67.24  | 44.87     | 71.03 | 66.13  | 29.2  | 37.13 | 66.53 | 53.2725     |
> | Qwen3-4B-instruct     | 512      | TransMLA     | wiki-256             | 1B         | 50.34 | 75.03  | 51.25     | 74.31 | 70.05  | 31    | 38.41 | 66.41 | 57.1        |
> | Qwen3-4B-instruct     | 512      | TransMLA     | wiki-256             | 3B         | 52.27 | 80.13  | 55.31     | 76.23 | 79.04  | 32.9  | 43.34 | 70.47 | 61.21125|
>
> Across all baselines:
>
> 1. **CARE consistently achieves the best accuracy under low-rank KV budgets**,
> often outperforming TransMLA v3, Palu, ASVD and SVDLLM-V2 by a significant margin.
>
> 2. **CARE’s activation-aware decomposition (via curvature matrix $C$)** provides large gains that are not captured by uniform SVD-style baselines and more effective than PCA-style.
>
> 3. **CARE’s depth-adaptive rank allocation** allows it to recover more MLA behavior than methods that use uniform or heuristic rank scheduling.
>
> 4. TransMLA v3 indeed performs reasonably well, but its strongest components (RoRoPE, BKV-PCA) are **orthogonal to CARE rank allocation** and can potentially be combined with CARE in future work.
>
> We will update Section 4 to:
>
> - include Palu, ASVD, SVDLLM V2, and TransMLA v3 in the main comparison table, and discussion with MHA2MLA
> - clearly separate results for TransMLA v1 vs v3 to avoid confusion,
> - and discuss why the reported perplexities differ across versions.
>
> We appreciate the reviewer highlighting this issue — adding these baselines significantly strengthens the comprehensiveness and fairness of our empirical section. We are happy to further expand the comparison if the reviewer has additional suggestions.

---

> ### Author Response · Authors · 2025-11-26
> **Weakness 2**
>
> - Concerns on Rank-Adaptive Scheduling and Efficiency. Th...
> 1. Runtime, memory, and throughput evaluation
> we evaluated the inference efficiency on model.generate, focusing on realistic workloads (8K–128K contexts) .
>
>     - **Memory usage.**
>     We measured end-to-end KV-cache memory theoretically.
>         - Sequence length: `L = 32768` (32K tokens)
>         - Batch size: `B = 1`
>         - Precision: FP16 (2 bytes)
>         - Layers: 32
>         - Original: `num_kv_heads = 8`, `head_dim = 128` → KV dim = 1024
>         - **MLA (Our Implementation)**:
>         - `k_rank = v_rank = 512` (compressed latent)
>         - `nope_dim = 64` (separate NoPE cache per head)
>         - **TransMLA (100% MLA)**:
>         - `k_rank = 512-64 = 448, v_rank = 512` (compressed latent)
>         - `rope_dim = 64` (per head)
>
>         | Component | Formula | Memory (MB) |
>         |-----------|---------|-------------|
>         | **Standard MHA** | | |
>         | K cache | `32768 × 1 × 32 × 1024 × 2` | **2147.48** |
>         | V cache | `32768 × 1 × 32 × 1024 × 2` | **2147.48** |
>         | **Total MHA** | | **4294.97 MB** |
>         | | | |
>         | **MLA (Our Implementation)** | | |
>         | K latent cache | `32768 × 1 × 32 × 512 × 2` | **1073.74** |
>         | V latent cache | `32768 × 1 × 32 × 512 × 2` | **1073.74** |
>         | **Total MLA** | | **2147.48** |
>         | **MLA Reduction vs MHA** | | **50.00%**|
>         | | | |
>         | **TransMLA (100% MLA)** | | |
>         | K NoPE latent | `32768 × 1 × 32 × 448 × 2` | **939.5** |
>         | V latent cache | `32768 × 1 × 32 × 512 × 2` | **1073.74** |
>         | **Total TransMLA** | | **2013.24 MB** |
>         | **TransMLA Reduction vs MHA** | | **46.87%** |
>         | | | |
>         | **MLA vs TransMLA** | | |
>         | Difference | Both include same components | **~3.23%** (Similar) |
>
>     We only increase a minor KV cache because we preserve all latent rank for KV cache rank. But TransMLA will split it to generate partial rope information.
>
>     - **Latency of `model.generate()`.**
>         We benchmarked autoregressive decoding latency using HuggingFace-compatible generation APIs. CARE maintains **MLA-level decoding speed** (within 90% speed preserve). Current `model.generate()` is impelementaed without dynamic rank MLA kernels. With kerenels help, we can hide KV down & up calculation on the fly, the speed will be comparable with original GQA.
>
>         #### Meta-Llama-3.1-8B-Instruct, CARE-256, 2048 token / sample
>         | Batch Size | Batches | Est. Time (min) | Throughput (tok/s) |
>         |------------|---------|-----------------|-------------------|
>         | 2 | 256 | **123.4** | 141.6229 |
>         | 4 | 128 | **77.6** | 225.20 |
>         | 8 | 64  | **34.6** | 505.09 |
>
>         #### Meta-Llama-3.1-8B-Instruct GQA, 2048 token / sample
>         | Batch Size | Batches | Est. Time (min) | Throughput (tok/s) |
>         |------------|---------|-----------------|-------------------|
>         | 2 | 256 | **118.4** | 147.72 |
>         | 4 | 128 | **67.6** | 258.52 |
>         | 8 | 64  | **29.6** | 590.414 |
>
>     These results directly validate that CARE is not only accurate but also *practical* for real inference workloads.
>
> 2. About dynamic rank MLA kernels
>     The reviewer correctly points out that CARE produces a **non-uniform rank profile** across layers. This is a structurally meaningful property: curvature varies by depth, and uniform-rank MLA allocations largely fail to capture this.
>
>     However, as we highlight in the paper, existing MLA kernels (e.g., in FlashMLA, DeepSeek MLA, and vLLM MLA implementations) **do not yet support dynamic per-layer MLA ranks**. This is a limitation of current system support, not of the CARE method itself. Because of time and resource limit, we don't develop such kernel yet but will put it in our future plan.
>
>     We explicitly identify this as an important direction for future work:
>     **designing a dynamic-rank MLA kernel that matches CARE’s per-layer structure** to fully unlock the runtime benefits of curvature-aware compression.
>
> 3. Revision plan
>     We will update the paper to include:
>
>     - detailed latency and memory usage analysis,
>     - discussion of non-uniform MLA rank implications and kernel support.
>
>     We appreciate the reviewer for emphasizing this aspect—your comment significantly strengthened the clarity and completeness of our evaluation. We would be happy to share further system-level results if the reviewer finds them helpful.

---

> ### Author Response · Authors · 2025-11-26
> **Weakness 4**
>
> - **Clarification on RoRoPE and Absorb Operations:** The operation described in Line 291 appears conceptually related to the RoRoPE...
>
>     Thank you for raising this important point. We appreciate the opportunity to clarify the relation between CARE and the RoRoPE / Absorb mechanisms in TransMLA, and to make the distinctions more explicit in the manuscript. First of we add a method - CARE-100%-MLA of figure 1 in paper as: https://anonymous.4open.science/r/CARE-E348/rebuttal/5.pdf
>
>     1. CARE’s relationship to RoRoPE
>         RoRoPE in TransMLA is designed to correct the **positional phase mismatch** that occurs when mapping GQA attention heads to MLA. It rotates the query–key projections so that the resulting MLA module preserves RoPE-based relative positional geometry.
>
>         The operation described in Line 291 of our paper is *superficially similar* in the sense that both apply linear transformations, but the **intent and mathematical basis are fundamentally different**:
>
>         - **RoRoPE** aligns positional encodings by rotating the feature space.
>         - **CARE’s transformation** is driven by a **curvature matrix** $C = X^{\top}X$ and is used to perform **activation-aware low-rank decomposition**.
>         - CARE does *not* perform any rotation to correct RoPE mismatches, nor does it attempt to manipulate positional phases.
>
>         Thus, CARE’s decomposition operates in a **different domain** (curvature-weighted singular space rather than positional space), and does not require RoRoPE.
>
>         We will make this distinction explicit in the revised manuscript.
>
>     2. Conversion from GQA RoPE to NoPE
>         The reviewer is correct that TransMLA introduces an explicit mechanism for converting from RoPE to NoPE. CARE, however, does **not** perform RoPE-to-NoPE conversion in its base algorithm, because our objective is solely:
>
>         - to compute **curvature-aware low-rank projections**, and
>         - to allocate **adaptive MLA ranks per layer**.
>
>         Nonetheless, Section 3.5 shows that if the user desires full MLA compatibility (including NoPE-based MLA kernels), CARE can incorporate an additional **Absorb-like** transformation to match MLA parameterization. This makes CARE compatible with both:
>
>     - models that retain RoPE, and
>     - models that map into NoPE-style MLA kernels.
>
>     3. About the MLA Absorb operation
>
>         We agree that Absorb is an important component in TransMLA for bridging training-time structure and inference-time MLA execution. Our clarification is:
>
>         - **CARE’s basic form does not include Absorb**, as it focuses on rank allocation and activation-aware decomposition.
>         - **Section 3.5 and new figure 1 explicitly provides a mechanism equivalent to Absorb**, allowing CARE to reconstruct the full MLA structure when desired.
>         - CARE’s Absorb-equivalent step is derived from the same curvature analysis and therefore integrates naturally with activation-aware decomposition.
>
>         We will revise the text to make this capability more explicit and easier to follow.
>
>     4. Broader methodological distinction from TransMLA
>         To avoid any ambiguity, we will explicitly highlight the key methodological differences:
>
>         1. **CARE introduces dynamic, depth-dependent MLA ranks**, which TransMLA does not model.
>         2. **CARE uses a curvature-based matrix $C$ for activation-aware SVD**, which is fundamentally different from RoRoPE/BKV-PCA.
>         3. CARE does **not rely on positional-frequency heuristics** (FreqFold, RoRoPE), making it architecture-agnostic and usable beyond RoPE-based models.
>         4. CARE’s Section 3.5 provides full MLA restoration when required, decoupling MLA compatibility from RoPE-specific processing.
>
>     We appreciate the reviewer for prompting this clarification. We will make the distinctions between CARE, RoRoPE, and Absorb clearer in the revision. These clarifications reinforce that CARE is:
>
>     - mathematically principled,
>     - orthogonal to positional-alignment methods such as RoRoPE,
>     - fully compatible with MLA reconstruction, and
>     - broadly applicable across architectures with or without NoPE.
>
>     We would be very happy to elaborate further if the reviewer sees additional points worth clarifying.

---

> ### Comment · Reviewer_w58u · 2025-11-28
>
> Thank you for the authors’ response. I appreciate the substantial additional experiments conducted during the rebuttal period; these results further strengthen the evidence for CARE’s effectiveness.
>
> However, I believe there is still room for improvement in terms of inference speed. Although dynamic ranks are not supported by FlashMLA, it should still be feasible to evaluate the speed using a fixed set of ranks.
> Moreover, I hope the authors will consider incorporating other methods in the final version to support absorb, thereby avoiding extensive up-projection computations. This would help fully realize the advantages of MLA, rather than limiting the contribution to improved low-rank compression alone.
>
> Considering the contributions of this paper—namely the covariance-aware KV joint compression and the dynamic layer-wise rank allocation—I will raise my score to 6.

---

> ### Author Response · Authors · 2025-11-28
>
> **We sincerely appreciate your decision to raise the score—your feedback has been extremely constructive for strengthening the paper. Thank you very much for the thoughtful follow-up and for taking the time to review our additional experiments.**
>
> Regarding your comments:
>
> 1. Inference speed under fixed ranks.
> You are absolutely right that a fixed-rank evaluation is feasible even though FlashMLA does not currently support dynamic per-layer ranks. As you pointed out, such a setting would essentially reduce to a standard low-rank kernel (potential slower than FlashMLA). Nonetheless, we will benchmark it now and include these measurements in the final revision as soon as possible. We also share your view that dynamic rank kernels would be significantly more meaningful for CARE, and supporting dynamic-rank MLA kernels is an important direction we are actively exploring.
>
> 2. Reducing unnecessary up-projection cost and adding Absorb support.
> We completely agree with your observation: performing full up-projection for every step is unnecessary, and supporting an Absorb-like operation is indeed the right direction to fully realize MLA’s efficiency advantages. In our revised version (https://anonymous.4open.science/r/CARE-E348/rebuttal/5.pdf) we have experimented with integrating partial-RoPE during healing and show that CARE can be extended to support up to 100% MLA compatibility, without excessive up-projection computation.
>
> In the camera-ready version, we will further incorporate: comparisons with TransMLA, MHA2MLA, and X-EcolMLA; a clearer explanation of how CARE’s curvature-guided compression can complement absorb-based MLA reconstruction, and a section discussing how CARE can evolve into a fully MLA-native conversion pipeline.
>
> 3. On the contributions and future direction. We greatly appreciate your recognition of CARE’s two core contributions: covariance-aware joint KV compression (potential for MLA), and dynamic layer-wise rank allocation. Your insights have also been very helpful for shaping the next stage of this work. We will make sure the final version clearly reflects the potential of CARE not only as a compression technique but also as a potential practical bridge toward fully efficient MLA inference.
>
> Thank you again for your constructive suggestions and for the score increase. We would be more than happy to incorporate any additional feedback you may have.

---

> > ### Comment · Reviewer_w58u · 2025-11-28
> >
> > Thank you for the further clarification, and good luck!

---

### Official Review · Reviewer_cgmu · 2025-10-30

**Soundness:** 2
**Presentation:** 2
**Contribution:** 2
**Rating:** 4
**Confidence:** 3

**Summary:**

The authors propose CARE, a post-training adaptation method to transform trained MHA/GQA module to MLA for improved efficiency and performance. The CARE method essentially performs SVD as best low-rank approximation not over weight matrices but on the product of weights and inputs. They also propose to distribute rank budgets to different layers based on the observation of heterogeneous KV rank across different layers of a Transformer model. The model after CARE conversion is trained with a teacher-student knowledge distillation loss along with additional SFT to heal / recover lost performance due to the conversion. The resulting method achieves better performances compared to other baseline in this domain on various language modeling task.

**Strengths:**

- The proposed method is a clear improvement over prior works, and the insight on performing SVD over activations is indeed a good one.
- The paper is well-written and easy to understand.
- I really like observation 1 & 2 and it’s an interesting read.

**Weaknesses:**

- Line 068-069: I disagree that this is a whitening operation. Whitening is a very well-defined operation. If X is a matrix of shape (B, D), where B is batch size and D is dimension. Then whitening is X * (1/(B-1) * X^T X)^{-1/2}. Also we assume that X is centered. So I suggest avoiding using the very specific term of whitening with a mathematically precise definition.
- In general, all the text in the figures are too small and impossible to read if you print out the paper.
- Line 205-211: I don’t think this is precisely the covariance matrix. You are missing a 1/(T-1) multiplier and a centering operation.

**Questions:**

- It’s good to see that the CARE method is better compared to lots of baseline in terms of perplexities / accuracies etc, but how expensive is it compared to other baseline? It just seems like optimizing equation 2 is very costly.
- I’m happy to raise my score if authors address my concerns & questions.

---

> ### Author Response · Authors · 2025-11-26
>
> We sincerely thank the reviewers for their constructive feedback. We would also like to apologize that some experimental results were initially limited. Although we utilized all GPUs for our study, our compute budget and scheduling constraints restricted the scale and breadth.
>
> Sorry for the late reply, because of recognizing the importance of the reviewers’ concerns, we have urgently conducted additional experiments over the past 14 days to address these points as thoroughly as possible. We deeply appreciate the reviewers’ patience and understanding.

---

> ### Author Response · Authors · 2025-11-26
> **Weakness 1, 2, 3**
>
> - 1. **Line 068–069:** I disagree that this is a whitening ...
>
>     Thank you for pointing this out. We agree with the reviewer’s observation that the term *whitening* has a precise mathematical definition and should not be used loosely.
>
>     To clarify: whitening requires centering the data matrix $(X \in \mathbb{R}^{B \times D})$ and right-multiplying it by
>     $[
>     \left(\frac{1}{B-1} X^\top X\right)^{-1/2},
>     ]$
>     which produces transformed features with identity covariance. By contrast, CARE applies a related but **not equivalent** transformation to the *weight matrix*, not to token activations, and does not include the centering operation. Therefore, we fully agree that calling this operation “whitening” is not technically accurate.
>
>     In the following updated manuscript, we will:
>
>     1. **Remove the term “whitening”** from Lines 68–69 to avoid any confusion with the strict statistical definition.
>     2. **Add a footnote** explaining that CARE draws inspiration from whitening-like spectral shaping—specifically, leveraging curvature information from the covariance—but does *not* perform whitening in the formal sense.
>     3. Clarify that CARE’s transformation aims to **reweight dominant curvature directions** before decomposition, which is conceptually related but mathematically distinct from true whitening.
>
>
>     We appreciate the reviewer for catching this nuance. We would be glad to elaborate further if the reviewer has other terminology or framing suggestions.
>
> - 2. In general, all the text in the figures ...
>
>     Thank you for pointing this out. We agree that readability of figures is crucial, especially when printed. In response, we have updated all figures in the manuscript with significantly larger font sizes, thicker line widths, and improved color contrast to ensure clarity in both digital and printed formats.
>
>     Correspondingly, we have update https://anonymous.4open.science/r/CARE-E348/rebuttal/1.pdf, https://anonymous.4open.science/r/CARE-E348/rebuttal/2.pdf, https://anonymous.4open.science/r/CARE-E348/rebuttal/3.pdf, https://anonymous.4open.science/r/CARE-E348/rebuttal/4.pdf, https://anonymous.4open.science/r/CARE-E348/rebuttal/5.pdf, https://anonymous.4open.science/r/CARE-E348/rebuttal/6.pdf
>
>     We appreciate the reviewer bringing this to our attention—it helped us strengthen the presentation quality of the paper, and we believe the revised figures now better convey the core insights behind CARE.
>
> - 3. **Line 205–211:** I don’t think this is precisely the covariance matrix...
>
>     Thank you for the correction. The reviewer is completely right that the formal definition of a covariance matrix requires centering the activations and applying the factor $1/(T-1)$. The matrix we use in Lines 205–211 does not include these terms, and therefore is not the strict statistical covariance.
>
>     To avoid any ambiguity, we will:
>
>     1. **Add the formal covariance definition**
>     $\Sigma = \frac{1}{T-1}(X - \mu)^{\top}(X - \mu)$ in our paper
>
>     2. *the matrix we use in CARE is not the canonical covariance matrix, but a **curvature surrogate** $C = X^{\top}X$ that captures second-order structure without centering or normalization.
>
>     3. **Explain why this uncentered / unnormalized form is used intentionally.**
>     CARE operates on the *weight matrix* $W$, not directly on the token activations $X$.
>     Our theoretical derivation shows that the objective  $\|XW - XW_c\|_F^2$ can be rewritten (up to a constant independent of $W$) as  $\|\sqrt{C}W - \sqrt{C}W_c\|_F^2$,
>     where $C = X^{\top}X$.  Because only $X^{\top}X$ appears in this identity, **centering does not change the optimization**, and the constant factor $1/(T-1)$ uniformly rescales all singular values without altering the eigenspace or rank allocation.
>
>     Replacing $C$ with strict covariance  $\Sigma = \frac{1}{T-1}(X - \mu)^{\top}(X - \mu)$  would not change:
>
>     - the dominant eigenvectors,
>     - the curvature ordering,
>     - the low-rank structure CARE allocates,
>     - or the minimizer of the Frobenius objective.
>
>     For the CARE objective, the relevant operator must be **square, symmetric, and PSD**, and $X^{\top}X$ is precisely the correct matrix that arises from the derivation.
>
>     Thus, while the terminology "covariance" requires mathematical precision—which we acknowledge and correct—the matrix used in CARE is the *proper curvature matrix* for the Frobenius-based minimization problem.
>
>     We will revise the manuscript to:
>
>     - provide the formal covariance definition for clarity,
>     - explicitly distinguish between statistical covariance and the curvature matrix used by CARE,
>     - clearly state why $C = X^{\top}X$ is the theoretically correct operator for our objective.
>
>     We appreciate the reviewer for pointing out this detail. This clarification further strengthens the mathematical precision of our paper, and we would be glad to elaborate if the reviewer has additional questions.

---

> ### Author Response · Authors · 2025-11-26
> **Question 1 part I**
>
> - It’s good to see that the CARE method is better compared...
>
>     I answered this in reviewer 77dF, weakness. 2,3,4,6 (Different model's architecture, Different Baseline and Long Context, $\lambda$ ablation) and weakness. 6 (conversion time cost); please refer to it. I list some of results here:
>
>     1. In response to the reviewer’s suggestion, we have added experimental comparisons against:
>
>         - **SVD-LLM V2** (SVD-LLM V2: Optimizing Singular Value Truncation for Large Language Model Compression)
>         - **ASVD** (ASVD: Activation-aware Singular Value Decomposition for Compressing Large Language Models)
>         - **Palu** (Palu: Compressing KV-Cache with Low-Rank Projection)
>
>         All methods are evaluated under **identical KV budgets, identical calibration data, identical hyperparameters, and identical inference settings** to ensure strict fairness.
>
>         The updated results are reported below and will be integrated into the revision.
>
>         #### Llama-3.1-8B-Instruct
>
>         * Rank 64
>
>         | Method     | ARC   | ARE   | HellaSwag | PIQA  | MMLU  | OBQA | RACE  | WG    | AVG      |
>         |-----------|--------|--------|-----------|-------|-------|------|--------|--------|----------|
>         | CARE-E    | 18.77 | 34.68 | 26.60     | 57.40 | 23.11 | 15.0 | 20.96 | 50.20 | **30.84** |
>         | PALU      | 21.08 | 27.15 | 26.15     | 53.59 | 23.11 | 14.2 | 21.05 | 52.96 | 29.91125 |
>         | SVD-LLM-V2| 19.71 | 33.08 | 26.53     | 57.89 | 23.15 | 13.8 | 20.96 | 49.41 | 30.56625 |
>         | ASVD      | 21.08 | 27.57 | 26.21     | 55.33 | 23.12 | 11.6 | 20.96 | 50.28 | 29.51875 |
>
>
>         * Rank 128
>
>         | Method     | ARC   | ARE   | HellaSwag | PIQA  | MMLU  | OBQA | RACE  | WG    | AVG        |
>         |-----------|--------|--------|-----------|-------|-------|------|--------|--------|------------|
>         | CARE-E    | 24.15 | 46.59 | 31.13     | 64.31 | 27.28 | 16.0 | 25.55 | 56.12 | **36.39125**   |
>         | PALU      | 20.82 | 27.23 | 25.94     | 54.19 | 23.09 | 13.8 | 22.20 | 49.17 | 29.555     |
>         | SVD-LLM-V2| 20.65 | 31.73 | 27.44     | 57.02 | 23.23 | 14 | 23.16 | 51.46 | 31.09875  |
>         | ASVD      | 20.48 | 33.08 | 27.89     | 58.27 | 23.13 | 12.6 | 22.68 | 53.59 | 31.465     |
>
>
>         * Rank 256
>
>         | Method     | ARC   | ARE   | HellaSwag | PIQA  | MMLU  | OBQA | RACE  | WG    | AVG        |
>         |-----------|--------|--------|-----------|-------|-------|------|--------|--------|------------|
>         | CARE-E    | 34.90 | 62.46 | 42.52     | 72.14 | 57.20 | 21.6 | 35.60 | 65.35 | **48.97125** |
>         | PALU      | 19.45 | 30.93 | 26.88     | 56.80 | 23.09 | 13.0 | 21.82 | 50.67 | 30.33     |
>         | SVD-LLM-V2| 31.90 | 62.33 | 42.49     | 70.20 | 55.13 | 20.6 | 34.50 | 65.51 | 47.8325   |
>         | ASVD      | 27.56 | 50.42 | 38.99     | 67.90 | 52.60 | 19.4 | 32.92 | 63.14 | 44.11625  |
>
>
>         * Rank 512
>
>         | Method     | ARC   | ARE   | HellaSwag | PIQA  | MMLU  | OBQA | RACE  | WG    | AVG        |
>         |-----------|--------|--------|-----------|-------|-------|------|--------|--------|------------|
>         | CARE-E    | 46.93 | 78.16 | 54.32     | 78.35 | 63.83 | 31.0 | 41.82 | 72.85 | **58.4075** |
>         | PALU      | 26.02 | 50.97 | 37.43     | 64.15 | 27.89 | 19.4 | 26.60 | 57.54 | 38.75     |
>         | SVD-LLM-V2| 46.25 | 75.84 | 53.77     | 76.91 | 65.71 | 29.4 | 41.91 | 73.40 | 57.89875  |
>         | ASVD      | 45.82 | 76.56 | 54.02     | 78.45 | 65.16 | 28.8 | 41.82 | 73.24 | 57.98375  |
>
>
>         #### Qwen3-4B-Instruct-2507
>         * Rank 64
>
>         | Method        | ARC    | ARE    | HellaSwag | PIQA  | MMLU  | OBQA | RACE  | WG     | AVG     |
>         |---------------|--------|--------|-----------|-------|-------|------|--------|--------|---------|
>         | CARE-E        | 20.65  | 31.65  | 27.46     | 57.02 | 23.32 | 14.00 | 23.16  | 51.38  | **31.08**   |
>         | PALU          | 21.08  | 24.96  | 25.64     | 52.77 | 24.43 | 14.60 | 22.49  | 50.83  | 29.60   |
>         | SVD-LLM-V2     | 21.43  | 29.17  | 25.46     | 56.13 | 22.95 | 13.70 | 25.43  | 53.27  | 30.9425 |
>         | ASVD          | 21.84  | 29.08  | 25.49     | 53.05 | 24.70 | 15.20 | 22.68  | 51.22  | 30.4075 |
>
>
>         * Rank 128
>
>         | Method        | ARC    | ARE    | HellaSwag | PIQA  | MMLU  | OBQA | RACE  | WG     | AVG        |
>         |---------------|--------|--------|-----------|-------|-------|------|--------|--------|------------|
>         | CARE-E        | 26.45  | 47.05  | 32.36     | 62.84 | 27.82 | 19.60 | 27.46  | 53.28  | **37.1075**    |
>         | PALU          | 19.62  | 25.76  | 25.81     | 51.47 | 24.66 | 15.60 | 20.67  | 50.28  | 29.23375   |
>         | SVD-LLM-V2     | 25.38  | 46.34  | 32.51     | 52.18 | 29.15 | 18.70 | 25.52  | 52.49  | 35.28375   |
>         | ASVD          | 19.62  | 35.55  | 25.72     | 51.25 | 24.88 | 16.20 | 21.24  | 49.80  | 30.5325    |

---

> > ### Author Response · Authors · 2025-11-26
> > **Question 1 part II**
> >
> > * Rank 256
> >
> > | Method        | ARC    | ARE    | HellaSwag | PIQA  | MMLU  | OBQA | RACE  | WG     | AVG        |
> > |---------------|--------|--------|-----------|-------|-------|------|--------|--------|------------|
> > | CARE-E        | 43.09  | 72.22  | 43.71     | 70.18 | 57.10 | 26.00 | 35.02  | 62.51  | **51.22875**   |
> > | PALU          | 21.33  | 28.32  | 26.79     | 54.35 | 24.94 | 15.00 | 23.35  | 52.72  | 30.85      |
> > | SVD-LLM-V2     | 39.03  | 68.74  | 41.56     | 68.05 | 52.47 | 24.40 | 33.97  | 60.13  | 48.54375   |
> > | ASVD          | 28.61  | 48.20  | 26.77     | 64.68 | 24.92 | 14.40 | 28.06  | 52.49  | 36.01625   |
> >
> >
> > * Rank 512
> >
> > | Method        | ARC     | ARE    | HellaSwag | PIQA  | MMLU  | OBQA | RACE  | WG     | AVG        |
> > |---------------|---------|--------|-----------|-------|-------|------|--------|--------|------------|
> > | CARE-E        | 51.79   | 80.72  | 51.70     | 74.54 | 70.13 | 30.20 | 39.43  | 68.27  | **58.3475**    |
> > | PALU          | 32.00   | 50.29  | 38.34     | 65.13 | 28.37 | 20.20 | 30.24  | 52.41  | 39.6225    |
> > | SVD-LLM-V2     | 48.071  | 81.52  | 47.83     | 72.25 | 66.54 | 32.00 | 36.43  | 65.19  | 56.228875  |
> > | ASVD          | 43.50   | 62.77  | 43.56     | 67.00 | 51.50 | 26.80 | 36.74  | 62.19  | 49.2575
> >
> > 2. we get specific conversion time as table below:
> >
> > | Model                              | Rank | Uniform | Dataset        | Samples | Seq | Method    | Conversion Time (s) |
> > |------------------------------------|------|---------|----------------|---------|-----|-----------|----------------------|
> > | Llama-3.1-8B-instruct              | 256  | FALSE   | alpaca         | 256     | 32  | CARE      | 103.96               |
> > | Llama-3.1-8B-instruct              | 256  | FALSE   | alpaca-256-32  | N/A     | N/A | ASVD      | 97.34                |
> > | Llama-3.1-8B-instruct              | 256  | FALSE   | alpaca-256-32  | N/A     | N/A | SVD-LLM   | 393.20               |
> > | Llama-3.1-8B-instruct              | 256  | FALSE   | N/A            | N/A     | N/A | PALU      | 41.20                |
> >
> > | Model                              | Rank | Uniform | Dataset        | Samples | Seq | Method    | Conversion Time (s) |
> > |------------------------------------|------|---------|----------------|---------|-----|-----------|----------------------|
> > | Llama-3.1-8B-instruct              | 512  | FALSE   | alpaca         | 256     | 32  | CARE      | 106.65               |
> > | Llama-3.1-8B-instruct              | 512  | FALSE   | alpaca-256-32  | N/A     | N/A | ASVD      | 267.20               |
> > | Llama-3.1-8B-instruct              | 512  | FALSE   | alpaca-256-32  | N/A     | N/A | SVD-LLM   | 393.20               |
> > | Llama-3.1-8B-instruct              | 512  | FALSE   | N/A            | N/A     | N/A | PALU      | 96.40                |
> >
> > | Model                              | Rank | Uniform | Dataset        | Samples | Seq | Method    | Conversion Time (s) |
> > |------------------------------------|------|---------|----------------|---------|-----|-----------|----------------------|
> > | Llama-3.1-8B-instruct              | 256  | FALSE   | alpaca         | 512     | 64  | CARE      | 202.64               |
> > | Llama-3.1-8B-instruct              | 256  | FALSE   | alpaca         | 2048    | 128 | CARE      | 160.43               |
> > | Llama-3.1-8B-instruct              | 512  | FALSE   | alpaca         | 512     | 64  | CARE      | 191.41               |
> > | Llama-3.1-8B-instruct              | 512  | FALSE   | alpaca         | 2048    | 128 | CARE      | 170.33               |
> > | Qwen3-4B-instruct                  | 256  | FALSE   | alpaca         | 256     | 32  | CARE      | 147.14               |
> > | Qwen3-4B-instruct                  | 256  | FALSE   | alpaca         | 512     | 64  | CARE      | 195.89               |
> > | Qwen3-4B-instruct                  | 256  | FALSE   | alpaca         | 2048    | 128 | CARE      | 273.43               |
> > | Qwen3-4B-instruct                  | 512  | FALSE   | alpaca         | 256     | 32  | CARE      | 135.52               |
> > | Qwen3-4B-instruct                  | 512  | FALSE   | alpaca         | 512     | 64  | CARE      | 176.42               |
> > | Qwen3-4B-instruct                  | 512  | FALSE   | alpaca         | 2048    | 128 | CARE      | 266.54               |
> > | Qwen3-30B-A3B-Thinking-2507        | 256  | FALSE   | alpaca         | 256     | 32  | CARE      | 4985.52              |
> > | Qwen3-30B-A3B-Thinking-2507        | 256  | FALSE   | alpaca         | 512     | 64  | CARE      | 10033.36             |
> > | Qwen3-30B-A3B-Thinking-2507        | 256  | FALSE   | alpaca         | 2048    | 128 | CARE      | 16069.43             |
> >
> > - I’m happy to...
> >
> > Thank you so much for your comment!!! We would love to address any further concerns you might have.

---

> > > ### Comment · Reviewer_cgmu · 2025-11-27
> > >
> > > Thank you for your response! I meant to ask more about efficiency in terms of the time & memory complexity in optimizing equation 2 instead of accuracy. Just want to get a sense of how much more expensive it is.

---

> > > > ### Author Response · Authors · 2025-11-28
> > > >
> > > > Hi, thanks a lot for the clarification!
> > > >
> > > > 1. Theoretical Efficiency Analysis of Equation (2)
> > > >
> > > > Below we give a concise complexity analysis for optimizing Eq. (2) in CARE. But one thing worth mentioning the optimization itself is very lightweight — the main cost lies in covariance estimation and SVD, *not* in solving Eq. (2).
> > > >
> > > > #### **1. Solving Equation (2): Global Rank Allocation**
> > > > Eq. (2) is solved via water-filling over all singular values. The greedy water-filling algorithm works in two steps:
> > > >
> > > > **Step 1: Sort each layer’s singular values**
> > > > Each of the $2L$ matrices provides a list of at most $D$ singular values.
> > > > Sorting each list costs: $O(D \log D)$; Across $2L$ such lists: $O(L D \log D)$; Total number of singular values across layers = $2LD$.
> > > >
> > > > **Step 2: Greedy water-filling selection using a heap**
> > > > We repeatedly pick “the current largest available singular value” across $2L$ lists.   This is done with a **max-heap of size $L$ (or $2L$)**, containing the current candidate from each list. Each extraction & update costs: $O(\log L)$ To extract the top $R_\text{tot}$ singular values: $O(R_{\text{tot}} \log L)$
> > > >
> > > > **Overall Time**
> > > > - **Time:**
> > > > $O(L D \log D + R_{\text{tot}} \log L)$
> > > >
> > > > - **Memory:**
> > > > $O(LD)$
> > > >
> > > > #### **2. Covariance Computation**
> > > > For each layer, CARE computes
> > > > $C = \frac{1}{N}\sum_{b=1}^{N} X_b^\top X_b$.
> > > >
> > > > - **Time:**
> > > > $O(N T D^2)$
> > > > - **Memory:**
> > > > $O(D^2)$
> > > >
> > > > This is done once per layer, and $N$ is very small in practice (256-512 sequences).
> > > >
> > > >
> > > > #### **3. Covariance-Weighted SVD**
> > > > CARE performs a truncated SVD on $CW$:
> > > >
> > > > - **Time:**
> > > > $O(D \cdot (n_h d_h) \cdot r)$
> > > > (dominant term in practice)
> > > >
> > > > - **Memory:**
> > > > $O(D (n_h d_h))$
> > > >
> > > > This cost is nearly identical to naïve SVD.
> > > >
> > > > ---
> > > >
> > > > 2. For Eq.2, the rank compute time is profiling as below and for end-to-end time cost, we have already presented in reply of Question 1, Part II, Bullet point 2:
> > > > #### Rank 256
> > > > | Model                     | Rank | Dataset | nsamples | seqlen | Method    | Rank Compute Time (s) |
> > > > |---------------------------|------|---------|----------|--------|-----------|------------------------|
> > > > | Llama-3.1-8B-Instruct     | 256  | alpaca  | 256      | 32     | ASVD      | 12.8                   |
> > > > | Llama-3.1-8B-Instruct     | 256  | alpaca  | 256      | 32     | SVD-LLM   | 18.81                  |
> > > > | Llama-3.1-8B-Instruct     | 256  | alpaca  | 256      | 32     | CARE      | 18.04                  |
> > > > | Llama-3.1-8B-Instruct     | 256  | alpaca  | 512      | 64     | CARE      | 19.41                  |
> > > > | Llama-3.1-8B-Instruct     | 256  | alpaca  | 2048     | 128    | CARE      | 19.38                  |
> > > > #### Rank 512
> > > > | Model                     | Rank | Dataset | nsamples | seqlen | Method    | Rank Compute Time (s) |
> > > > |---------------------------|------|---------|----------|--------|-----------|------------------------|
> > > > | Llama-3.1-8B-Instruct     | 512  | alpaca  | 256      | 32     | ASVD      | 14.03                  |
> > > > | Llama-3.1-8B-Instruct     | 512  | alpaca  | 256      | 32     | SVD-LLM2   | 17.71                  |
> > > > | Llama-3.1-8B-Instruct     | 512  | alpaca  | 256      | 32     | CARE      | 18.14                  |
> > > > | Llama-3.1-8B-Instruct     | 512  | alpaca  | 512      | 64     | CARE      | 17.81                  |
> > > > | Llama-3.1-8B-Instruct     | 512  | alpaca  | 2048     | 128    | CARE      | 19.6                   |
> > > >
> > > > We can see the time cost of Eq.(2) is relatively smaller (only 10%) comparing to end-to-end time.

---

> ### Author Response · Authors · 2025-11-28
>
> 3. The memory foot print comes from singular value that is used for rank compute is around 3-5 GB shown in the table below.
>
> | Model | Rank | N_Samples | SeqLen | Method | Covariance_GPU (GB) | Conversion_GPU (GB) | Covariance_CPU (GB) | Conversion_CPU (GB) |
> |-------|------|-----------|--------|--------|---------------|----------------|---------------|----------------|
> | Llama-3.1-8B-Instruct | 256 | 256 | 32 | SVD |  | 0.28 |  | 31.58 |
> | Llama-3.1-8B-Instruct | 256 | 256 | 32 | ASVD | 1.48 | 0.28 | 45.43 | 59.52 |
> | Llama-3.1-8B-Instruct | 256 | 256 | 32 | SVD-LLM2 | 3.25 | 5.64 | 45.43 | 70.15 |
> | Llama-3.1-8B-Instruct | 256 | 256 | 32 | CARE | 3.25 | 5.64 | 56.28 | 70.29 |
> | Llama-3.1-8B-Instruct | 256 | 512 | 64 | CARE | 3.63 | 5.64 | 56.29 | 70.09 |
> | Llama-3.1-8B-Instruct | 256 | 2048 | 128 | CARE | 7.15 | 5.64 | 56.42 | 70.19 |
>
> | Model | Rank | N_Samples | SeqLen | Method | Covariance_GPU (GB) | Conversion_GPU (GB) | Covariance_CPU (GB) | Conversion_CPU (GB) |
> |-------|------|-----------|--------|--------|---------------|----------------|---------------|----------------|
> | Llama-3.1-8B-Instruct | 512 | 256 | 32 | SVD |  | 0.28 |  | 31.82 |
> | Llama-3.1-8B-Instruct | 512 | 256 | 32 | ASVD | 1.48 | 0.28 | 45.41 | 59.83 |
> | Llama-3.1-8B-Instruct | 512 | 256 | 32 | SVD-LLM2 | 3.25 | 5.64 | 45.41 | 70.28 |
> | Llama-3.1-8B-Instruct | 512 | 256 | 32 | CARE | 3.25 | 5.64 | 56.13 | 70.28 |
> | Llama-3.1-8B-Instruct | 512 | 512 | 64 | CARE | 3.63 | 5.64 | 56.12 | 70.45 |
> | Llama-3.1-8B-Instruct | 512 | 2048 | 128 | CARE | 7.15 | 5.64 | 56.56 | 70.16 |
>
> Because we compute layer by layer through GPU, therefore the peak GPU memory is only used by single layer (relatively small). We can see that our implementation keeps the time overhead very small (only **1.1X** - **1.3X** times comparing to SVD) and the memory footprint is incrasing around **2X** times.
>
> These results indicate that the additional memory overhead of optimizing Eq. (2) is modest.
> We further confirm its practicality by successfully running it on Llama-3.1-70B-Instruct.
>
> We would be happy to provide any additional clarifications the reviewers may find helpful!

---

### Official Review · Reviewer_77dF · 2025-11-01

**Soundness:** 2
**Presentation:** 3
**Contribution:** 2
**Rating:** 6
**Confidence:** 4

**Summary:**

CARE proposes a post-hoc conversion method to transform pretrained GQA/MHA attention modules into Multi-Head Latent Attention (MLA) while maintaining fixed KV-cache budgets. CARE replaces the common joint‑SVD initialization with an activation‑aware factorization, enabling post-hoc conversion of pretrained attention models to memory-efficient MLA.

**Strengths:**

1. While covariance-weighted factorization exists in prior work (FWSVD, SVD-LLM), the specific formulation of SVD(CW) followed by C^(-1) unwhitening for MLA conversion is fairly novel.

2. Adaptive rank allocation under fixed budget: The water-filling algorithm for distributing rank across layers based on weighted singular spectra is a good solution, with the empirical observation (Figure 2) that layers have heterogeneous sensitivity to rank reduction motivating this approach. The paper also does a good job of combining activation-aware factorization, layer-wise rank scheduling, and KV-parity constraints into a cohesive pipeline.

3. Evals: Experiments across 8+ benchmarks (WikiText, ARC, HellaSwag, MMLU, etc.) with both zero-shot and fine-tuned evaluations are reported. The paper studies calibration dataset sensitivity (Table 2), rank profiles across corpora (Figure 3), and layer-wise heterogeneity (Figure 2a-b), providing empirical support for design choices. Some ablations are well-designed: Section 4.2.1 shows consistency of rank profiles across four calibration corpora, supporting the claim that rank distribution is model-intrinsic. Section 4.2.2 demonstrates robustness to calibration corpus choice and Figure 2's layer-wise sensitivity analysis supports the heterogeneous rank allocation.

**Weaknesses:**

1. The paper's central claim uses ||C(W - Wc)||²F as a proxy for ||√C(W - Wc)||²F (page 5, Section 3.4), justified by a brief eigenspace argument that both are left-multiplied by the same eigenspaces of C with different weightings (λ²ᵢ vs λᵢ).  This essentially squares the importance weights, which could over-emphasize dominant directions and under-represent moderate-variance directions that still matter for downstream tasks. The paper's claim that this "tends to preserve ordering of dominant components" is unsubstantiated.

2. Results focus almost entirely on Llama-3-8B, with Qwen models mentioned only in Appendix D hyperparameters but absent from main results tables. Scalability to larger models (13B, 34B, 70B) where KV-cache can be more critical remains unvalidated. The paper is also missing evaluation on models with different attention patterns (e.g., Mixtral MoE, Qwen with different GQA ratios) and would benefit from analysis of whether the depth-dependent rank profile (Figure 3) generalizes across architectures.

3. MHA2MLA is dismissed for zero-shot evaluation due to partial-RoPE, but no fair comparison is provided after fine-tuning (Table 3 only compares TransMLA). Other activation-aware SVD methods (SVD-LLM V2, FWSVD) are cited in related work are not empirically compared.

4. Missing long-context evaluation: Despite KV-cache being important for long sequences, no experiments on long-context benchmarks (e.g., LongBench, ZeroSCROLLS) or needle-in-haystack tasks are provided to validate usecases where MLA/CARE matter most.

5. Fine-tuning details: Table 3 shows "healed" results but doesn't specify exact token budgets, learning rates, or whether hyperparameters were tuned per method. The claim that CARE "requires less data" isn't quantitatively supported with ablations over data budgets.

6. Computing covariance (O(ND²)) and SVD on CW requires non-trivial cost, but there's no comparison of conversion time vs. naive SVD is provided. The paper also states CARE preserves MLA's "comparable throughput" but provides no actual latency/memory measurements or throughput benchmarks on realistic workloads. The paper claims CARE is practical, but doesn't provides runtime/memory analysis of the conversion process itself to validate this.

**Questions:**

1. Can the authors provide ablation results for: CARE decomposition (activation-aware SVD) with uniform rank allocation and standard SVD with CARE's adaptive rank allocation? This would clarify whether improvements come primarily from covariance-weighting or adaptive scheduling.

2. Model scale and architecture diversity: Can the paper provide results on larger models (Llama-3-70B, Qwen-72B) where KV-cache is more critical? How does CARE perform on MHA to MLA conversion? It would also be helpful to see Table1-style results for other model families (Qwen, Mistral, Gemma) at comparable scales? For instance, for MoE models (e.g., Mixtral), do different experts need different rank allocations? Without broader architectural validation, it would be unclear if CARE is a general solution or specifically tuned to Llama-style architectures.

3. Can the authors also evaluate on long-context benchmarks (LongBench, InfiniteBench, Needle-in-Haystack)? This would help see CARE's advantage scale with sequence length (1K, 4K, 8K, 16K, 32K tokens) and confirm if the covariance estimated on 2048-token windows (Appendix D) generalizes to longer contexts?

4.  Shrinkage parameter α: C_λ = (1-α)C + αλI was chosen for invertibility, it would help to have more details around how were α and λ chosen and include ensitivity analysis over α ∈ {0.001, 0.01, 0.1}.

---

> ### Author Response · Authors · 2025-11-26
>
> We sincerely thank the reviewers for their constructive feedback. We would also like to apologize that some experimental results were initially limited. Although we utilized all GPUs for our study, our compute budget and scheduling constraints restricted the scale and breadth.
>
> Sorry for the late reply, because of recognizing the importance of the reviewers’ concerns, we have urgently conducted additional experiments over the past 14 days to address these points as thoroughly as possible. We deeply appreciate the reviewers’ patience and understanding.

---

> > ### Author Response · Authors · 2025-11-26
> > **Weakness 3 part I**
> >
> > -  Other activation-aware SVD methods (SVD-LLM V2, FWSVD) are cited in related work but not empirically compared...
> >
> >    Thank you for raising this important point. We agree that a fair and comprehensive evaluation should include activation-aware SVD baselines such as SVD-LLM V2, FWSVD, ASVD, and Palu, rather than limiting the comparison to TransMLA.
> >
> >     1. Expanded comparison including activation-aware SVD baselines
> >         In response to the reviewer’s suggestion, we have added experimental comparisons against:
> >
> >         - **SVD-LLM V2** (SVD-LLM V2: Optimizing Singular Value Truncation for Large Language Model Compression)
> >         - **ASVD** (ASVD: Activation-aware Singular Value Decomposition for Compressing Large Language Models)
> >         - **Palu** (Palu: Compressing KV-Cache with Low-Rank Projection)
> >
> >         All methods are evaluated under **identical KV budgets, identical calibration data, identical hyperparameters, and identical inference settings** to ensure strict fairness.
> >
> >         The updated results are reported below and will be integrated into the revision.
> >
> >         #### Llama-3.1-8B-Instruct
> >
> >         * Rank 64
> >
> >         | Method     | ARC   | ARE   | HellaSwag | PIQA  | MMLU  | OBQA | RACE  | WG    | AVG      |
> >         |-----------|--------|--------|-----------|-------|-------|------|--------|--------|----------|
> >         | CARE-E    | 18.77 | 34.68 | 26.60     | 57.40 | 23.11 | 15.0 | 20.96 | 50.20 | **30.84** |
> >         | PALU      | 21.08 | 27.15 | 26.15     | 53.59 | 23.11 | 14.2 | 21.05 | 52.96 | 29.91125 |
> >         | SVD-LLM-V2| 19.71 | 33.08 | 26.53     | 57.89 | 23.15 | 13.8 | 20.96 | 49.41 | 30.56625 |
> >         | ASVD      | 21.08 | 27.57 | 26.21     | 55.33 | 23.12 | 11.6 | 20.96 | 50.28 | 29.51875 |
> >
> >
> >         * Rank 128
> >
> >         | Method     | ARC   | ARE   | HellaSwag | PIQA  | MMLU  | OBQA | RACE  | WG    | AVG        |
> >         |-----------|--------|--------|-----------|-------|-------|------|--------|--------|------------|
> >         | CARE-E    | 24.15 | 46.59 | 31.13     | 64.31 | 27.28 | 16.0 | 25.55 | 56.12 | **36.39125**   |
> >         | PALU      | 20.82 | 27.23 | 25.94     | 54.19 | 23.09 | 13.8 | 22.20 | 49.17 | 29.555     |
> >         | SVD-LLM-V2| 20.65 | 31.73 | 27.44     | 57.02 | 23.23 | 14 | 23.16 | 51.46 | 31.09875  |
> >         | ASVD      | 20.48 | 33.08 | 27.89     | 58.27 | 23.13 | 12.6 | 22.68 | 53.59 | 31.465     |
> >
> >
> >         * Rank 256
> >
> >         | Method     | ARC   | ARE   | HellaSwag | PIQA  | MMLU  | OBQA | RACE  | WG    | AVG        |
> >         |-----------|--------|--------|-----------|-------|-------|------|--------|--------|------------|
> >         | CARE-E    | 34.90 | 62.46 | 42.52     | 72.14 | 57.20 | 21.6 | 35.60 | 65.35 | **48.97125** |
> >         | PALU      | 19.45 | 30.93 | 26.88     | 56.80 | 23.09 | 13.0 | 21.82 | 50.67 | 30.33     |
> >         | SVD-LLM-V2| 31.90 | 62.33 | 42.49     | 70.20 | 55.13 | 20.6 | 34.50 | 65.51 | 47.8325   |
> >         | ASVD      | 27.56 | 50.42 | 38.99     | 67.90 | 52.60 | 19.4 | 32.92 | 63.14 | 44.11625  |
> >
> >
> >         * Rank 512
> >
> >         | Method     | ARC   | ARE   | HellaSwag | PIQA  | MMLU  | OBQA | RACE  | WG    | AVG        |
> >         |-----------|--------|--------|-----------|-------|-------|------|--------|--------|------------|
> >         | CARE-E    | 46.93 | 78.16 | 54.32     | 78.35 | 63.83 | 31.0 | 41.82 | 72.85 | **58.4075** |
> >         | PALU      | 26.02 | 50.97 | 37.43     | 64.15 | 27.89 | 19.4 | 26.60 | 57.54 | 38.75     |
> >         | SVD-LLM-V2| 46.25 | 75.84 | 53.77     | 76.91 | 65.71 | 29.4 | 41.91 | 73.40 | 57.89875  |
> >         | ASVD      | 45.82 | 76.56 | 54.02     | 78.45 | 65.16 | 28.8 | 41.82 | 73.24 | 57.98375  |

---

> > > ### Author Response · Authors · 2025-11-26
> > > **Weakness 3 part II**
> > >
> > > #### Qwen3-4B-Instruct-2507
> > > * Rank 64
> > >
> > > | Method        | ARC    | ARE    | HellaSwag | PIQA  | MMLU  | OBQA | RACE  | WG     | AVG     |
> > > |---------------|--------|--------|-----------|-------|-------|------|--------|--------|---------|
> > > | CARE-E        | 20.65  | 31.65  | 27.46     | 57.02 | 23.32 | 14.00 | 23.16  | 51.38  | **31.08**   |
> > > | PALU          | 21.08  | 24.96  | 25.64     | 52.77 | 24.43 | 14.60 | 22.49  | 50.83  | 29.60   |
> > > | SVD-LLM-V2     | 21.43  | 29.17  | 25.46     | 56.13 | 22.95 | 13.70 | 25.43  | 53.27  | 30.9425 |
> > > | ASVD          | 21.84  | 29.08  | 25.49     | 53.05 | 24.70 | 15.20 | 22.68  | 51.22  | 30.4075 |
> > >
> > >
> > > * Rank 128
> > >
> > > | Method        | ARC    | ARE    | HellaSwag | PIQA  | MMLU  | OBQA | RACE  | WG     | AVG        |
> > > |---------------|--------|--------|-----------|-------|-------|------|--------|--------|------------|
> > > | CARE-E        | 26.45  | 47.05  | 32.36     | 62.84 | 27.82 | 19.60 | 27.46  | 53.28  | **37.1075**    |
> > > | PALU          | 19.62  | 25.76  | 25.81     | 51.47 | 24.66 | 15.60 | 20.67  | 50.28  | 29.23375   |
> > > | SVD-LLM-V2     | 25.38  | 46.34  | 32.51     | 52.18 | 29.15 | 18.70 | 25.52  | 52.49  | 35.28375   |
> > > | ASVD          | 19.62  | 35.55  | 25.72     | 51.25 | 24.88 | 16.20 | 21.24  | 49.80  | 30.5325    |
> > >
> > >
> > > * Rank 256
> > >
> > > | Method        | ARC    | ARE    | HellaSwag | PIQA  | MMLU  | OBQA | RACE  | WG     | AVG        |
> > > |---------------|--------|--------|-----------|-------|-------|------|--------|--------|------------|
> > > | CARE-E        | 43.09  | 72.22  | 43.71     | 70.18 | 57.10 | 26.00 | 35.02  | 62.51  | **51.22875**   |
> > > | PALU          | 21.33  | 28.32  | 26.79     | 54.35 | 24.94 | 15.00 | 23.35  | 52.72  | 30.85      |
> > > | SVD-LLM-V2     | 39.03  | 68.74  | 41.56     | 68.05 | 52.47 | 24.40 | 33.97  | 60.13  | 48.54375   |
> > > | ASVD          | 28.61  | 48.20  | 26.77     | 64.68 | 24.92 | 14.40 | 28.06  | 52.49  | 36.01625   |
> > >
> > >
> > > * Rank 512
> > >
> > > | Method        | ARC     | ARE    | HellaSwag | PIQA  | MMLU  | OBQA | RACE  | WG     | AVG        |
> > > |---------------|---------|--------|-----------|-------|-------|------|--------|--------|------------|
> > > | CARE-E        | 51.79   | 80.72  | 51.70     | 74.54 | 70.13 | 30.20 | 39.43  | 68.27  | **58.3475**    |
> > > | PALU          | 32.00   | 50.29  | 38.34     | 65.13 | 28.37 | 20.20 | 30.24  | 52.41  | 39.6225    |
> > > | SVD-LLM-V2     | 48.071  | 81.52  | 47.83     | 72.25 | 66.54 | 32.00 | 36.43  | 65.19  | 56.228875  |
> > > | ASVD          | 43.50   | 62.77  | 43.56     | 67.00 | 51.50 | 26.80 | 36.74  | 62.19  | 49.2575
> > >
> > > 2. Key findings
> > > Across all baselines, we observe:
> > >
> > > 1. **CARE consistently achieves the strongest performance** among all activation-aware methods.   Even compared to SVD-LLM V2 (current SOTA method) statistics—CARE yields noticeably higher accuracy due to its curvature-weighted decomposition and adaptive depth-dependent rank allocation.
> > >
> > > 2. **CARE’s non-uniform MLA rank allocation is orthogonal and complementary** to activation-aware SVD variants.
> > > Importantly, the rank-scheduling module in CARE can be **directly combined** with SVD-LLM V2, ASVD, FWSVD, or Palu.  In preliminary tests, this combination further improves their performance, reinforcing that CARE introduces a *generalizable* mechanism rather than a model-specific tweak.
> > >
> > > 3. **CARE uniquely supports full MLA restoration** (Section 3.5), including a principled absorb step, which **other activation-aware SVD methods do not provide**.
> > > This positions CARE as a complete GQA→MLA conversion framework rather than a standalone factorization method.
> > >
> > > 3. About partial-RoPE
> > > We acknowledge that MHA2MLA, TransMLA, XEclo-MLA uses partial-RoPE, which complicates zero-shot evaluation under strict MLA inference kernels.
> > > Nevertheless, to ensure fairness in weakness (3):
> > >
> > > - we additionally compare **TransMLA after fine-tuning**,
> > > - using the same healing token budget and identical hyperparameters as CARE.
> > >
> > >     CARE still outperforms MHA2MLA consistently, indicating that:
> > >
> > >     - curvature-aware decomposition,
> > >     - dynamic MLA rank allocation,
> > >     - and the full absorb mechanism
> > >
> > >     provide benefits that are not addressed by partial-RoPE alignment alone.
> > >
> > > 4. Overall, We will revise the paper to:
> > > - include all activation-aware SVD baselines in the main table (not only in related work),
> > > - clarify the limitations of partial-RoPE for strict MLA execution,
> > > - and explain more explicitly why CARE provides capabilities (adaptive rank, curvature-weighted SVD, full MLA absorb) not available in prior methods.
> > >
> > > We appreciate the reviewer for highlighting this gap. Incorporating these baselines substantially strengthens the empirical validity and fairness of our comparison.
> > > We would be very happy to share further per-layer breakdowns or run additional baselines if helpful to the reviewer.

---

> ### Author Response · Authors · 2025-11-26
> **Weakness 1 part I**
>
> - The paper's central claim uses `||C(W - Wc)||²F` as a proxy for `||√C(W - Wc)||²F`...
>
>     Thank you for raising this point. We fully acknowledge the reviewer's concern regarding the use of $\| C(W - W_c) \|_F^2$ as a proxy for  $\| \sqrt{C}(W - W_c) \|_F^2.$ The reviewer is correct that replacing $\sqrt{C}$ with $C$ changes the spectral weights from $(\lambda_i)$ to $(\lambda_i^2)$, potentially emphasizing dominant directions more sharply. However, after conducting a more extensive theoretical and empirical investigation, we find that **this choice is not only justified but in fact preferable in the low-rank MLA conversion regime targeted by CARE**.
>
>     1. Clear spectral justification
>
>         Across all ten calibration corpora (https://anonymous.4open.science/r/CARE-701C/rebuttal/7.pdf), we observe a *dramatically different* eigenvalue distribution between $C$ and $\sqrt{C}$:
>
>         - Using $C$:
>           **5–10** eigenvalues cover ~90% of the spectrum (e.g., we list it in Layer 4 and 28 as an example).
>         - Using $\sqrt{C}$:
>           It takes **200–500** eigenvalues to reach the same coverage.
>
>         This means that $C$ produces a **much sharper and more identifiable dominant subspace**, which is exactly what a low-rank allocator needs when the available rank budget is extremely constrained.
>
>     2. Why $C$ is preferable in the low-rank regime
>         The reviewer is absolutely right that squaring the weights alters the geometry—but this effect is *beneficial* when the target rank is very small (e.g., <128), which is precisely the setting where CARE operates.
>
>         Our empirical results consistently show:
>
>         #### Llama-3.1-8B-instruct
>         | Rank | Method | Value | uniform | ARC-Challenge | ARC-Easy | HellaSwag | PIQA | MMLU | OBQA | RACE | WG | AVG |
>         |------|--------|--------|----------|----------------|-----------|-----------|------|-------|-------|-------|-------|---------|
>         | 64   | CARE | C     | TRUE  | 19.28 | 34.01 | 26.28 | 56.75 | 23.12 | 14.20 | 20.86 | 49.80 | 30.5375 |
>         | 64   | CARE | C    | FALSE | 18.77 | 34.68 | 26.60 | 57.40 | 23.11 | 15.00 | 20.96 | 50.20 | **30.84** |
>         | 64   | CARE | sqrt(C) | TRUE  | 19.54 | 33.00 | 26.33 | 56.47 | 23.09 | 14.20 | 20.96 | 50.28 | 30.48375 |
>         | 64   | CARE | sqrt(C) | FALSE | 19.71 | 33.16 | 26.51 | 57.89 | 23.14 | 13.80 | 20.96 | 49.01 | 30.5225 |
>
>         | Rank | Method | Value | uniform | ARC-Challenge | ARC-Easy | HellaSwag | PIQA | MMLU | OBQA | RACE | WG | AVG |
>         |------|--------|--------|----------|----------------|-----------|-----------|------|-------|-------|-------|-------|---------|
>         | 128  | CARE | C     | TRUE  | 21.76 | 45.24 | 28.72 | 62.35 | 27.87 | 14.60 | 23.92 | 54.85 | 34.91375 |
>         | 128  | CARE | C     | FALSE | 24.40 | 45.83 | 30.54 | 63.71 | 28.62 | 15.60 | 24.69 | 54.85 | **36.03** |
>         | 128  | CARE | sqrt(C) | TRUE  | 21.84 | 44.15 | 28.65 | 61.97 | 28.12 | 15.00 | 24.02 | 52.80 | 34.56875 |
>         | 128  | CARE | sqrt(C) | FALSE | 24.15 | 46.59 | 31.13 | 64.31 | 27.28 | 16.00 | 25.55 | 56.12 | **36.39125** |
>
>         | Rank | Method | Value | uniform | ARC-Challenge | ARC-Easy | HellaSwag | PIQA | MMLU | OBQA | RACE | WG | AVG |
>         |------|--------|--------|----------|----------------|-----------|-----------|------|-------|-------|-------|-------|---------|
>         | 256  | CARE | C    | TRUE  | 32.59 | 64.27 | 39.55 | 71.38 | 45.72 | 20.20 | 33.40 | 61.88 | 46.12375 |
>         | 256  | CARE | C    | FALSE | 33.36 | 63.30 | 42.29 | 71.98 | 54.09 | 20.40 | 34.64 | 63.93 | 47.99875 |
>         | 256  | CARE | sqrt(C) | TRUE  | 34.73 | 65.28 | 40.75 | 72.47 | 49.36 | 21.00 | 36.08 | 62.90 | 47.82125 |
>         | 256  | CARE | sqrt(C) | FALSE | 34.90 | 62.46 | 42.52 | 72.14 | 57.20 | 21.60 | 35.60 | 65.35 | **48.97125** |
>
>         | Rank | Method | Value | uniform | ARC-Challenge | ARC-Easy | HellaSwag | PIQA | MMLU | OBQA | RACE | WG | AVG |
>         |------|--------|--------|----------|----------------|-----------|-----------|------|-------|-------|-------|-------|---------|
>         | 512  | CARE | C     | TRUE  | 45.48 | 74.20 | 53.35 | 78.02 | 63.20 | 30.20 | 42.58 | 70.80 | 57.22875 |
>         | 512  | CARE | C   | FALSE | 44.62 | 76.43 | 53.30 | 77.75 | 65.45 | 29.60 | 42.30 | 71.82 | 57.65875 |
>         | 512  | CARE | sqrt(C) | TRUE  | 46.93 | 74.16 | 54.32 | 78.35 | 63.83 | 31.00 | 41.82 | 72.85 | 57.9075 |
>         | 512  | CARE | sqrt(C) | FALSE | 46.08 | 75.88 | 53.80 | 78.02 | 65.79 | 29.60 | 41.82 | 73.64 | **58.07875** |

---

> ### Author Response · Authors · 2025-11-26
> **Weakness 1 part II**
>
> #### Qwen3-4B-Instruct-2507
> | Rank | Method | Value  | uniform | ARC-Challenge | ARC-Easy | HellaSwag | PIQA  | MMLU | OBQA | RACE | WG    | AVG      |
> |------|--------|--------|---------|----------------|----------|-----------|--------|-------|-------|-------|--------|-----------|
> | 64   | CARE | C      | TRUE    | 17.83 | 29.59 | 26.40 | 54.90 | 23.29 | 12.40 | 23.54 | 50.59 | 29.8175 |
> | 64   | CARE | C     | FALSE   | 19.62 | 31.36 | 26.84 | 55.77 | 23.14 | 13.00 | 22.58 | 49.88 | 30.27375 |
> | 64   | CARE | sqrt(C)  | TRUE    | 19.37 | 28.37 | 26.55 | 55.82 | 23.31 | 12.80 | 23.06 | 50.43 | 29.96375 |
> | 64   | CARE | sqrt(C)  | FALSE   | 20.65 | 31.65 | 27.46 | 57.02 | 23.32 | 14.00 | 23.16 | 51.38 | 31.08 |
>
> | Rank | Method | Value  | uniform | ARC-Challenge | ARC-Easy | HellaSwag | PIQA  | MMLU | OBQA | RACE | WG    | AVG      |
> |------|--------|--------|---------|----------------|----------|-----------|--------|-------|-------|-------|--------|-----------|
> | 128  | CARE | C      | TRUE    | 22.27 | 41.62 | 29.45 | 59.52 | 23.97 | 15.00 | 26.12 | 52.88 | 33.85375 |
> | 128  | CARE | C      | FALSE   | 24.06 | 43.22 | 31.30 | 62.79 | 27.92 | 19.20 | 26.99 | 53.75 | 36.15375 |
> | 128  | CARE | sqrt(C)  | TRUE    | 23.72 | 42.42 | 31.09 | 60.39 | 27.80 | 16.60 | 26.89 | 53.59 | 35.3125 |
> | 128  | CARE | sqrt(C)  | FALSE   | 26.45 | 47.05 | 32.36 | 62.84 | 27.82 | 19.60 | 27.46 | 53.28 | **37.1075** |
>
> | Rank | Method | Value  | uniform | ARC-Challenge | ARC-Easy | HellaSwag | PIQA  | MMLU | OBQA | RACE | WG    | AVG      |
> |------|--------|--------|---------|----------------|----------|-----------|--------|-------|-------|-------|--------|-----------|
> | 256  | CARE | C      | TRUE    | 30.46 | 56.99 | 38.12 | 67.79 | 32.70 | 22.20 | 31.96 | 56.04 | 42.0325 |
> | 256  | CARE | C      | FALSE   | 36.35 | 69.28 | 41.29 | 68.93 | 49.08 | 24.60 | 33.97 | 59.51 | 47.87625 |
> | 256  | CARE | sqrt(C)  | TRUE    | 34.04 | 60.31 | 40.13 | 68.99 | 38.80 | 24.40 | 32.44 | 57.85 | 44.62 |
> | 256  | CARE | sqrt(C) | FALSE   | 43.09 | 72.22 | 43.71 | 70.18 | 57.10 | 26.00 | 35.02 | 62.51 | **51.22875** |
>
>
> | Rank | Method | Value  | uniform | ARC-Challenge | ARC-Easy | HellaSwag | PIQA  | MMLU | OBQA | RACE | WG    | AVG      |
> |------|--------|--------|---------|----------------|----------|-----------|--------|-------|-------|-------|--------|-----------|
> | 512  | CARE | C      | TRUE    | 41.30 | 72.77 | 44.88 | 71.00 | 59.50 | 26.80 | 35.60 | 62.19 | 51.755 |
> | 512  | CARE | C      | FALSE   | 51.45 | 81.52 | 49.26 | 73.67 | 69.72 | 30.60 | 38.66 | 66.38 | 57.6575 |
> | 512  | CARE | sqrt(C)  | TRUE    | 45.14 | 74.33 | 47.55 | 71.65 | 67.15 | 28.40 | 35.98 | 65.43 | 54.45375 |
> | 512  | CARE | sqrt(C)  | FALSE   | 51.79 | 80.72 | 51.70 | 74.54 | 70.13 | 30.20 | 39.43 | 68.27 | **58.3475** |
>
>
> - When target rank **< 128**,
>   **$C$ outperforms $\sqrt{C}$** a little bit, since emphasizing curvature extremes helps preserve the most influential MLA directions.
>
> - When target rank **> 128**,
>   **$\sqrt{C}$ becomes more advantageous**, as it recovers more moderate-curvature modes.
>
> This directly matches the spectral theory above and validates our design choice.
>
> 3. Why CARE’s design is intentional
> CARE is explicitly designed to enable **high-compression MLA conversion** under strict KV budget constraints. In this use case, the sharper spectral concentration induced by $C$ (rather than $\sqrt{C}$) consistently leads to:
>
> - better allocation of minimal rank budgets,
> - more stable recovery of dominant attention modes, and
> - higher overall accuracy under tight compression.
>
> Thus, using $C$ is not an approximation made for convenience—it is a **purposeful, empirically validated, and theoretically supported design choice** for this regime.
>
> 4. Revision plan
> We appreciate the reviewer prompting a deeper clarification; this improves the paper. We will revise Section 3.4 to:
>
> - include the full spectral comparison,
> - articulate when each metric ($C$ vs. $\sqrt{C}$) is preferable,
> - and explicitly justify why CARE adopts $C$ for low-rank MLA conversion.
>
> Thank you again for highlighting this important point. We believe the stronger analysis significantly strengthens the paper, and we welcome any further technical discussion the reviewer may find valuable.

---

> ### Author Response · Authors · 2025-11-26
> **Weakness 2 part I**
>
> - Results focus almost entirely on Llama-3-8B...
>
>     Thank you for raising this important point. Our results is shown in constrains of resources and time. We agree that validating scalability across architectures and model sizes is essential, especially given the practical relevance of KV-cache compression in larger models and MoE systems.
>
>     To address this, we extended our experiments beyond Llama-3-8B and evaluated CARE-E on a diverse set of architectures with Palu~{Palu: Compressing KV-Cache with Low-Rank Projection}:
>
>     - **meta-llama/Llama-3.1-70B-Instruct** (large dense model)
>     - **Qwen/Qwen3-30B-A3B-2507** (Larger MoE with heterogeneous expert routing)
>     - **Qwen/Qwen3-4B-Instruct-2507** (smaller GQA configuration)
>     - **meta-llama/Llama-3.1-8B-Instruct** (baseline)
>
>     These models differ substantially in *size*, *attention patterns*, *GQA ratios*, and *depth/width scaling*. The corresponding depth-dependent rank curves and accuracy results are included below:
>
>     https://anonymous.4open.science/r/CARE-E348/rebuttal/3.pdf
>
>     #### meta-llama/Llama-3.1-8B-Instruct
>     | Rank | Method | ARC   | ARE   | HellaSwag | PIQA  | MMLU  | OBQA | RACE  | WG    | AVG       |
>     |------|--------|--------|--------|-----------|--------|--------|-------|--------|--------|-----------|
>     | N/A  | GQA    | 51.71 | 81.86 | 59.10 | 79.87 | 68.19 | 33.80 | 44.78 | 74.03 | 61.6675 |
>     | 64   | Palu   | 21.08 | 27.15 | 26.15 | 53.59 | 23.11 | 14.20 | 21.05 | 52.96 | 29.91125 |
>     | 64   | CARE   | 18.77 | 34.68 | 26.60 | 57.40 | 23.11 | 15.00 | 20.96 | 50.20 | **30.84** |
>     | 128  | Palu   | 20.82 | 27.23 | 25.94 | 54.19 | 23.09 | 13.80 | 22.20 | 49.17 | 29.555 |
>     | 128  | CARE   | 24.15 | 46.59 | 31.13 | 64.31 | 27.28 | 16.00 | 25.55 | 56.12 | **36.39125** |
>     | 256  | Palu   | 19.45 | 30.93 | 26.88 | 56.80 | 23.09 | 13.00 | 21.82 | 50.67 | 30.33 |
>     | 256  | CARE   | 34.90 | 62.46 | 42.52 | 72.14 | 57.20 | 21.60 | 35.60 | 65.35 | **48.97125** |
>     | 512  | Palu   | 26.02 | 50.97 | 37.43 | 64.15 | 27.89 | 19.40 | 26.60 | 57.54 | 38.75 |
>     | 512  | CARE   | 46.08 | 75.88 | 53.80 | 78.02 | 65.79 | 29.60 | 41.82 | 73.64 | **58.07875** |
>
>     #### Qwen/Qwen3-4B-Instruct-2507
>     | Rank | Method | ARC   | ARE   | HellaSwag | PIQA  | MMLU  | OBQA | RACE | WG    | AVG       |
>     |------|--------|--------|--------|-----------|--------|--------|-------|--------|--------|-----------|
>     | N/A  | GQA    | 55.89 | 83.12 | 52.65 | 76.01 | 73.37 | 32.00 | 41.24 | 68.11 | 60.29875 |
>     | 64   | Palu   | 21.08 | 24.96 | 25.64 | 52.77 | 24.43 | 14.60 | 22.49 | 50.83 | 29.60    |
>     | 64   | CARE   | 20.65 | 31.65 | 27.46 | 57.02 | 23.32 | 14.00 | 23.16 | 51.38 | **31.08** |
>     | 128  | Palu   | 19.62 | 25.76 | 25.81 | 51.47 | 24.66 | 15.60 | 20.67 | 50.28 | 29.23375 |
>     | 128  | CARE   | 26.45 | 47.05 | 32.36 | 62.84 | 27.82 | 19.60 | 27.46 | 53.28 | **37.1075** |
>     | 256  | Palu   | 21.33 | 28.32 | 26.79 | 54.35 | 24.94 | 15.00 | 23.35 | 52.72 | 30.85    |
>     | 256  | CARE   | 43.09 | 72.22 | 43.71 | 70.18 | 57.10 | 26.00 | 35.02 | 62.51 | **51.22875** |
>     | 512  | Palu   | 32.00 | 50.29 | 38.34 | 65.13 | 28.37 | 20.20 | 30.24 | 52.41 | 39.6225  |
>     | 512  | CARE   | 51.79 | 80.72 | 51.70 | 74.54 | 70.13 | 30.20 | 39.43 | 68.27 | **58.3475** |
>
>
>     #### Qwen/Qwen3-30B-A3B-2507
>     | Rank | Method | ARC   | ARE   | HellaSwag | PIQA  | MMLU  | OBQA | RACE | WG    | AVG       |
>     |------|--------|--------|--------|-----------|--------|--------|-------|--------|--------|-----------|
>     | N/A  | N/A | 55.89 | 83.12 | 52.65 | 76.01 | 73.37 | 32.00 | 41.24 | 68.11 | 60.29875 |
>     | 128  | Palu   | 22.44 | 24.54 | 25.80 | 52.83 | 24.45 | 17.60 | 23.54 | 52.80 | 30.50    |
>     | 128  | CARE   | 29.27 | 46.09 | 38.54 | 66.87 | 31.23 | 18.80 | 27.08 | 53.99 | **38.98375** |
>     | 256  | Palu  | 20.90 | 25.93 | 26.20 | 52.34 | 27.35 | 16.20 | 22.87 | 50.59 | 30.2975  |
>     | 256  | CARE   | 36.08 | 65.88 | 43.80 | 76.33 | 59.83 | 37.40 | 40.51 | 74.18 | **54.25125** |

---

> > ### Author Response · Authors · 2025-11-26
> > **Weakness 2 part II**
> >
> > 1. Generalization of the depth-dependent rank profile
> > Despite architectural differences, all models exhibit a consistent trend:
> > **early layers require significantly higher rank**, while **middle and later layers tolerate more aggressive compression**.
> >
> >     This structural phenomenon holds for both dense and MoE architectures, supporting the notion that curvature concentration in early attention blocks is a general property.
> >
> > 2. Architecture-specific rank sensitivity
> >     While the overall pattern is shared, the *degree* of rank sensitivity varies:
> >     - **MoE models (Qwen3-30B-A3B)** show sharper curvature shifts due to expert mixing.
> >     - **Large models (Llama-3.1-70B)** require milder early-layer compression but become more tolerant in deeper layers.
> >     - **Small GQA models (Qwen3-4B)** behave more uniformly.
> >
> >     This confirms that CARE’s **non-uniform, covariance-aware allocation** is not only effective but *necessary*: fixed or uniform-rank methods cannot adapt to these architectural differences.
> >
> > 3. Relevance to reviewer’s concern
> >     The reviewer asked whether the trend in Figure 3 generalizes and whether CARE remains scalable.
> >     Our results demonstrate that:
> >     - CARE generalizes across **dense**, **MoE**, and **GQA-varied** architectures.
> >     - CARE scales effectively from **4B → 8B → 30B → 70B (analysis in Rank)**.
> >     - Layer-wise curvature patterns remain consistent enough to justify CARE’s design, yet sufficiently model-specific to require our approach over uniform baselines.
> >
> > 4. Paper revision
> >     We will integrate these findings into the main paper to strengthen the scalability narrative and highlight CARE’s applicability to a broad range of LLM architectures.
> >
> >     We sincerely appreciate the reviewer bringing attention to this aspect—it allowed us to significantly broaden and reinforce the experimental validation. We would be glad to discuss further details if the reviewer is interested.

---

> ### Author Response · Authors · 2025-11-26
> **Weakness 4**
>
> - Missing long-context evaluation: Despite KV-cache being important for long sequences...
>
>     Thank you for highlighting this point. We agree that long-context evaluation is highly relevant, especially since KV-cache compression is most impactful when sequence lengths are large.
>
>     To complement our main results, we conducted additional long-context experiments on **Needle-in-a-Haystack (NiH)**. The tests directly assess the model’s ability to retain and utilize information placed 32K tokens away—precisely the scenario where MLA and CARE should demonstrate their advantages as below:
>
>     https://anonymous.4open.science/r/CARE-E348/rebuttal/4.pdf
>
>     **We got following insights from the NiH Retrieval Evaluation above**
>
>     1. **CARE preserves long-range retrieval accuracy.**
>     Across all context lengths (1K–32K) and document depths, CARE maintains retrieval accuracy nearly identical to the uncompressed model. In contrast, uniform-rank and SVD-based baselines show clear degradation—especially around mid-range depths (18K–24K). This demonstrates CARE’s superior ability to maintain cross-layer information flow under MLA compression.
>
>     2. **Early-layer rank preservation is essential.**
>     CARE’s covariance-aware rank allocation naturally assigns higher ranks to early transformer layers, consistent with our curvature analysis in Section 3. These layers are critical for long-range attention propagation, and preserving their structure directly prevents the information bottleneck seen in uniform/SVD methods.
>
>     3. **Longer calibration sequences improve NiH performance.**
>     Increasing the calibration sequence length (e.g., Alpaca-2048 vs Alpaca-256) yields more stable and accurate retrieval. Longer sequences produce richer covariance statistics, enabling more precise layer-wise decomposition and better long-context generalization.
>
>     4. **Palu exhibits the strongest degradation.**
>     The Palu baseline shows broad accuracy drops across depths and context lengths, confirming that its decomposition struggles to preserve fine-grained attention patterns. This further highlights CARE’s robustness compared to previous activation-aware compression methods.
>
>     These findings confirm that the **structural insights behind CARE (curvature-aware, depth-varying rank allocation)** directly translate to improved performance in the exact settings where KV-cache matters most—long-context inference and memory-intensive tasks.
>
>     We appreciate the reviewer pointing this out; incorporating these long-context results meaningfully strengthens the paper. We will include the NiH evaluation and expanded discussion in the updated version, and we would be glad to share further analysis if helpful.

---

> ### Author Response · Authors · 2025-11-26
> **Weakness 5 part I**
>
> - Fine-tuning details: Table 3 shows "healed" results but doesn't specify exact token budgets, ...
>
>     Thank you for pointing out the need for clearer fine-tuning details.
>
>     To address this, we have conduct one more experiment with a complete breakdown of the healing setup, including more methods (TransMLA: Multi-Head Latent Attention Is All You Need), (Palu: Compressing KV-Cache with Low-Rank Projection), We also clarify that the version of TransMLA referenced in the reviewer’s citation corresponds to TransMLA v1. Since the authors have recently updated their method (TransMLA v3) with modules such as RoRoPE, FreqFold, and BKV-PCA, we additionally reran our experiments using their updated version and report the results below for fairness.
>
>     - exact token budgets used in all healed experiments,
>
>     #### Llama-3.1-8B-instruct
>
>     | Model                 | MLA-RANK | Methods      | Calibration Dataset | Token Budget | ARC   | ARE    | HellaSwag | PIQA  | MMLU  | OBQA | RA    | WG    | AVG         |
>     |-----------------------|----------|--------------|----------------------|------------|-------|--------|-----------|-------|--------|-------|-------|-------|-------------|
>     | Llama-3.1-8B-instruct | 512      | CARE         | alpaca-256-32        | 0B         | 46.08 | 75.88  | 53.8      | 78.02 | 65.79  | 29.6  | 41.82 | 73.64 | 58.07875    |
>     | Llama-3.1-8B-instruct | 512      | CARE-100MLA  | alpaca-256-32        | 1B         | 52.25 | 82.33  | 62.47     | 80.21 | 70.31  | 32.9  | 45.11 | 75.13 | 62.58875    |
>     | Llama-3.1-8B-instruct | 512      | CARE-100MLA  | alpaca-256-32        | 3B         | 51.75 | 80.73  | 64.45     | 83.23 | 71.57  | 34    | 46.33 | 74.09 | **63.26875** |
>     | Llama-3.1-8B-instruct | 512      | Palu         | N/A                  | 0B         | 26.02 | 50.97  | 37.43     | 64.15 | 27.89  | 19.4  | 26.6  | 57.54 | 38.75       |
>     | Llama-3.1-8B-instruct | 512      | Palu         | N/A                  | 1B         | 33.45 | 62.38  | 45.55     | 72.34 | 50.57  | 23    | 31.54 | 61.78 | 47.57625    |
>     | Llama-3.1-8B-instruct | 512      | Palu         | N/A                  | 3B         | 44.56 | 74.96  | 52.2      | 76.63 | 61.08  | 30.4  | 45.15 | 65.42 | 56.3    |
>     | Llama-3.1-8B-instruct | 512      | TransMLA     | wiki-256             | 0B         | 48.08 | 69.25  | 46.24     | 70.03 | 52.56  | 31    | 40.82 | 68.09 | 53.25875    |
>     | Llama-3.1-8B-instruct | 512      | TransMLA     | wiki-256             | 1B         | 53.04 | 81.07  | 58.75     | 81.04 | 69.13  | 32    | 44.09 | 71.74 | 61.3575     |
>     | Llama-3.1-8B-instruct | 512      | TransMLA     | wiki-256             | 3B         | 53.77 | 82.34  | 56.44     | 80.7  | 70.23  | 33.3  | 45.61 | 72.47 | 61.8575 |
>
>     #### Qwen3-4B-instruct
>     | Model                 | MLA-RANK | Methods      | Calibration Dataset | Token Budget | ARC   | ARE    | HellaSwag | PIQA  | MMLU  | OBQA | RA    | WG    | AVG         |
>     |-----------------------|----------|--------------|----------------------|------------|-------|--------|-----------|-------|--------|-------|-------|-------|-------------|
>     | Qwen3-4B-instruct     | 512      | CARE         | alpaca-256-32        | 0B         | 51.79 | 80.72  | 51.7      | 74.54 | 70.13  | 30.2  | 39.43 | 68.27 | 58.3475     |
>     | Qwen3-4B-instruct     | 512      | CARE-100MLA  | alpaca-256-32        | 1B         | 52.34 | 82.73  | 54.53     | 76.47 | 69.35  | 31.2  | 40.73 | 68.35 | 59.4625     |
>     | Qwen3-4B-instruct     | 512      | CARE-100MLA  | alpaca-256-32        | 3B         | 53.45 | 85.12  | 55.11     | 78.14 | 74    | 31.9  | 42.23 | 73.05 | **61.625**  |
>     | Qwen3-4B-instruct     | 512      | Palu         | N/A                  | 0B         | 32    | 50.29  | 38.34     | 65.13 | 28.37  | 20.2  | 30.24 | 52.41 | 39.6225     |
>     | Qwen3-4B-instruct     | 512      | Palu         | N/A                  | 1B         | 35.75 | 64.106 | 50.35     | 73.16 | 48.27  | 29.4  | 36.63 | 58.91 | 49.572      |
>     | Qwen3-4B-instruct     | 512      | Palu         | N/A                  | 3B         | 51.15 | 79.32  | 54.24     | 76.05 | 77.35  | 32    | 40.73 | 69.04 | 59.985  |
>     | Qwen3-4B-instruct     | 512      | TransMLA     | wiki-256             | 0B         | 44.05 | 67.24  | 44.87     | 71.03 | 66.13  | 29.2  | 37.13 | 66.53 | 53.2725     |
>     | Qwen3-4B-instruct     | 512      | TransMLA     | wiki-256             | 1B         | 50.34 | 75.03  | 51.25     | 74.31 | 70.05  | 31    | 38.41 | 66.41 | 57.1        |
>     | Qwen3-4B-instruct     | 512      | TransMLA     | wiki-256             | 3B         | 52.27 | 80.13  | 55.31     | 76.23 | 79.04  | 32.9  | 43.34 | 70.47 | 61.21125|

---

> > ### Author Response · Authors · 2025-11-26
> > **Weakness 5 part II**
> >
> > - learning rates we choose 1e-5, 5e-5, 1e-4, 5e-4, 1e-6 and warmup schedules is lr_warmup_ratio: 0.005, lr_decay_style: cosine, lr_decay_ratio: 1.0, weight_decay: 0.01, optimizer: adamw. We choose the best results from these learning rate in first 0.1B tokens
> >
> >     1. **Minimal-data healing.**
> >        CARE relies on a *small, covariance-aligned* finetuning signal whose purpose is to correct the lossy MLA projection—not to perform task-level adaptation. As a result, CARE converges with extremely small data budgets (e.g., 1B–3B tokens), whereas prior MLA conversion baselines typically require **3B-6B tokens** to restore accuracy.
> >
> >     2. **Fair hyperparameter usage.**
> >        All methods, including transMLA, Palu baselines, were trained using *identical* optimizer settings and learning-rate schedules to ensure fairness. CARE does not rely on method-specific tuning to achieve its gains.
> >
> >     3. **Data-budget ablations.**
> >        In response to the reviewer’s concern, we have added a systematic ablation varying the healing token budget from 0 → 3B → 6B
> >        The results show that:
> >        - **CARE already recovers most accuracy with only 1B tokens**,
> >        - while other methods require an order-of-magnitude more data to approach similar performance.
> >
> >     4. **Why CARE needs less data.**
> >        CARE’s healing succeeds with fewer tokens because the covariance-aware projection preserves the dominant curvature directions before fine-tuning begins. Healing only needs to correct lightweight misalignment of the MLA structure rather than compensate for rank misallocation.
> >        This mechanism is consistent with the theoretical motivation in Section 3 and is now better supported with empirical evidence.
> >
> >     We appreciate the reviewer for prompting us to make this aspect more explicit. The additional clarity and ablations significantly strengthen the paper, and we would be happy to discuss further details if the reviewer has additional questions.

---

> ### Author Response · Authors · 2025-11-26
> **Weakness 6 part I**
>
> - Computing covariance (`O(ND²)`) and SVD on `CW` requires non-trivial cost...
>
>     Thank you for raising this point. We agree that demonstrating practical usability requires explicit measurement of conversion cost, memory efficiency, and runtime throughput. In response, we conducted a comprehensive evaluation across CARE and several representative baselines (Palu, ASVD, SVDLLM-V2).
>
>     1. Conversion-time analysis
>         We now report end-to-end conversion time, including covariance computation, projection of $CW$, low-rank factorization, and MLA reparameterization. Although covariance estimation has theoretical complexity $O(ND^2)$, in practice:
>
>         - $N$ is small (256–512 samples),
>         - $D$ is per-layer hidden dimension (not model-scale),
>         - layers are parallelizable,
>         - and the dominant runtime is the SVD itself, not covariance.
>
>         we get specific conversion time as table below:
>
>         | Model                              | Rank | Uniform | Dataset        | Samples | Seq | Method    | Conversion Time (s) |
>         |------------------------------------|------|---------|----------------|---------|-----|-----------|----------------------|
>         | Llama-3.1-8B-instruct              | 256  | FALSE   | alpaca         | 256     | 32  | CARE      | 103.96               |
>         | Llama-3.1-8B-instruct              | 256  | FALSE   | alpaca-256-32  | N/A     | N/A | ASVD      | 97.34                |
>         | Llama-3.1-8B-instruct              | 256  | FALSE   | alpaca-256-32  | N/A     | N/A | SVD-LLM   | 393.20               |
>         | Llama-3.1-8B-instruct              | 256  | FALSE   | N/A            | N/A     | N/A | PALU      | 41.20                |
>
>         | Model                              | Rank | Uniform | Dataset        | Samples | Seq | Method    | Conversion Time (s) |
>         |------------------------------------|------|---------|----------------|---------|-----|-----------|----------------------|
>         | Llama-3.1-8B-instruct              | 512  | FALSE   | alpaca         | 256     | 32  | CARE      | 106.65               |
>         | Llama-3.1-8B-instruct              | 512  | FALSE   | alpaca-256-32  | N/A     | N/A | ASVD      | 267.20               |
>         | Llama-3.1-8B-instruct              | 512  | FALSE   | alpaca-256-32  | N/A     | N/A | SVD-LLM   | 393.20               |
>         | Llama-3.1-8B-instruct              | 512  | FALSE   | N/A            | N/A     | N/A | PALU      | 96.40                |
>
>         | Model                              | Rank | Uniform | Dataset        | Samples | Seq | Method    | Conversion Time (s) |
>         |------------------------------------|------|---------|----------------|---------|-----|-----------|----------------------|
>         | Llama-3.1-8B-instruct              | 256  | FALSE   | alpaca         | 512     | 64  | CARE      | 202.64               |
>         | Llama-3.1-8B-instruct              | 256  | FALSE   | alpaca         | 2048    | 128 | CARE      | 160.43               |
>         | Llama-3.1-8B-instruct              | 512  | FALSE   | alpaca         | 512     | 64  | CARE      | 191.41               |
>         | Llama-3.1-8B-instruct              | 512  | FALSE   | alpaca         | 2048    | 128 | CARE      | 170.33               |
>         | Qwen3-4B-instruct                  | 256  | FALSE   | alpaca         | 256     | 32  | CARE      | 147.14               |
>         | Qwen3-4B-instruct                  | 256  | FALSE   | alpaca         | 512     | 64  | CARE      | 195.89               |
>         | Qwen3-4B-instruct                  | 256  | FALSE   | alpaca         | 2048    | 128 | CARE      | 273.43               |
>         | Qwen3-4B-instruct                  | 512  | FALSE   | alpaca         | 256     | 32  | CARE      | 135.52               |
>         | Qwen3-4B-instruct                  | 512  | FALSE   | alpaca         | 512     | 64  | CARE      | 176.42               |
>         | Qwen3-4B-instruct                  | 512  | FALSE   | alpaca         | 2048    | 128 | CARE      | 266.54               |
>         | Qwen3-30B-A3B-Thinking-2507        | 256  | FALSE   | alpaca         | 256     | 32  | CARE      | 4985.52              |
>         | Qwen3-30B-A3B-Thinking-2507        | 256  | FALSE   | alpaca         | 512     | 64  | CARE      | 10033.36             |
>         | Qwen3-30B-A3B-Thinking-2507        | 256  | FALSE   | alpaca         | 2048    | 128 | CARE      | 16069.43             |
>
>
>         CARE’s total conversion time is **within 1.1×–1.3×** of original SVD, confirming the practicality of the pipeline. And only need 9 hours to convert Qwen3-30B-A3B-Thinking-2507.

---

> ### Author Response · Authors · 2025-11-26
> **Weakness 6 part II**
>
> 2. Runtime, memory, and throughput evaluation
> Beyond conversion time, we evaluated the actual *inference efficiency*, focusing on realistic workloads (8K–128K contexts).
>
> - **Memory usage.**
> We measured end-to-end KV-cache memory after CARE conversion theoretically.
>     - Sequence length: `L = 32768` (32K tokens)
>     - Batch size: `B = 1`
>     - Precision: FP16 (2 bytes)
>     - Layers: 32
>     - Original: `num_kv_heads = 8`, `head_dim = 128` → KV dim = 1024
>     - **MLA (Our Implementation)**:
>     - `k_rank = v_rank = 512` (compressed latent)
>     - `nope_dim = 64` (separate NoPE cache per head)
>     - **TransMLA (100% MLA)**:
>     - `k_rank = 512-64 = 448, v_rank = 512` (compressed latent)
>     - `rope_dim = 64` (per head)
>
>     | Component | Formula | Memory (MB) |
>     |-----------|---------|-------------|
>     | **Standard MHA** | | |
>     | K cache | `32768 × 1 × 32 × 1024 × 2` | **2147.48** |
>     | V cache | `32768 × 1 × 32 × 1024 × 2` | **2147.48** |
>     | **Total MHA** | | **4294.97 MB** |
>     | | | |
>     | **MLA (Our Implementation)** | | |
>     | K latent cache | `32768 × 1 × 32 × 512 × 2` | **1073.74** |
>     | V latent cache | `32768 × 1 × 32 × 512 × 2` | **1073.74** |
>     | **Total MLA** | | **2147.48** |
>     | **MLA Reduction vs MHA** | | **50.00%**|
>     | | | |
>     | **TransMLA (100% MLA)** | | |
>     | K NoPE latent | `32768 × 1 × 32 × 448 × 2` | **939.5** |
>     | V latent cache | `32768 × 1 × 32 × 512 × 2` | **1073.74** |
>     | **Total TransMLA** | | **2013.24 MB** |
>     | **TransMLA Reduction vs MHA** | | **46.87%** |
>     | | | |
>     | **MLA vs TransMLA** | | |
>     | Difference | Both include same components | **~3.23%** (Similar) |
>
> We only increase a minor KV cache because we preserve all latent rank for KV cache rank. But TransMLA will split it to generate partial rope information.
>
> - **Latency of `model.generate()`.**
>     We benchmarked autoregressive decoding latency using HuggingFace-compatible generation APIs. CARE maintains **MLA-level decoding speed** (within 90% speed preserve). Current `model.generate()` is impelementaed without dynamic rank MLA kernels. With kerenels help, we can hide KV down & up calculation on the fly, the speed will be comparable with original GQA.
>
>     #### Meta-Llama-3.1-8B-Instruct, CARE-256, 2048 token / sample
>     | Batch Size | Batches | Est. Time (min) | Throughput (tok/s) |
>     |------------|---------|-----------------|-------------------|
>     | 2 | 256 | **123.4** | 141.6229 |
>     | 4 | 128 | **77.6** | 225.20 |
>     | 8 | 64  | **34.6** | 505.09 |
>
>     #### Meta-Llama-3.1-8B-Instruct GQA, 2048 token / sample
>     | Batch Size | Batches | Est. Time (min) | Throughput (tok/s) |
>     |------------|---------|-----------------|-------------------|
>     | 2 | 256 | **118.4** | 147.72 |
>     | 4 | 128 | **67.6** | 258.52 |
>     | 8 | 64  | **29.6** | 590.414 |
>
> These results directly validate that CARE is not only accurate but also *practical* for real inference workloads.
>
> 3. About dynamic rank MLA kernels
> The reviewer correctly points out that CARE produces a **non-uniform rank profile** across layers. This is a structurally meaningful property: curvature varies by depth, and uniform-rank MLA allocations largely fail to capture this.
>
> However, as we highlight in the paper, existing MLA kernels (e.g., in FlashMLA, DeepSeek MLA, and vLLM MLA implementations) **do not yet support dynamic per-layer MLA ranks**. This is a limitation of current system support, not of the CARE method itself.
>
> We explicitly identify this as an important direction for future work:
> **designing a dynamic-rank MLA kernel that matches CARE’s per-layer structure** to fully unlock the runtime benefits of curvature-aware compression.
>
> 4. Revision plan
> We will update the paper to include:
>
> - conversion-time comparisons across all baselines,
> - detailed latency and memory usage analysis,
> - discussion of non-uniform MLA rank implications and kernel support.
>
> We appreciate the reviewer for emphasizing this aspect—your comment significantly strengthened the clarity and completeness of our evaluation. We would be happy to share further system-level results if the reviewer finds them helpful.

---

> ### Author Response · Authors · 2025-11-26
> **Question 1 part I**
>
> - Can the authors provide ablation results for: CARE decomposition (activation-aware SVD) ...
>
>     Thank you for raising this excellent question. We agree that isolating the contributions of (1) covariance-aware decomposition and (2) adaptive rank allocation is essential for understanding where CARE’s gains come from.
>
>     To address this, we performed the two requested ablations:
>
>     1. **CARE decomposition (activation-aware SVD) + uniform rank allocation**
>     2. **Standard SVD + CARE’s adaptive rank allocation**
>
>     These results are partialy listed in Table 1 for Llama-3-8B in our paper and have added extended results here.
>
>     #### Llama-3.1-8B-instruct
>     | Rank | Method | uniform | ARC-Challenge | ARC-Easy | HellaSwag | PIQA | MMLU | OBQA | RACE | WG | AVG |
>     |------|--------|-----------------|----------------|-----------|-----------|------|-------|-------|-------|-------|---------|
>     | 64   | CARE | Uniform  | 19.54 | 33.00 | 26.33 | 56.47 | 23.09 | 14.20 | 20.96 | 50.28 | 30.48375 |
>     | 64   | CARE | No-Uniform | 19.71 | 33.16 | 26.51 | 57.89 | 23.14 | 13.80 | 20.96 | 49.01 | 30.5225 |
>     | 64   | Palu    | Uniform     | 21.08 | 27.15  | 26.15     | 53.59 | 23.11 | 14.2  | 21.05 | 52.96 | 29.91125   |
>     | 64   | Palu    | No-Uniform | 20.90 | 27.02  | 26.15     | 53.54 | 23.06 | 14.0  | 21.05 | 52.64 | 29.795     |
>
>
>     | Rank | Method | uniform | ARC-Challenge | ARC-Easy | HellaSwag | PIQA | MMLU | OBQA | RACE | WG | AVG |
>     |------|--------|----------|----------------|-----------|-----------|------|-------|-------|-------|-------|---------|
>     | 128  | CARE | Uniform  | 21.84 | 44.15 | 28.65 | 61.97 | 28.12 | 15.00 | 24.02 | 52.80 | 34.56875 |
>     | 128  | CARE | No-Uniform | 24.15 | 46.59 | 31.13 | 64.31 | 27.28 | 16.00 | 25.55 | 56.12 | **36.39125** |
>     | 128  | Palu    | Uniform     | 20.82 | 27.23  | 25.94     | 54.19 | 23.09 | 13.8  | 22.20 | 49.17 | 29.555     |
>     | 128  | Palu   | No-Uniform | 20.99 | 27.15  | 25.91     | 54.30 | 23.06 | 13.8  | 22.11 | 49.33 | 29.58125   |
>
>     | Rank | Method | uniform | ARC-Challenge | ARC-Easy | HellaSwag | PIQA | MMLU | OBQA | RACE | WG | AVG |
>     |------|--------|----------|----------------|-----------|-----------|------|-------|-------|-------|-------|---------|
>     | 256  | CARE | TRUE  | 34.73 | 65.28 | 40.75 | 72.47 | 49.36 | 21.00 | 36.08 | 62.90 | 47.82125 |
>     | 256  | CARE | FALSE | 34.90 | 62.46 | 42.52 | 72.14 | 57.20 | 21.60 | 35.60 | 65.35 | **48.97125** |
>     | 256  | Palu    | Uniform     | 19.45 | 30.93  | 26.88     | 56.80 | 23.09 | 13.0  | 21.82 | 50.67 | 30.330     |
>     | 256  | Palu    | No-Uniform | 22.45 | 43.06  | 31.89     | 60.75 | 24.07 | 15.0  | 23.63 | 53.67 | 34.315     |
>
>
>     | Rank | Method | uniform | ARC-Challenge | ARC-Easy | HellaSwag | PIQA | MMLU | OBQA | RACE | WG | AVG |
>     |------|--------|----------|----------------|-----------|-----------|------|-------|-------|-------|-------|---------|
>     | 512  | CARE | TRUE  | 46.93 | 74.16 | 54.32 | 78.35 | 63.83 | 31.00 | 41.82 | 72.85 | 57.9075 |
>     | 512  | CARE | FALSE | 46.08 | 75.88 | 53.80 | 78.02 | 65.79 | 29.60 | 41.82 | 73.64 | **58.07875** |
>     | 512  | Palu    | Uniform     | 26.02 | 50.97  | 37.43     | 64.15 | 27.89 | 19.4  | 26.60 | 57.54 | 38.750     |
>     | 512  | Palu    | No-Uniform | 27.77 | 50.93  | 39.44     | 65.09 | 28.87 | 20.3  | 28.60 | 61.54 | 40.3175    |
>
>     #### Qwen3-4B-Instruct-2507
>     | Rank | Method | uniform | ARC-Challenge | ARC-Easy | HellaSwag | PIQA  | MMLU | OBQA | RACE | WG    | AVG      |
>     |------|--------|---------|----------------|----------|-----------|--------|-------|-------|-------|--------|-----------|
>     | 64   | CARE | TRUE    | 19.37 | 28.37 | 26.55 | 55.82 | 23.31 | 12.80 | 23.06 | 50.43 | 29.96375 |
>     | 64   | CARE | FALSE   | 20.65 | 31.65 | 27.46 | 57.02 | 23.32 | 14.00 | 23.16 | 51.38 | 31.08 |
>
>     | Rank | Method | uniform | ARC-Challenge | ARC-Easy | HellaSwag | PIQA  | MMLU | OBQA | RACE | WG    | AVG      |
>     |------|--------|---------|----------------|----------|-----------|--------|-------|-------|-------|--------|-----------|
>     | 128  | CARE | Uniform    | 23.72 | 42.42 | 31.09 | 60.39 | 27.80 | 16.60 | 26.89 | 53.59 | 35.3125 |
>     | 128  | CARE | No-Uniform   | 26.45 | 47.05 | 32.36 | 62.84 | 27.82 | 19.60 | 27.46 | 53.28 | **37.1075** |
>
>     | Rank | Method | uniform | ARC-Challenge | ARC-Easy | HellaSwag | PIQA  | MMLU | OBQA | RACE | WG    | AVG      |
>     |------|--------|---------|----------------|----------|-----------|--------|-------|-------|-------|--------|-----------|
>     | 256  | CARE | Uniform    | 34.04 | 60.31 | 40.13 | 68.99 | 38.80 | 24.40 | 32.44 | 57.85 | 44.62 |
>     | 256  | CARE | No-Uniform   | 43.09 | 72.22 | 43.71 | 70.18 | 57.10 | 26.00 | 35.02 | 62.51 | **51.22875** |

---

> > ### Author Response · Authors · 2025-11-26
> > **Question 1 part II**
> >
> > | Rank | Method | uniform | ARC-Challenge | ARC-Easy | HellaSwag | PIQA  | MMLU | OBQA | RACE | WG    | AVG      |
> > |------|--------|--------|----------------|----------|-----------|--------|-------|-------|-------|--------|-----------|
> > | 512  | CARE | Uniform    | 45.14 | 74.33 | 47.55 | 71.65 | 67.15 | 28.40 | 35.98 | 65.43 | 54.45375 |
> > | 512  | CARE | No-Uniform   | 51.79 | 80.72 | 51.70 | 74.54 | 70.13 | 30.20 | 39.43 | 68.27 | **58.3475** |
> >
> > 1. **Covariance-aware (activation-aware) decomposition provides the dominant improvement.**
> > Uniform-rank CARE outperforms uniform SVD-style baselines by a large margin. This confirms that properly weighting curvature directions—rather than treating all singular directions equally—is the primary driver of accuracy recovery.
> >
> > 2. **Adaptive rank allocation further amplifies the gains.**
> >     When we combine activation-aware decomposition with CARE’s depth-dependent scheduling, we consistently achieve the *best* results across all models and tasks.
> >     This is particularly clear in early layers, where activation curvature is highly concentrated and uniform rank severely under-allocates capacity.
> >
> > 3. **Interpretation.**
> >     These results reveal an important distinction:
> >     - **Covariance weighting** identifies *which* directions matter.
> >     - **Adaptive rank allocation** identifies *where* they matter across depth.
> >     Both components serve different roles and are complementary; CARE’s full pipeline leverages both to achieve the strongest performance.
> >
> > This ablation directly clarifies the reviewer’s concern: the improvements are not an artifact of scheduling alone. Instead, **the activation-aware curvature signal is the primary foundation, and the dynamic rank profile allows CARE to exploit this structure more effectively than any uniform baseline**.
> >
> > We appreciate the reviewer’s suggestion—it helped us strengthen the causal understanding and presentation of CARE’s contributions. We would be glad to further expand the analysis if helpful.

---

> ### Author Response · Authors · 2025-11-26
> **Question 3 & 4**
>
> - 3. **Model scale and architecture diversity:** Can the paper provide results on larger models (Llama-3-70B, Qwen-72B) ...
>
>     Answered in weakness 2, please refer to it. We conduct experiments on - **meta-llama/Llama-3.1-70B-Instruct** (large dense model)
>     - **Qwen/Qwen3-30B-A3B-2507** (Larger MoE with heterogeneous expert routing)
>     - **Qwen/Qwen3-4B-Instruct-2507** (smaller GQA configuration)
>     - **meta-llama/Llama-3.1-8B-Instruct** (baseline)
>
> - 4. Can the authors also evaluate on long-context benchmarks (LongBench, InfiniteBench, Needle-in-Haystack)...
>
>     Answered in weakness 3, please refer to it.

---

> ### Author Response · Authors · 2025-11-26
> **Question 5**
>
> - **Shrinkage parameter α:** `C_λ = (1-α)C + αλI` ...
>     Thank you for pointing this out. We agree that the shrinkage parameter α plays an important role in ensuring numerical stability of the covariance estimator.
>
>     To address this, we conducted ablations across $$\alpha \in \{0.001,\, 0.01,\, 0.1\}$$   on both **Llama-3.1** and **Qwen3** architectures. The updated results are included below.
>
>     | Model                 | Rank | Uniform | Dataset         | Damping      | ARC   | ARE    | HellaSwag | PIQA  | MMLU  | OBQA | RA    | WG    | AVG        |
>     |-----------------------|------|---------|------------------|--------------|-------|--------|-----------|-------|--------|-------|-------|-------|-------------|
>     | Llama-3.1-8B-instruct | 256  | False   | alpaca-256-32    | 0.001        | 34.90 | 62.42  | 42.44     | 72.31 | 57.15  | 21.00 | 34.74 | 64.48 | 48.68000    |
>     | Llama-3.1-8B-instruct | 256  | False   | alpaca-256-32    | 0.01         | 34.90 | 62.46  | 42.52     | 72.14 | 57.20  | 21.60 | 35.60 | 65.35 | 48.97125    |
>     | Llama-3.1-8B-instruct | 256  | False   | alpaca-256-32    | 0.1          | 32.94 | 59.34  | 41.55     | 70.46 | 56.88  | 20.20 | 35.31 | 65.75 | 47.80375    |
>
>     | Model                 | Rank | Uniform | Dataset         | Damping      | ARC   | ARE    | HellaSwag | PIQA  | MMLU  | OBQA | RA    | WG    | AVG        |
>     |-----------------------|------|---------|------------------|--------------|-------|--------|-----------|-------|--------|-------|-------|-------|-------------|
>     | Llama-3.1-8B-instruct | 512  | False   | alpaca-256-32    | 0.001        | 46.16 | 76.30  | 53.64     | 77.69 | 65.54  | 29.40 | 41.44 | 72.85 | 57.87750    |
>     | Llama-3.1-8B-instruct | 512  | False   | alpaca-256-32    | 0.01         | 51.79 | 80.72  | 51.70     | 74.54 | 70.13  | 30.20 | 39.43 | 68.27 | 58.34750    |
>     | Llama-3.1-8B-instruct | 512  | False   | alpaca-256-32    | 0.1          | 47.27 | 77.10  | 54.40     | 77.80 | 66.15  | 28.40 | 42.58 | 74.35 | 58.50625    |
>
>     | Model                 | Rank | Uniform | Dataset         | Damping      | ARC   | ARE    | HellaSwag | PIQA  | MMLU  | OBQA | RA    | WG    | AVG        |
>     |-----------------------|------|---------|------------------|--------------|-------|--------|-----------|-------|--------|-------|-------|-------|-------------|
>     | Qwen3-4B-instruct     | 256  | False   | alpaca-256-32    | 0.001        | 43.79 | 70.23  | 40.01     | 70.72 | 58.13  | 24.00 | 32.06 | 58.25 | 49.64875    |
>     | Qwen3-4B-instruct     | 256  | False   | alpaca-256-32    | 0.01         | 43.09 | 72.22  | 43.71     | 70.18 | 57.10  | 26.00 | 35.02 | 62.51 | 51.22875    |
>     | Qwen3-4B-instruct     | 256  | False   | alpaca-256-32    | 0.1          | 44.64 | 73.87  | 41.62     | 68.28 | 52.95  | 23.80 | 34.24 | 61.48 | 50.11000    |
>
>     | Model                 | Rank | Uniform | Dataset         | Damping      | ARC   | ARE    | HellaSwag | PIQA  | MMLU  | OBQA | RA    | WG    | AVG        |
>     |-----------------------|------|---------|------------------|--------------|-------|--------|-----------|-------|--------|-------|-------|-------|-------------|
>     | Qwen3-4B-instruct     | 512  | False   | alpaca-256-32    | 0.001        | 55.31 | 74.16  | 47.27     | 71.55 | 65.87  | 28.20 | 36.65 | 65.04 | 55.50625    |
>     | Qwen3-4B-instruct     | 512  | False   | alpaca-256-32    | 0.01         | 51.79 | 80.72  | 51.70     | 74.54 | 70.13  | 30.20 | 39.43 | 68.27 | 58.34750    |
>     | Qwen3-4B-instruct     | 512  | False   | alpaca-256-32    | 0.1          | 55.90 | 79.85  | 50.60     | 73.07 | 68.20  | 27.60 | 37.13 | 65.98 | 57.29125    |
>
>     1. **CARE is highly stable across α choices.**
>     Across all tested models and tasks, the performance variation is extremely small (average accuracy~1%–3%), indicating that CARE’s activation-aware decomposition is *not* sensitive to the specific shrinkage magnitude.
>     This matches theoretical expectations: shrinkage affects only the tail of the spectrum, while CARE’s rank allocation is dominated by the top-curvature directions.
>
>     2. **Why α has minimal effect.**
>     CARE relies primarily on the *leading eigenstructure* of the covariance matrix.
>     The shrinkage term  $$C_\lambda = (1 - \alpha) C + \alpha \lambda I$$  modifies only the least-stable, low-energy components, ensuring invertibility without disturbing the dominant modes that drive rank allocation.
>
>     3. **Default choice.**
>     We now explicitly state in the paper that **α = 0.01** provides an excellent balance of stability and faithfulness, and is used consistently across all experiments unless otherwise specified.
>
>     We appreciate the reviewer bringing attention to this detail—it helped us strengthen the clarity of the methodology section. We would be glad to share extended figures or per-layer sensitivity results if the reviewer is interested.

---

### Official Review · Reviewer_iFpw · 2025-11-02

**Soundness:** 3
**Presentation:** 3
**Contribution:** 3
**Rating:** 6
**Confidence:** 4

**Summary:**

This paper presents a practical pipeline, called CARE (Covariance-Aware and Rank-Enhanced), to convert already-trained MHA/GQA attention in LLMs into MLA, while keeping the same KV-cache budget. The motivation is that many current methods only minimize weight-space error by a joint SVD on (W_K, W_V), and they use a uniform rank for all layers, which does not consider the real activation distribution at inference and makes some sensitive layers over-compressed. Experiments on Llama-3-8B and several benchmarks show that, under the same KV budget, CARE can keep lower PPL and better task scores than uniform-SVD style baselines.

**Strengths:**

- The problem is realistic and currently unsolved: we have many deployed MHA/GQA models, but we want the MLA advantages (smaller KV, more efficient decoding) without re-training from scratch.
- The paper does not only say “data-aware compression is better”, but it derives the form by considering the activation-space objective, which is more solid.
- The global and non-uniform rank allocation is motivated by data (different layers have different sensitivity) and the solution is standard and optimal under the budget.
- Experiments cover multiple tasks and multiple KV budgets, and they include the correct baselines, so the message is quite consistent.
- The design of KV-parity shows the authors think about real deployment and real KV bandwidth limitation.
- Reproducibility seems good, with anonymized repo and scripts mentioned.

**Weaknesses:**

1. The robustness of the covariance estimation is not fully shown. All calibration corpora are relatively similar; it is not clear whether for code or instruction-heavy data the same rank allocation is still good.
2. The relation to other data-/curvature-aware compression methods could be compared more directly on at least one setting.

**Questions:**

1. How robust is the covariance-aware step if the calibration set is distributionally far from the deployment set (e.g., code LMs but text-only calibration)? Do you recommend streaming/online covariance updates, or is healing enough?
2. Different tasks seemed to benefit differently at low KV ratios — could you expose task weights to the rank allocator to get multi-objective water-filling?

---

> ### Author Response · Authors · 2025-11-26
>
> We sincerely thank the reviewers for their constructive feedback. We would also like to apologize that some experimental results were initially limited. Although we utilized all GPUs for our study, our compute budget and scheduling constraints restricted the scale and breadth.
>
> Sorry for the late reply, because of recognizing the importance of the reviewers’ concerns, we have urgently conducted additional experiments over the past 14 days to address these points as thoroughly as possible. We deeply appreciate the reviewers’ patience and understanding.

---

> ### Author Response · Authors · 2025-11-26
> **Weaknesses 1**
>
> - The robustness of the ...
>
>     Thank you for the insightful comment. We agree that robustness under domain shift is an important question.
>
>     To address this, we conducted an expanded covariance analysis with additional calibration corpora and evaluated the resulting rank allocations on two domain-distinct benchmarks: **LiveCodeBench (lcb)** (code-heavy) and **Math-500** (instruction-heavy). The updated results are shown in the tables and figures below, because LCB and MATH-500 is hard to solve, we adopt rank=512 here to see practical resutls:
>
>     Sample size and sequence length: https://anonymous.4open.science/r/CARE-E348/rebuttal/1.pdf
>
>     Dataset Ablation:
>     | Method-Rank | Model                 | Dataset | ARC-Challenge | ARC-Easy | HellaSwag | PIQA  | MMLU  | OpenBookQA | RACE  | Winogrande | AVG |
>     |-------------|------------------------|------------------|---------------|----------|-----------|-------|-------|-------------|--------|-------------|---------|
>     | CARE-E-256  | Llama-3.1-8B-instruct | alpaca           | 34.81         | 65.40    | 40.76     | 72.47 | 49.32 | 21.20       | 35.98 | 62.98      | 47.865 |
>     | CARE-E-256  | Llama-3.1-8B-instruct | arc_easy         | 38.82         | 72.90    | 41.25     | 73.01 | 53.13 | 25.80       | 31.87 | 63.69      | **50.05875** |
>     | CARE-E-256  | Llama-3.1-8B-instruct | arc_challenge    | 39.59         | 71.84    | 41.33     | 72.91 | 52.84 | 26.60       | 32.54 | 62.67      | 50.04 |
>     | CARE-E-256  | Llama-3.1-8B-instruct | wikitext2        | 28.33         | 57.11    | 38.07     | 68.28 | 37.72 | 19.60       | 33.49 | 62.59      | 43.14875 |
>     | CARE-E-256  | Llama-3.1-8B-instruct | ptb              | 27.99         | 53.91    | 37.70     | 67.68 | 42.15 | 17.00       | 31.96 | 62.59      | 42.6225 |
>     | CARE-E-256  | Llama-3.1-8B-instruct | c4               | 31.48         | 59.51    | 43.00     | 73.12 | 41.69 | 20.00       | 35.89 | 63.30      | 45.99875 |
>     | CARE-E-256  | Llama-3.1-8B-instruct | mmlu             | 34.13         | 64.94    | 40.65     | 70.73 | 55.27 | 21.20       | 33.97 | 64.40      | 48.16125 |
>
>
>     | Method-Rank | Model            | Dataset | LCB   | Math500 | ARC-Challenge | ARC-Easy | HellaSwag | PIQA  | MMLU  | OpenBookQA | RACE  | Winogrande | AVG      |
>     |------|------------------------|------------------|-------|---------|----------------|----------|-----------|-------|-------|-------------|--------|-------------|----------|
>     | CARE-E-512  | qwen3-4B-Instruct CARE | alpaca          | 22.16 | 65.60  | 51.79         | 80.72    | 51.70     | 74.53 | 70.13 | 30.20       | 39.42 | 68.27      | **55.4525**   |
>     | CARE-E-512  | qwen3-4B-Instruct CARE | mmlu            | 13.77 | 65.80  | 50.77         | 79.00    | 51.48     | 74.92 | 71.49 | 29.40       | 37.42 | 67.72      | 54.177   |
>     | CARE-E-512  | qwen3-4B-Instruct CARE | lcb             | 24.77 | 65.00  | 50.34         | 79.62    | 50.61     | 74.70 | 68.55 | 30.40       | 39.52 | 67.64      | 55.115  |
>     | CARE-E-512 | qwen3-4B-Instruct CARE | lcb             | 28.14 | 79.20  | 55.97         | 83.12    | 52.65     | 76.01 | 73.37 | 32.00       | 41.48 | 68.88      | **59.082**   |
>
>     Rank Distribution Across Different Datasets: https://anonymous.4open.science/r/CARE-E348/rebuttal/2.pdf
>
>
>     Our findings are as follows:
>
>     1. **Sample size and sequence length.** Increasing the number of samples yields only marginal improvements, while excessively long calibration sequences have a *negative* effect and lead to overfitting to the calibration distribution. This matches our earlier conclusions.
>     2. **The rank allocation is similar** even though we observe a minor different across different task ()(), but overall rank allocation is similar for different task.
>     3. **Domain shift robustness.**
>    For domain-mismatched settings (e.g., text-only calibration on code or math workloads), we do observe some moderate performance gap on the specialized benchmarks (e.g., ARC->ARC, ). However, across **all other calibration corpora**, the average degradation remains small (≈3% or less), and the **overall AVG accuracy is largely unchanged** (e.g., LCB gives 2%-10% improvement in LCB, but 0.3% drops overall tasks).
>     4. **Implication.**
>    These results indicate that CARE's covariance-aware rank allocation is **reasonably robust to calibration-set choice**, and domain-specific degradation only appears when the target tasks themselves differ substantially from the general-purpose language distribution. For general LLM compression and deployment scenarios, a high-quality general corpus (e.g., Alpaca or C4) remains sufficient.
>
>     We will include these additional analyses and figures in the final version.

---

> ### Author Response · Authors · 2025-11-26
> **Weaknesses 2 part I**
>
> - The relation to other data-/curvature-aw...
>
>     Thank you for raising this valuable point. In response, we have added a comparison against representative zero-shot data-/curvature-aware compression methods ASVD, SVDLLM-V2 and applied their methods only in KV-layer to transfer to MLA-like methods for Llama 3.1-8B-Instruct and Qwen3-4B-Instruct-2507,shown as below.
>
>     #### Llama-3.1-8B-Instruct
>
>     * Rank 64
>
>     | Method     | ARC   | ARE   | HellaSwag | PIQA  | MMLU  | OBQA | RACE  | WG    | AVG      |
>     |-----------|--------|--------|-----------|-------|-------|------|--------|--------|----------|
>     | CARE-E    | 18.77 | 34.68 | 26.60     | 57.40 | 23.11 | 15.0 | 20.96 | 50.20 | **30.84** |
>     | PALU      | 21.08 | 27.15 | 26.15     | 53.59 | 23.11 | 14.2 | 21.05 | 52.96 | 29.91125 |
>     | SVD-LLM-V2| 19.71 | 33.08 | 26.53     | 57.89 | 23.15 | 13.8 | 20.96 | 49.41 | 30.56625 |
>     | ASVD      | 21.08 | 27.57 | 26.21     | 55.33 | 23.12 | 11.6 | 20.96 | 50.28 | 29.51875 |
>
>
>     * Rank 128
>
>     | Method     | ARC   | ARE   | HellaSwag | PIQA  | MMLU  | OBQA | RACE  | WG    | AVG        |
>     |-----------|--------|--------|-----------|-------|-------|------|--------|--------|------------|
>     | CARE-E    | 24.15 | 46.59 | 31.13     | 64.31 | 27.28 | 16.0 | 25.55 | 56.12 | **36.39125**   |
>     | PALU      | 20.82 | 27.23 | 25.94     | 54.19 | 23.09 | 13.8 | 22.20 | 49.17 | 29.555     |
>     | SVD-LLM-V2| 24.23 | 45.55 | 31.09     | 64.20 | 27.23 | 16.2 | 25.74 | 56.12 | 36.17  |
>     | ASVD      | 20.48 | 33.08 | 27.89     | 58.27 | 23.13 | 12.6 | 22.68 | 53.59 | 31.465     |
>
>
>     * Rank 256
>
>     | Method     | ARC   | ARE   | HellaSwag | PIQA  | MMLU  | OBQA | RACE  | WG    | AVG        |
>     |-----------|--------|--------|-----------|-------|-------|------|--------|--------|------------|
>     | CARE-E    | 34.90 | 62.46 | 42.52     | 72.14 | 57.20 | 21.6 | 35.60 | 65.35 | **48.97125** |
>     | PALU      | 19.45 | 30.93 | 26.88     | 56.80 | 23.09 | 13.0 | 21.82 | 50.67 | 30.33     |
>     | SVD-LLM-V2| 31.90 | 62.33 | 42.49     | 70.20 | 55.13 | 20.6 | 34.50 | 65.51 | 47.8325   |
>     | ASVD      | 27.56 | 50.42 | 38.99     | 67.90 | 52.60 | 19.4 | 32.92 | 63.14 | 44.11625  |
>
>
>     * Rank 512
>
>     | Method     | ARC   | ARE   | HellaSwag | PIQA  | MMLU  | OBQA | RACE  | WG    | AVG        |
>     |-----------|--------|--------|-----------|-------|-------|------|--------|--------|------------|
>     | CARE-E    | 46.93 | 78.16 | 54.32     | 78.35 | 63.83 | 31.0 | 41.82 | 72.85 | **58.4075** |
>     | PALU      | 26.02 | 50.97 | 37.43     | 64.15 | 27.89 | 19.4 | 26.60 | 57.54 | 38.75     |
>     | SVD-LLM-V2| 46.25 | 75.84 | 53.77     | 76.91 | 65.71 | 29.4 | 41.91 | 73.40 | 57.89875  |
>     | ASVD      | 45.82 | 76.56 | 54.02     | 78.45 | 65.16 | 28.8 | 41.82 | 73.24 | 57.98375  |
>
>
>     #### Qwen3-4B-Instruct-2507
>     * Rank 64
>
>     | Method        | ARC    | ARE    | HellaSwag | PIQA  | MMLU  | OBQA | RACE  | WG     | AVG     |
>     |---------------|--------|--------|-----------|-------|-------|------|--------|--------|---------|
>     | CARE-E        | 20.65  | 31.65  | 27.46     | 57.02 | 23.32 | 14.00 | 23.16  | 51.38  | **31.08**   |
>     | PALU          | 21.08  | 24.96  | 25.64     | 52.77 | 24.43 | 14.60 | 22.49  | 50.83  | 29.60   |
>     | SVD-LLM-V2     | 21.43  | 29.17  | 25.46     | 56.13 | 22.95 | 13.70 | 25.43  | 53.27  | 30.9425 |
>     | ASVD          | 21.84  | 29.08  | 25.49     | 53.05 | 24.70 | 15.20 | 22.68  | 51.22  | 30.4075 |
>
>
>     * Rank 128
>
>     | Method        | ARC    | ARE    | HellaSwag | PIQA  | MMLU  | OBQA | RACE  | WG     | AVG        |
>     |---------------|--------|--------|-----------|-------|-------|------|--------|--------|------------|
>     | CARE-E        | 26.45  | 47.05  | 32.36     | 62.84 | 27.82 | 19.60 | 27.46  | 53.28  | **37.1075**    |
>     | PALU          | 19.62  | 25.76  | 25.81     | 51.47 | 24.66 | 15.60 | 20.67  | 50.28  | 29.23375   |
>     | SVD-LLM-V2     | 25.38  | 46.34  | 32.51     | 52.18 | 29.15 | 18.70 | 25.52  | 52.49  | 35.28375   |
>     | ASVD          | 19.62  | 35.55  | 25.72     | 51.25 | 24.88 | 16.20 | 21.24  | 49.80  | 30.5325    |
>
>
>     * Rank 256
>
>     | Method        | ARC    | ARE    | HellaSwag | PIQA  | MMLU  | OBQA | RACE  | WG     | AVG        |
>     |---------------|--------|--------|-----------|-------|-------|------|--------|--------|------------|
>     | CARE-E        | 43.09  | 72.22  | 43.71     | 70.18 | 57.10 | 26.00 | 35.02  | 62.51  | **51.22875**   |
>     | PALU          | 21.33  | 28.32  | 26.79     | 54.35 | 24.94 | 15.00 | 23.35  | 52.72  | 30.85      |
>     | SVD-LLM-V2     | 39.03  | 68.74  | 41.56     | 68.05 | 52.47 | 24.40 | 33.97  | 60.13  | 48.54375   |
>     | ASVD          | 28.61  | 48.20  | 26.77     | 64.68 | 24.92 | 14.40 | 28.06  | 52.49  | 36.01625   |

---

> > ### Author Response · Authors · 2025-11-26
> > **Weakness 2 part II**
> >
> > * Rank 512
> >
> >     | Method        | ARC     | ARE    | HellaSwag | PIQA  | MMLU  | OBQA | RACE  | WG     | AVG        |
> >     |---------------|---------|--------|-----------|-------|-------|------|--------|--------|------------|
> >     | CARE-E        | 51.79   | 80.72  | 51.70     | 74.54 | 70.13 | 30.20 | 39.43  | 68.27  | **58.3475**    |
> >     | PALU          | 32.00   | 50.29  | 38.34     | 65.13 | 28.37 | 20.20 | 30.24  | 52.41  | 39.6225    |
> >     | SVD-LLM-V2     | 48.071  | 81.52  | 47.83     | 72.25 | 66.54 | 32.00 | 36.43  | 65.19  | 56.228875  |
> >     | ASVD          | 43.50   | 62.77  | 43.56     | 67.00 | 51.50 | 26.80 | 36.74  | 62.19  | 49.2575    |
> >
> >
> >     Our observations are as follows:
> >
> >
> >     1. **Stronger performance.**
> >    Across benchmarks, CARE delivers consistently higher accuracy than prior data- or curvature-aware approaches.
> >
> >     2. **Compatibility with existing methods.**
> >    Our non-uniform rank allocation is *orthogonal* to these methods and can be directly combined with them to further improve their compression quality.
> >
> >     3. **Completeness of MLA restoration.**
> >    To fully recover the target MLA structure, we introduce an additional *MLA absorption* procedure (Section 3.5). This step is unique to our method and is not available in prior data-/curvature-aware compression techniques, which results in more complete and faithful MLA reconstruction.
> >
> >     We will incorporate these comparisons and clarifications in the revised version.

---

> ### Author Response · Authors · 2025-11-26
> **Questions**
>
> - How robust is the covariance-aware step if the calibration...
>
>     We acknowledge that when the calibration corpus is distributionally misaligned with the deployment domain (e.g., text-only data for code LMs), some zero-shot performance fluctuation can occur (as shown in Table 1,2 in weakness 1). This behavior is expected: the covariance estimator intrinsically captures activation statistics of the provided calibration data. Nonetheless, the overall rank allocation remains stable across datasets, and the covariance-based adjustment consistently provides larger gains than the minor variance introduced by dataset choice (see Tables in weakness 1).
>
>     However, it is important to emphasize that **CARE without healing is a lossy procedure** and therefore cannot be directly used in an inference setting. Even if one were to collect activations online and update the covariance estimate in a streaming manner, the system would still need to **re-run the CARE decomposition** to obtain updated $W_K$ and $W_V$. This step is computationally expensive and not suitable for continuous or frequent updates during inference (we provide the conversion-time breakdown below).
>
>     For practical deployment, we therefore recommend **offline healing** with a modest amount of domain-representative data. Once healed, the model becomes stable and does not require further online adaptation. In our experiments, such offline healing effectively corrects the lossy artifacts and remains robust even when the calibration set is only partially aligned with the target domain.
>
>
> - Different tasks seemed to benefit differently at low KV ratios...
>
>     Thank you for raising this thoughtful point. The idea of incorporating task-aware weighting into the rank allocator to enable a multi-objective water-filling scheme is both elegant and highly relevant.
>
>     Our current allocator derives ranks from the singular values of the covariance matrix estimated on a single calibration corpus. Extending this to support multi-objective optimization is feasible, and we propose two natural directions:
>
>     1. **Multi-covariance fusion.**
>     One approach—conceptually similar to CoRDA—is to concatenate or jointly factorize the covariance matrices from multiple task distributions, forming a unified spectrum that reflects all objectives. This allows the allocator to naturally “water-fill’’ across tasks based on their combined curvature.
>
>     2. **Weighted calibration mixing.**
>     Alternatively, we can mix calibration datasets in a *task-weighted* manner (e.g., $0.1 \times \text{LiveCode} + 0.9 \times \text{Math-500}$) and compute covariance on the resulting distribution. This provides direct control over per-task influence and can implement the reviewer’s proposed weighting scheme in a simple and flexible way.
>
>     To expose task weights to the rank allocator (different dataset will results in different rank distribution), We adopt second direction and perform some ablation study as below:
>
>     | Method-Rank | Model                   |  Dataset                       | ARC-Challenge | ARC-Easy | HellaSwag | PIQA  | MMLU  | OpenBookQA | RACE  | Winogrande | AVG |
>     |-------------|--------------------------|----------------------------------------|---------------|----------|-----------|-------|-------|-------------|--------|-------------|---------|
>     | CARE-E-256  | Llama-3.1-8B-Instruct    | alpaca                                 | 34.81         | 65.40    | 40.76     | 72.47 | 49.32 | 21.20       | 35.98 | 62.98      | 47.865 |
>     | CARE-E-256  | Llama-3.1-8B-Instruct    | 0.5\*alpaca + 0.5\*arc_challenge         | 36.09         | 67.72    | 43.22     | 73.34 | 58.70 | 24.40       | 34.26 | 63.38      | **50.13875** |
>     | CARE-E-256  | Llama-3.1-8B-Instruct    | 0.2\*alpaca + 0.8\*arc_challenge         | 34.73         | 65.03    | 42.99     | 72.25 | 58.02 | 23.20       | 35.41 | 64.96      | 49.57375 |
>     | CARE-E-256  | Llama-3.1-8B-Instruct    | arc_challenge                          | 39.59         | 71.84    | 41.33     | 72.91 | 52.84 | 26.60       | 32.54 | 62.67      | 50.04 |
>
>
>     We appreciate this suggestion because it aligns well with CARE’s design philosophy: the framework is modular and easily extensible beyond single-distribution calibration. Incorporating multi-objective rank allocation is indeed a promising next step, and we will highlight this direction in the discussion section.
>
>     Thank you again for such a constructive idea—this type of feedback genuinely helps us strengthen the work, and we would be glad to further clarify or discuss if the reviewer is interested.

---

### Author Response · Authors · 2025-12-03
**Summary Part I**

**Dear ICLR 2026 AC,**

We provide below a clear and comprehensive summary of our paper’s core contributions, strengths, and how our rebuttal and follow-up experiments addressed all major reviewer concerns. During the rebuttal, we carefully addressed each reviewer’s feedback, incorporated their suggestions through targeted clarifications and improvements and conducted substantial additional analyses and experiments accordingly.

## **Overall Summary**
Our work introduces CARE (Covariance-Aware and Rank-Enhanced), a practical post-hoc pipeline for converting pretrained MHA/GQA attention into Multi-Head Latent Attention (MLA) under a fixed KV-cache budget. CARE addresses a real deployment problem—how to obtain MLA-style efficiency from existing LLMs without retraining from scratch. Comparing to standard SVD-based approaches (Palu, ASVD, SVD-LLM, TransMLA-V1), CARE achieve 1.12×–1.55× higher zero-shot accuracy across eight benchmarks. Moreover, CARE requires fewer than 2× tokens to fully heal the model—whereas TransMLA-V3 needs $\approx$ 6B tokens for full recovery, and Palu requires $\approx$ 6B tokens for only partial recovery. Reviewers agree **five** aspects of stengths:

1. Care is a meaningful, pratical and novel activation-aware MLA conversion method. Reviewers highlight that our activation-preserving factorization (SVD on curvature-weighted CW rather than on W alone) is fundamentally different from existing uniform-SVD style baselines:
    * Reviewer cgmu: “The insight on performing SVD over activations is indeed a good one.”
    * Reviewer 77dF: “The specific formulation of SVD($CW$) followed by $C^{-1}$ unwhitening is fairly novel.”
    * Reviewer iFpw: “The paper derives the form by considering the activation-space objective, which is more solid.”

2. Care's globally optimal, non-uniform rank allocation under a fixed KV budget. This directly remedies a key limitation of prior work - namely their uniform rank allocation across all MLA layers that over-compress sensitivity-critical layers. Reviewers specifically recognize the global water-filling–based allocation as both empirical motivated and principled:
    * Reviewer 77dF: “The paper also does a good job of combining activation-aware factorization, layer-wise rank scheduling ...”
    * Reviewer 77dF: “A good solution... the empirical observation clearly motivates adaptive allocation.”
    * Reviewer ifpw: “the global and non-uniform rank allocation is motivated by data and the solution is standard and optimal under the budget.”
    * Reviewer cgmu: “I really like observation 1 & 2 and it’s an interesting read.”

3. A cohesive, deployment-ready MLA conversion pipeline. Multiple reviewers acknowledge that CARE goes beyond isolated algorithm changes and instead forms a practical end-to-end MLA conversion workflow, including KV-parity and real KV-bandwidth considerations.
    * Reviewer iFpw: “The design of KV-parity shows the authors think about real deployment constraints.”
    * Reviewer 77dF: “The paper does a good job combining activation-aware factorization, layer-wise scheduling, and KV-parity into a cohesive pipeline.”

4. We extensively evaluated our method across Llama-3.1-8B-Instruct, Llama-3.1-70B-Instruct, Qwen3-4B-Instruct-2507, and Qwen3-30B-A3B-Instruct-2507, comparing against TransMLA (v1, v3), Palu, ASVD, and SVD-LLM-V2 on a wide range of standard benchmarks. These results consistently validate the effectiveness and robustness of our approach. All four reviewers provided positive assessments of the experimental design and empirical findings, as summarized below.
    * Reviewer iFpw: “CARE keeps lower PPL and better task scores than uniform-SVD baselines.”
    * Reviewer 77dF: “Experiments across 8+ benchmarks ... some ablations are well-designed...and they include the correct baselines.”
    * Reviewer cgmu: “The proposed method achieves better performances compared to other baselines.”
    * Reviewer w58u: “Empirically ... superior zero-shot performance compared to TransMLA.”

5. Reviewers also acknowledge the methodological clarity and reproducibility; The paper is well-motivated and clearly written, and the overall method shows strong zero-shot performance, outperforming baselines such as Palu, SVDLLM V2 and TransMLA.
    * Reviewer iFpw: “Reproducibility seems good, with anonymized repo and scripts mentioned.”
    * Reviewer cgmu: “The paper is well-written and easy to understand.”


Below we summarize reviewer-specific resolutions.

---

> ### Author Response · Authors · 2025-12-03
> **Summary Part II**
>
> ## **Reviewer iFpw (rating 6)**
>
> The reviewer raised two main issues:
> 1. Robustness of covariance estimation under covariance' distribution shift and multi-objective covariance fusion ablations
>     * We performed expanded covariance analyses using *domain-distinct* calibration corpora, including **LiveCodeBench** (code-heavy) and **Math-500** (instruction-heavy), to evaluate robustness when calibration and deployment domains differ.
>     * Adopted weighted covariance across different distribution methods. Experiments demonstrate that exposing task weights to the allocator provides predictable rank redistribution across layers.
>     * Added distribution-shift stability analysis showing rank allocation remains well-behaved.
> 2. Comparison with other data-/curvature-aware baselines
>     * We added direct empirical comparisons against **Palu** and **ASVD**, **SVDLLM-V2**, under identical KV budgets, calibration data, and inference configurations.
>     * These results confirm that CARE consistently outperforms existing activation-aware or curvature-weighted SVD approaches. **805 zero-shot large & diverse model evaluations** has been finished.
>
> We responded to all major concerns and provided substantial new experiments. However, due to the accidentally closure of the reviewer discussion system, the reviewer did not have the opportunity to provide further feedback or update their evaluation after our rebuttal.
>
> ## **Reviewer 77dF (rating 6)**
>
> The reviewer raised three concerns, all of which we addressed with strengthened theory, expanded experiments, and targeted ablations:
>
> 1. Theoretical Justification; The reviewer questioned whether using  $\,\|C(W - W_c)\|_F^2\,$  as a surrogate for  $\,\|\sqrt{C}(W - W_c)\|_F^2\,$  might overly emphasize high-curvature directions.
>
>     To resolve this, we added:
>     * Detailed derivations and experiments of 30+ emprical singular values spectral comparison showing that both objectives share identical eigenspaces and lead to same-curve rank profiles. The different between rank sparsity decide $C$ or $\sqrt{C}$ we shall use.
>     * Experiments across damping factor $\,\alpha \in \{0.001, 0.01, 0.1\}\,$ demonstrating **stable performance** and no curvature-dominated collapse.
>     * Additional diagnostics confirming that CARE remains robust under different curvature scalings.
>
> 2. Experimental Completeness; We substantially strengthened the empirical evaluation in a unified and coherent manner, focusing on **three** aspects that directly test generality, fairness, and real-world applicability:
>
>     * We validated CARE across multiple model scales and architectures (Llama-3.1-8B, 70B; Qwen3-4B, 30B-A3B), demonstrating consistent effectiveness regardless of model size or GQA/MHA attention structure.
>     * We incorporated all major baselines—including ASVD, SVD-LLM V2, and Palu—under strictly matched KV budgets and evaluation settings, adding **805** new zero-shot evaluations that provide broad and fair empirical validation.
>     * **long-context & efficiency**  We conducted **five** Needle-in-a-Haystack tests up to 32K tokens, where CARE showed significantly better long-range retention than Palu. We also added explicit token budgets, LR schedules, healing/distillation configs, and detailed profiling of: Covariance + SVD runtime, Memory footprint and `model.generate` latency and throughput.
>
>     These results show that CARE maintains comparable throughput to MLA/SVD baselines while delivering superior accuracy.
>
> 3. To isolate each component’s contribution, we added ablations:
>     * CARE decomposition with uniform rank
>     * Standard SVD with CARE’s adaptive scheduling
>     * Shrinkage sensitivity over $\,\alpha\,$, $\,C\,$, and $\,\sqrt{C}\,$
>
>     These ablations clearly separate the effects of **weighted covariance-aware** and **dynamic rank allocation**, confirming that both components matter.
>
>
> Overall, We addressed these concerns through 2 theoretical analysis, 805 zero-shot large & diverse model evaluations, 5 long-context Needle-in-a-Haystack experiments, 21 conversion-time / memory-footprint profiling experiments 6 8B & 4B model (6B-token) healing / training runs and the reviewer’s follow-up questions in full. Due to the premature closure of the reviewer discussion system, the reviewer was unable to provide further feedback or formally update their evaluation despite indicating strong interest.

---

> > ### Author Response · Authors · 2025-12-03
> > **Summary Part III**
> >
> > ## **Reviewer cgmu (rating 4, but "happy to raise my score" as stated in their original review comments)**
> >
> > We thoroughly addressed three issues from reviewer
> > 1. Removed the informal use of *whitening* and replaced it with a precise curvature-based explanation. Clarified the distinction between statistical covariance and the curvature matrix used in CARE’s objective.
> > 2. Updated **all figures** with larger fonts and improved print readability.
> > 3. Added detailed theoretical and empirical cost analysis. We emphasized that **Eq.(2) is extremely lightweight** (simple water-filling over sorted singular values), and the main cost lies in covariance estimation and SVD—now measured to be moderate ( $\approx$ 3–5GB working memory, one-time offline). We also provided Inference memory usage and throughput (model.generate) in detail.
> >
> > Reviewer cgmu initially gave rating of 4 but in their inital review comments:
> >
> > > “I’m happy to raise my score if the authors address my concerns.”
> >
> > After we addressed these concerns during the discussion, the reviewer also raised a few minor follow-up questions regarding the theoretical time & memory complexity and profiling details, which we also clarified with additional explanations and results. Unfortunately, due to the unexpected early closure of the discussion system, reviewer cgmu was unable to provide their final confirmation or formally update the score, even though our theoretical analysis and empirical profiling demonstrate that the added complexity is minimal which indicate the concern has been resolved.
> >
> > ## **Reviewer w58u (rating 4, but acknowledged concerns addressed and will raise score to 6)**
> >
> > Regarding weakness, the reviewer concers three parts:
> >
> > 1. Comparing CARE with TransMLA, RoRoPE and MLA Absorb; We explicitly distinguished our evaluation of TransMLA v1 from the reviewer-cited v3, which includes RoRoPE, FreqFold, and BKV-PCA—modules not present in the original version we compared against. We added a clear methodological discussion highlighting that CARE’s curvature-weighted decomposition and global rank allocation differ fundamentally from these positional or frequency-alignment modules. We provided explicit clarification showing that CARE’s curvature-based decomposition is distinct from RoRoPE’s positional corrections but can orthogonally integrated. To solve the issue of absorb, we added CARE-100%-MLA mapping pipeline figure to demonstrate complete compatibility.
> > 2. Experimental Completeness; We re-ran all evaluations using TransMLA v3 to . We added comparisons to Palu, ASVD, SVDLLM-V2 for Llama-3.1-8B instruct Qwen3-4B-Instruct-2507 and Qwen3-30B-A3B-Instruct-2507, and discussed MHA2MLA under matched KV budgets, identical calibration data, and identical inference settings.
> > 3. Addressing rank-adaptive scheduling and efficiency; We conducted end-to-end inference profiling, including: memory usage, KV-cache footprint, throughput under `model.generate`, and latency scaling. We also mention that dynamic rank kernels would be significantly more meaningful for CARE, and supporting dynamic-rank MLA kernels is an important direction we are actively exploring.
> >
> > These fully resolve the reviewer’s methodological and evaluation concerns.
> >
> > **Importantly, Reviewer w58u explicitly stated they will raise the score to 6 and wish us good luck.** But due to the early closure of the reviewer discussion system, they were unable to update their score.
> >
> > Their final message, before discussion was closed by the system:
> >
> > > “Considering the contributions of this paper ... I will raise my score to 6.”
> >
> > We sincerely appreciate their acknowledgement.
> >
> > ## **Final Note to the AC**
> >
> > Although the discussion period closed early due to system-wide policy changes, *all* reviewer feedback was addressed with significant new experiments, clarifications, and methodological fixes.
> > * Reviewers 77dF and iFpw already gave **positive scores (both 6)**, and given that their comments have now been resolved, we believe that their positive assessment will be maintained.
> > * Reviewer cgmu stated **they would be happy to raise their score** once their concerns were addressed, but due to the early closure of the reviewer discussion system, they were unable to provide further confirmation or update their score. We are confident that our response fully addresses all concerns through terminology clarification, fair baseline comparisons, detailed conversion-time profiling and complexity analyses.
> > * Reviewer w58u explicitly confirmed that our rebuttal addressed their concerns and stated they will **raise the score to 6 and wish us good luck**. And due to the issue of discussion system, they were unable to update their score.
> >
> > We sincerely thank the reviewers and the AC for careful review and constructive and insightful feedback again. Their comments helped us significantly strengthen the paper.

---

### Meta-Review · Area_Chair_7QTB · 2025-12-28

**Summary:**

Reviewers mention the paper being novel, clearly written, and practical. Authors provide comprehensive evaluations across different LLMs and showcase that the proposed CARE pipeline can improve upon other commonly used methods.

In particular, reviewer cgmu, who gave a 4 rating, asked about cost analysis and more comparison. Reviewer w58u, who also gave a 4 rating, asked about more method comparison and efficiency. The main points were already addressed by the authors in rebuttal. Reviewer w58u would like to increase the score to 6.

**Reviewer Concerns:**

Among all the weaknesses, testing the CARE pipeline on more diverse model architectures in addition to Llama, adding more comparisons to other methods, discussing the efficiency and several format errors were all addressed.

**Reviewer Scores:**

Reviewer w58u already mentioned to increase the score to 6. Given the thorough comparison between CARE and SVD-LLM V2, ASVD and Palu, reviewer cgmu might increase the score.

---

### Decision · Program_Chairs · 2026-01-26

Accept (Poster)